# Higher-Matter and Landau-Ginzburg Theory of Higher-Group Symmetries

Ruizhi Liu[1,2], Ran Luo[3], Yi-Nan Wang[3,4]

[1] *Department of Mathematics and Statistics,*
*Dalhousie University,NS B3H 1Z9, Canada*

[2] *Perimeter Institute of Theoretical Physics,*
*Waterloo, ON N2L 2Y5, Canada*

[3] *School of Physics,*
*Peking University, Beijing 100871, China*

[4] *Center for High Energy Physics, Peking University,*
*Beijing 100871, China*

Higher-matter is defined by higher-representation of a symmetry algebra, such as the $p$-form symmetries, higher-group symmetries or higher-categorical symmetries. In this paper, we focus on the cases of higher-group symmetries, which are formulated in terms of the strictification of weak higher-groups. We systematically investigate higher-matter charged under 2-group symmetries, defined by automorphism 2-representations. Furthermore, we construct a Lagrangian formulation of such higher-matter fields coupled to 2-group gauge fields in the path space of the spacetime manifold. We interpret such model as the Landau-Ginzburg theory for 2-group symmetries, and discuss the spontaneous symmetry breaking (SSB) of 2-group symmetries under this framework. Examples of discrete and continuous 2-groups are discussed. Interestingly, we find that a non-split 2-group symmetry can admit an SSB to a split 2-group symmetry, where the Postnikov class is trivialized. We also briefly discuss the strictification of weak 3-groups, weak 3-group gauge fields and 3-representations in special cases.

# Contents

**7 Landau-Ginzburg model of higher-group symmetries**     **44**

**8 Discussions**     **59**

**A Triviality of Postnikov class after the strictification**     **60**

**B More examples of strictification of weak 2-groups**     **61**

# 1 Introduction

In the recent years, there have been a plethora of approaches to extend the notion of symmetries beyond the group theory paradigm, see the review articles [1–5]. The most general algebraic structure for generalized symmetries is an $n$-category, where the morphisms correspond to symmetry transformations. If these morphisms are all invertible, the algebraic structure is called an $n$-group. Physically, it consists of codimension-1, 2, …, $n$-topological generators of symmetry transformations. Higher-group symmetries have been extensively discussed in physics literature, see e.g. [6–52].

For 2-groups, there are actually two different mathematical formulations that are used in different contexts:

1. *Strict 2-group*, which is defined as a strict 2-category whose morphisms are invertible and associative. Strict 2-groups commonly appears in the mathematical formulations

of 2-group gauge theories [6, 7, 53–55]. Algebraically, a strict 2-group is described by a crossed module $(G, H, \partial, \triangleright)$, as reviewed in section 2.

2. *Weak 2-group*, which is defined as a weak 2-category (bicategory) whose morphisms are invertible but not associative. A weak 2-group is characterized by a tuple $(\Pi_1, \Pi_2, \rho, \beta)$, where $\Pi_1$ and $\Pi_2$ are interpreted as the physical "0-form" and 1-form symmetries, $\rho : \Pi_1 \to \mathrm{Aut}(\Pi_2)$ is a group action and $\beta \in H^3_\rho(B\Pi_1; \Pi_2)$ is called the Postnikov class. When the Postnikov class $\beta = 0(\neq 0)$, the weak 2-group is called split (non-split). In condensed matter physics literature, a split 2-group symmetry is also denoted as "symmetry fractionalization", see e.g. [20, 56, 57]. Weak 2-group is the most commonly used notion of 2-groups in the generalized symmetry literature.

Different strict 2-groups could be equivalent to the same weak 2-group via weak equivalence. On the other hand, for a given weak 2-group $(\Pi_1, \Pi_2, \rho, \beta)$, there are multiple ways to construct a corresponding strict 2-group. Such a choice of strict 2-group (crossed module) is called a *strictification* of the weak 2-group $(\Pi_1, \Pi_2, \rho, \beta)$.

Although the weak 2-group description is more appropriate to define symmetry generators, we find that the strict 2-group description is more convenient in many physical aspects.

**Strictification of gauge fields**   First, when discussing the gauging of 2-group symmetry and constructing a 2-group gauge theory, the gauge transformation rules of strict 2-group gauge theory in terms of the strict gauge fields $(A, B)$ is much simpler than that of weak 2-group gauge theory with weak gauge fields $(a, b)$ [7]. We present a derivation of strict 2-group gauge fields from the weak 2-group counterpart in section 4.

**Higher-matter**   For an ordinary symmetry group $G$, it acts on matter fields $\phi \in V$ in forms of a *representation* $G \to \mathrm{Hom}(V, V)$. More abstractly, a representation is defined as a functor $G \to \mathbf{Vect}$ from the 1-category $G$ into the category of vector spaces $\mathbf{Vect}$.

As a generalization, the (higher-)matter fields charged under higher-group $\mathcal{G}$ global or gauge symmetries are described by a *higher-representation*, defined as a functor $\mathcal{G} \to n\mathbf{Vect}$ from the higher-group to an $n$-category of $n$-vector spaces (also known as the *higher charge*) [43, 58]. The detailed definitions and structures of general higher-representations are open problems in mathematics. For weak 2-groups, the characterization and classification of 2-representations are well studied, see e.g. [43, 45, 59, 60]. Nonetheless, we would still like to ask a question: how to write down higher-matter coupled to higher gauge fields in a more physical, Lagrangian formulation?

It turns out that the higher vector spaces are not convenient for our purpose. In this paper, we instead propose to use the *automorphism 2-representation* of an algebra $A$, to model the 2-representations of 2-groups in the strict description. In particular, we often use the group algebra $A = \mathbb{C}[K]$ for a group $K$, which can correspond to a collection of "Wilson loop" operators physically.

Then there is another question: what is the physical operator on which the 2-group act, in the Lagrangian form?

Our answer is that the proper way for writing down the coupling of 2-group gauge fields with 2-matter is to work in the path space (loop space) $\mathcal{P}(M)$ of the spacetime manifold $M$, which we elaborate in section 6. Again we would utilize the strict 2-group gauge fields, and realizing the 2-matter in terms of an automorphism 2-representation.

**Landau-Ginzburg model for strict 2-group symmetries**    For the physical applications, we establish the Landau-Ginzburg model of 2-group symmetries as a strict 2-gauge theory in the path space in section 7. We discuss the SSB (Spontaneous Symmetry Breaking) of 2-group symmetry using this effective field theory. When the 2-group only contains a 1-form symmetry, our model is reduced to the Mean String Field Theory, which is the Landau-Ginzburg model of 1-form symmetries [61].

We apply the formalism to a simple example of non-split weak 2-group $(\mathbb{Z}_2, \mathbb{Z}_2, \mathrm{id.}, \beta \neq 0 \in H^3(B\mathbb{Z}_2; \mathbb{Z}_2))$, which has a strictification $(\mathbb{Z}_4, \mathbb{Z}_4, \partial = \times 2, 1 \triangleright a = 4 - a)$ (note that we use the additive notation for $\mathbb{Z}_n$ cyclic groups in the paper). Interestingly, we found that in the strict formulation, the non-split 2-group can be SSB to a split 2-group with a trivial Postnikov class, without breaking $\Pi_1 = \mathbb{Z}_2$ or $\Pi_2 = \mathbb{Z}_2$!

We also attempt to discuss the SSB of a continuous non-split weak 2-group $(\mathbb{Z}_N, U(1), \mathrm{id.}, \beta \neq 0 \in H^3(B\mathbb{Z}_N; U(1))$ by the approximation $\lim_{M \to \infty} \mathbb{Z}_M = U(1)$. The SSB behavior of this 2-group Landau-Ginzberg model can break the symmetry to $\Pi_1 = \mathbb{Z}_K$, $\Pi_2 = \mathbb{Z}_P$, and the Postnikov class is broken to $\beta \bmod \gcd(P, K)$. With suitably chosen parameters, we can still realize a symmetry breaking from a non-split 2-group to a split 2-group.

**3-groups**    We have also extended beyond the level of 2-category, and explored the cases of 3-group symmetries, whose physical applications were explored in [16, 22, 62]. In this case, a

*weak 3-group* is formulated by the following Postnikov tower [63]:

$$
\begin{array}{ccc}
B^3\Pi_3 & \rightarrow & B\mathbb{G}_3 \\
 & & \downarrow \\
B^2\Pi_2 & \rightarrow & B\mathbb{G}_2 \\
 & & \downarrow \\
 & & B\Pi_1
\end{array}
\tag{1.1}
$$

and is parameterized by the tuple $(\Pi_1, \Pi_2, \Pi_3, \rho, \beta, \gamma)$. Where $\rho$ is a group action of $\Pi_1$ and $\beta, \gamma$ are two group cohomology elements:

$$
\beta \in H^3(B\Pi_1, \Pi_2) \ , \ \gamma \in H^4(B\mathbb{G}_2, \Pi_3) \,.
\tag{1.2}
$$

For a general weak 2-category, one can always strictify it into a strict 2-category. But for a general weak 3-category, it can only be strictified into a semi-strict (Gray) 3-category [64]. Hence for a weak 3-group, one can only strictify it into a *semi-strict 3-group* in general, which is described by the algebraic structure of 2-crossed-module $(G, H, L, \partial_1, \partial_2, \rhd, \{-, -\})$. We discuss the definitions of these notions of 3-groups and the strictification in the special cases of either $\Pi_1 = 0$ or $\Pi_2 = 0$ in section 3. We have also shortly discussed the 3-representations and physical aspects of strict 3-gauge theories.

## Structure of This Paper

In pursuit of higher-gauge theory with matter under higher representations, we must clarify the algebraic structure of higher-groups (2-group and 3-group), and work out how the weak-category language of predecessors aligns with the strict-category language we predominantly utilize.

In section 2, we briefly review the basic algebraic structure of 2-groups in both strict and weak languages, and discuss their transitions. We also explicitly construct both discrete and continuous examples. More examples can be found in appendix B.

In section 3, we investigate the algebraic structure of semi-strict and weak 3-groups, and construct algebraic models for strictification of weak 3-groups in the special cases of $\Pi_2 = 0$ or $\Pi_1 = 0$.

In section 4, we present a dictionary of gauge fields, gauge transformations and observables between the weak-/strict- categorical languages in 2-group gauge theories. We also build the strictification of 3-gauge theories.

With the algebraic structures and categorical languages clarified, we can define higher-matter through automorphism higher-representations of higher-groups, and fit them into gauge theories.

In section 5, we introduce automorphism 2-representation with explicit examples, and discuss its relation with previous definition of higher-representation. We also extend the automorphism 2-representation to define 3-representation for semi-strict 3-groups with $\Pi_1 = 0$.

In section 6, we build 2-group gauge theories with 2-matter using the path space formalism developed in [54]. We give explicit construction of 2-matter in both continuous and discrete cases using fields in the path space, and discuss $n$-matter forming brane fields.

In section 7, with all the previously defined 2-matter, we construct the Lagrangian Landau-Ginzberg theory of 2-groups in both continuous and discrete languages, and discuss their spontaneously symmetry breaking patterns.

## 2 Strictification of 2-groups

### 2.1 A review of 2-groups

As mentioned in section 1, there are two notions of 2-groups that are used in different references. We review the relation between strict 2-groups and weak 2-groups in this section.

We first write down the formulation of a *strict 2-group* in terms of a crossed module $(G, H, \partial, \rhd)$. $G$ and $H$ are two groups, the map $\partial : H \to G$ is a group homomorphism, i.e. $\partial(h_1 h_2) = (\partial h_1)(\partial h_2)$ for all $h_1, h_2 \in H$, and $\rhd : G \to \mathrm{Aut}(H)$ is an action of $G$ on $H$. In particular, for any $g \in G$, $g \rhd$ induces a group automorphism of $H$, and it is required that $g \rhd 1_H = 1_H$.

Furthermore, for all $g \in G$, $h, h' \in H$, the following equations hold:

$$\partial(g \rhd h) = g(\partial h)g^{-1} \tag{2.1}$$

$$(\partial h) \rhd h' = hh'h^{-1}. \tag{2.2}$$

If we define the action of $G$ on itself $\rhd : G \times G \to G$ as

$$g \rhd g' = gg'g^{-1}, \tag{2.3}$$

(2.1) can be rewritten as

$$\partial(g \rhd h) = g \rhd (\partial h) \tag{2.4}$$

Following [7], a better way to classify a crossed module $(G, H, \partial, \rhd)$ is by the equivalent classes of strict 2-groups, which preserves the groups $\Pi_1 = G/\mathrm{im}(\partial)$ and $\Pi_2 = \ker(\partial)$ instead of $G$ and $H$.

One defines the quadruple $(\Pi_1, \Pi_2, \rho, \beta)$, which is called a *weak 2-group* out of the crossed module $(G, H, \partial, \rhd)$. The action of $\rho : \Pi_1 \times \Pi_2 \to \Pi_2$ is naturally induced from $\rhd : G \times H \to H$.

The non-trivial element $\beta \in H_\rho^3(B\Pi_1; \Pi_2) \cong H_{\mathrm{grp},\rho}^3(\Pi_1; \Pi_2)$ is called the Postnikov class of the weak 2-group. If the group action $\rho$ is trivial, $\beta$ is an element of the group cohomology, otherwise it is an element of the twisted group cohomology with action $\rho$.

Now we want to ask the following question: given a weak 2-group $(\Pi_1, \Pi_2, \rho, \beta)$, how to construct its corresponding strict 2-group $(G, H, \partial, \rhd)$ (which is not unique). If the Postnikov class $\beta$ is trivial, we can simply construct the strict 2-group as

$$G = \Pi_1 \ , \ \ H = \Pi_2 \ , \ \ \partial h = 1_G \ (\forall h \in H) \ , \ \ \rhd = \rho \, . \tag{2.5}$$

If $\beta$ is a non-trivial element, the answer is not obvious, and we need to construct an exact sequence accompanied with an action $\rhd : G \to \mathrm{Aut}(H)$:

$$1 \to \Pi_2 \xrightarrow{\ i\ } H \xrightarrow{\ \partial\ } G \xrightarrow{\ p\ } \Pi_1 \to 1 \, , \tag{2.6}$$

such that it gives the correct group cohomology element $\beta \in H^3(B\Pi_1; \Pi_2)$.

After the strictification, the pullback of the Postnikov class is trivial, $p^*(\beta) = 0$, see appendix A for a proof.

We use the procedure and notations in [65] to compute the Postnikov class $\beta$. First we choose a cross-section function $s : \Pi_1 \to G$, which satisfies $p \circ s = \mathrm{id}..$ The failure of group associativity for $s(g)$ is encoded in a map $f : \Pi_1 \times \Pi_1 \to \ker(p)$, defined as

$$s(g)s(h) = s(gh)f(g, h) \, . \tag{2.7}$$

The function $f(g, h)$ satisfies the 3-cocycle condition

$$s(g)f(h, k)s(g)^{-1}f(g, hk) = f(g, h)f(gh, k) \, . \tag{2.8}$$

Now we uplift $f$ to a function $F : \Pi_1 \times \Pi_1 \to H$ by requiring that $\partial(F(g)) \equiv f(g)$ for all $g \in \Pi_1$. The failure of cocycle condition for $F(g, h)$ can be written in terms of

$$(s(g) \rhd F(h, k))F(g, hk) = i(\beta(g, h, k))F(g, h)F(gh, k) \, , \tag{2.9}$$

where $\beta : \Pi_1 \times \Pi_1 \times \Pi_1 \to \Pi_2$ is the desired Postnikov class in $H^3(B\Pi_1; \Pi_2) \, .$

It is worth noting that if two exact sequences of strictification with the same $(\Pi_1, \Pi_2, \rho)$ can weave a commutative diagram:

$$
\begin{array}{ccc}
& H_1 \xrightarrow{\partial_1} G_1 & \\
\nearrow^{i_1} \ \Big\downarrow{t_H} \quad \Big\downarrow{t_G} \quad \searrow^{p_1} & \\
1 \longrightarrow \Pi_2 \qquad\qquad \Pi_1 \longrightarrow 1 & \\
\searrow_{i_2} \ \Big\downarrow \quad \Big\downarrow \quad \nearrow_{p_2} & \\
& H_2 \xrightarrow{\partial_2} G_2 &
\end{array}
\tag{2.10}
$$

then the two strict two groups renders the same Postnikov class $\beta$, and are thus equivalent up to the classification by $(\Pi_1, \Pi_2, \rho, \beta)$ [65].

We have to emphasize the point that middle arrows above (i.e. $t_H$, $t_G$) are not necessarily isomorphisms. In this case, there is no "inverse" of above morphism between 2-groups in general. In mathematics, this equivalence relation is called weak equivalence.

As a consequence, given a weak 2-group, we can strictify it into many different crossed modules. Although these crossed modules have the same Postnikov class, there might be no weak equivalence between them.

For the later application to gauge theory, to construct a sensible 2-bundle, we require that $\text{im}(\partial) \subset Z(G)$ ($Z(G)$ is the center of $G$), see (4.2.2).

## 2.2    A general procedure of strictification

In this section, we describe a general procedure of strictification that works for (finitely generated) abelian groups as well as $U(1)$. This construction is a slight generalization of [66].

Let $c \in H^3(B\Pi_1; \Pi_2)$ be a cocycle that is additive w.r.t the first argument. Let us denote $A = \text{Hom}(\Pi_1; \Pi_2)$ in this section. Thus, $\omega$ determines a 2-cocycle $Z \in H^2(B\Pi_1; A)$, that is

$$Z(g_2, g_3)(g_1) := c(g_1, g_2, g_3). \tag{2.11}$$

Hence $Z$ determines a central extension

$$1 \to A \to G \to \Pi_1 \to 1. \tag{2.12}$$

Locally, $G$ can be described by a pair $(\gamma, g) \in A \times \Pi_1$. However, the group operation of $G$ is twisted by $Z$ as

$$(\gamma, g) \cdot (\gamma', g') = (\gamma \gamma' Z(g, g'), gg'). \tag{2.13}$$

In the next step, we fix an extension

$$1 \to \Pi_2 \to H \to A \to 1. \tag{2.14}$$

It turns out that the choice of $H$ is rather arbitrary, and we can simply choose $H = \Pi_2 \times A$ and use the trivial extension[1]. Now the decomposition $H \to A \to G$ gives a desired boundary map $\partial$:

$$\partial(b, \gamma) = (\gamma, 0). \tag{2.15}$$

$0$ is the identity element of $\Pi_1$. The group action of $G$ on $H$ can be described as ($\Pi_2$ is additive)

$$(\gamma, g) \triangleright (b, \gamma') = (b + \gamma'(g), \gamma'). \tag{2.16}$$

---

[1] Other choices are also possible, but following steps must be modified as well.

Now we show that above construction indeed gives our desired Postnikov class. It's easy to see that

$$F(g_1, g_2) = (0, Z(g_1, g_2)) \,. \tag{2.17}$$

Now we have

$$\beta(g_1, g_2, g_3) = (Z(g_2, g_3)(g_1), \delta Z(g_1, g_2, g_3)) = (\beta(g_1, g_2, g_3), 0) \tag{2.18}$$

where we used the fact $\delta Z = 0$ (cocycle condition of $Z$).

So the only question is, how general it is to assume that cocycle $\beta \in H^3(B\Pi_1; \Pi_2)$ is additive (w.r.t. at least one argument). Actually, it is indeed the case provided $\Pi_1$ is finitely generated abelian groups or their product of $U(1)$. This can be deduced by investigating the structure of cohomology ring and the additivity of generators.

In the following subsections, we present three examples of weak 2-groups and discuss their strictifications. We show more examples in appendix B.

## 2.3   Example: $\Pi_1 = \Pi_2 = \mathbb{Z}_2$

The first example is a case of discrete 2-group, with $\Pi_1 = \mathbb{Z}_2$, $\Pi_2 = \mathbb{Z}_2$. Since $H^3(B\mathbb{Z}_2; \mathbb{Z}_2) = \mathbb{Z}_2$, there is only one non-trivial element $\beta \neq 0 \in H^3(B\mathbb{Z}_2; \mathbb{Z}_2)$.

Because $\text{Aut}(\mathbb{Z}_2) = 1$, the action of $\Pi_1 = \mathbb{Z}_2$ on $\Pi_2 = \mathbb{Z}_2$ can only be taken as the trivial one.

For the strictification, let us choose $G = H = \mathbb{Z}_4$ and the following exact sequence:

$$1 \to \mathbb{Z}_2 \xrightarrow{i} \mathbb{Z}_4 \xrightarrow{\partial} \mathbb{Z}_4 \xrightarrow{p} \mathbb{Z}_2 \to 1 \tag{2.19}$$

The maps are

$$i: \ a \to 2a \,, \quad \partial: \ a \to 2a \,, \quad p: \ (\text{mod } 2) \,. \tag{2.20}$$

Note that we are using the additive notation for the group elements of $\mathbb{Z}_n$.

Besides the exact sequence, we also need to specify the action $\rhd : G \times H \to H$, which trivializes after restricted to $\rhd : \Pi_1 \times \Pi_2 \to \Pi_2$.

We show that there are two different choices of $\rhd$, which realizes the two different elements of $H^3(B\mathbb{Z}_2; \mathbb{Z}_2) = \mathbb{Z}_2$.

In the computation of the Postnikov class $\beta$, we choose the cross-section $s : \Pi_1 \to G$ as the identity map $s(g) \equiv g$. Then the function $f : \Pi_1 \times \Pi_1 \to \ker(p)$ defined as

$$s(g)s(h) = f(g, h)s(gh) \tag{2.21}$$

takes the following values:

$$f(0,0) = f(0,1) = f(1,0) = 0 \ , \ f(1,1) = 2 \,. \tag{2.22}$$

After lifting $f$ to a function $F : \Pi_1 \times \Pi_1 \to H$ by requiring that $\partial(F(g,h)) \equiv f(g,h)$, we can choose

$$F(0,0) = F(0,1) = F(1,0) = 0 \ , \ F(1,1) = 1 \,. \tag{2.23}$$

The failure of cocycle condition for $F(g,h)$ can be written in terms of

$$(s(g) \triangleright F(h,k))F(g,hk) = i(\beta(g,h,k))F(g,h)F(gh,k) \,, \tag{2.24}$$

where $\beta : \Pi_1 \times \Pi_1 \times \Pi_1 \to \Pi_2$ is the desired Postnikov class in $H^3(B\Pi_1; \Pi_2)$.

Now there are the following two choices of $\triangleright$:

1. Trivial action: $a \triangleright b = b$ for any $a \in G, b \in H$. In this case, one can check that

$$F(h,k)F(g,hk) = F(g,h)F(gh,k) \tag{2.25}$$

   holds for every $g, h, k \in \Pi_1$. Hence we have a trivial Postnikov class $\beta = 0 \in H^3(B\mathbb{Z}_2; \mathbb{Z}_2) = \mathbb{Z}_2$.

2. Non-trivial action: $a \triangleright b = (2a+1)b \pmod 4$. One can check that the action is indeed trivial for $b \in \mathrm{im}(i) \subset H$.

   In this case, we can compute the evaluation table for $\beta(g,h,k)$:

$$
\begin{array}{c|ccc}
\beta(g,h,k) & g & h & k \\
\hline
0 & 0 & 0 & 0 \\
0 & 0 & 0 & 1 \\
0 & 0 & 1 & 0 \\
0 & 0 & 1 & 1 \\
0 & 1 & 0 & 0 \\
0 & 1 & 0 & 1 \\
0 & 1 & 1 & 0 \\
1 & 1 & 1 & 1 \\
\end{array}
\tag{2.26}
$$

Hence $\beta(g,h,k)$ leads to the non-trivial Postnikov class $\beta = 1 \in H^3(B\mathbb{Z}_2; \mathbb{Z}_2) = \mathbb{Z}_2$.

## 2.4 Example: $\Pi_1 = \mathbb{Z}_N$, $\Pi_2 = U(1)$

We take an example where $\Pi_1$ is discrete and $\Pi_2$ is continuous. When $\Pi_1 = \mathbb{Z}_N$, $\Pi_2 = U(1)$, we can choose the following exact sequence

$$1 \to U(1) \xrightarrow{i} U(1) \times \mathbb{Z}_N \xrightarrow{\partial} \mathbb{Z}_N.\mathbb{Z}_N \xrightarrow{p} \mathbb{Z}_N \to 1 \tag{2.27}$$

We use additive notations for $\mathbb{Z}_N$ and multiplicative notations for $U(1)$. $G = \mathbb{Z}_N.\mathbb{Z}_N$ denotes a generally non-split group extension of $\mathbb{Z}_N$ by $\mathbb{Z}_N$, and its group elements are in form of $(a, b)$, $(0 \le a, b < N)$. The group elements of $H = U(1) \times \mathbb{Z}_N$ are $(e^{2\pi i a}, b)$, $(0 \le a < 1$ , $0 \le b < N)$.

The maps are

$$i : \ e^{2\pi i a} \to (e^{2\pi i a}, 0) \,, \quad \partial : (e^{2\pi i a}, b) \to (b, 0) \,, \quad p : (a, b) \to b \,. \tag{2.28}$$

The group action $\triangleright : G \to \mathrm{Aut}(H)$ is

$$(a, b) \triangleright (e^{2\pi i c}, d) = (e^{2\pi i(c + bd/N)}, d) \,. \tag{2.29}$$

The Postnikov class element

$$m \in H^3(B\mathbb{Z}_N; U(1)) = \mathbb{Z}_N \tag{2.30}$$

is encoded in the group operation on $G = \mathbb{Z}_N.\mathbb{Z}_N$, which takes the form of

$$(a, b) + (c, d) = \begin{cases} (a + c, b + d) & (b + d < N) \\ (a + c + m, b + d) & (\text{otherwise}) \end{cases} \,. \tag{2.31}$$

Hence for different Postnikov class elements, in order to strictify the weak 2-group, we should choose different groups $G$ with $N^2$ elements. If $m = 0$, the 2-group is split and we have $G = \mathbb{Z}_N \times \mathbb{Z}_N$.

## 2.5   Example: $\Pi_1 = \Pi_2 = U(1)$

We also discuss the case of continuous weak 2-group of $\Pi_1 = \Pi_2 = U(1)$. This case was extensively studied in [9]. where the weak 2-group symmetry $(U(1), U(1), 1, m)$ is denoted as $U(1)^{(0)} \times_m U(1)^{(1)}$. The element of $H^3(BU(1); U(1)) = \mathbb{Z}$ is characterized by an integer $m$. We can use the following exact sequence

$$1 \to U(1) \xrightarrow{i} U(1) \times \mathbb{Z} \xrightarrow{\partial} \mathbb{Z}.U(1) \xrightarrow{p} U(1) \to 1 \tag{2.32}$$

We use additive notations for $\mathbb{Z}$ and multiplicative notations for $U(1)$.

The maps are

$$i : \ e^{2\pi i a} \to (e^{2\pi i a}, 0) \,, \quad \partial : (e^{2\pi i a}, b) \to (b, 1) \,, \quad p : (a, e^{2\pi i b}) \to e^{2\pi i b} \,. \tag{2.33}$$

The group action $\triangleright : G \to \mathrm{Aut}(H)$ is

$$(a, e^{2\pi i b}) \triangleright (e^{2\pi i c}, d) = (e^{2\pi i(c + bd)}, d) \,. \tag{2.34}$$

The Postnikov class element

$$m \in H^3(BU(1); U(1)) = \mathbb{Z} \tag{2.35}$$

is encoded in the group operation on $G = \mathbb{Z}.U(1)$, which takes the form of

$$(a, e^{2\pi i b}) + (c, e^{2\pi i d}) = \begin{cases} (a + c, e^{2\pi i (b+d)}) & (b + d < 1) \\ (a + c + m, e^{2\pi i (b+d)}) & \text{(otherwise)} \end{cases} . \tag{2.36}$$

# 3  Strictification of 3-groups

## 3.1  Semi-strict 3-group as a 2-crossed-module

In this section we discuss the strictification of weak 3-groups. Let us review the notion of a *semi-strict* 3-group in terms of a 2-crossed-module: $(G, H, L, \partial_1, \partial_2, \triangleright, \{-, -\})$, which has the following components (see e.g. the appendix of [16]):

1. $G$, $H$ and $L$ are groups.

2. $\partial_1 : H \to G$ and $\partial_2 : L \to H$ are group homomorphisms, which means that $\partial_1(h_1 h_2) = (\partial_1 h_1)(\partial_1 h_2)$ for all $h_1, h_2 \in H$ and $\partial_2(l_1 l_2) = (\partial_2 l_1)(\partial_2 l_2)$ for all $l_1, l_2 \in L$. Furthermore, they satisfy

$$\partial_1 \circ \partial_2(l) = 1_G \tag{3.1}$$

   for all $l \in L$.

3. There are group actions $\triangleright : G \times G \to G$, $\triangleright : G \times H \to H$, $\triangleright : G \times L \to L$, where the first one is

$$g \triangleright g' = g g' g^{-1} \tag{3.2}$$

   for all $g, g' \in G$.

4. $G$-equivariance of $\partial_1$, $\partial_2$: for all $g \in G$, $h \in H$, $l \in L$,

$$\partial_1(g \triangleright h) = g \triangleright (\partial_1 h) , \ \partial_2(g \triangleright l) = g \triangleright (\partial_2 l) . \tag{3.3}$$

5. Finally, there is a map called Peiffer lifting: $\{-, -\} : H \times H \to L$, which satisfies for any $h_{1,2,3} \in H$, $l, l_1, l_2 \in L$:

$$\begin{aligned}
\partial_2\{h_1, h_2\} &= h_1 h_2 h_1^{-1} (\partial_1 h_1) \triangleright h_2^{-1} , \\
g \triangleright \{h_1, h_2\} &= \{g \triangleright h_1, g \triangleright h_2\} , \\
\{\partial_2 l_1, \partial_2 l_2\} &= l_1 l_2 l_1^{-1} l_2^{-1} , \\
\{h_1 h_2, h_3\} &= \{h_1, h_2 h_3 h_2^{-1}\}(\partial_1 h_1) \triangleright \{h_2, h_3\} , \\
\{h_1, h_2 h_3\} &= \{h_1, h_2\}\{h_1, h_3\}\{\partial_2\{h_1, h_3\}^{-1}, (\partial_1 h_1) \triangleright h_2\} , \\
\{\partial_2 l, h\}\{h, \partial_2 l\} &= l(\partial_1 h) \triangleright l^{-1} .
\end{aligned} \tag{3.4}$$

A 3-group $(G, H, L, \partial_1, \partial_2, \triangleright, \{-, -\})$ contains a 2-group $(H, L, \partial_2, \triangleright')$ as a subgroup, where the action $\triangleright'$ is defined as

$$h \triangleright' h' = hh'h^{-1}$$
$$h \triangleright' l = l\{\partial_2 l^{-1}, h\}$$

(3.5)

for $h, h' \in H$ and $l \in L$.

If we want to realize the 0-form, 1-form and 2-form symmetry of a physical theory using a 2-crossed-module. It is natural to let $H$ and $L$ be abelian, which simplifies a lot of axioms in the previous section. Furthermore, similar to the classification of 2-groups, the physical symmetry group is given by a *weak 3-group*, with the group components

$$\Pi_1 = G/\mathrm{im}(\partial_1) \ , \ \ \Pi_2 = \ker(\partial_1)/\mathrm{im}(\partial_2) \ , \ \ \Pi_3 = \ker(\partial_2) \,.$$

(3.6)

The maps $\partial_1$ and $\partial_2$ becomes trivial when acting on $\Pi_2$ and $\Pi_3$: $\partial_1 h = 1_{\Pi_1}$ for any $h \in \Pi_2$, $\partial_2 l = 1_{\Pi_2}$ for any $l \in \Pi_3$.

In general, the classification of weak 3-groups is based on the following Postnikov tower [63]:

$$\begin{array}{ccc} B^3\Pi_3 & \rightarrow & B\mathbb{G}_3 \\ & & \downarrow \\ B^2\Pi_2 & \rightarrow & B\mathbb{G}_2 \\ & & \downarrow \\ & & B\Pi_1 \end{array}$$

(3.7)

Here $\mathbb{G}_3$ is the structure group of the whole 3-group, and $\mathbb{G}_2 \subset \mathbb{G}_3$ is the structure group involving $\Pi_1$ and $\Pi_2$.

A general weak 3-group $(\Pi_1, \Pi_2, \Pi_3, \rho, \beta, \gamma)$ is characterized by two group cohomology elements

$$\beta \in H^3(B\Pi_1; \Pi_2) \ , \ \ \gamma \in H^4(B\mathbb{G}_2; \Pi_3) \,.$$

(3.8)

The group homology $H^3(B\Pi_1; \Pi_2)$ characterizes the equivalence class of the following exact sequence:

$$0 \rightarrow \Pi_2 \rightarrow H/(\mathrm{im}\partial_2) \overset{\partial_1}{\rightarrow} G \rightarrow \Pi_1 \rightarrow 0 \,.$$

(3.9)

This sequence is exact because $\partial_1 \circ \partial_2(l) = 1_G$ always holds for any $l \in L$, hence the image of the map $\partial_1 : H/(\mathrm{im}\partial_2) \rightarrow G$ is exactly the same as the image of the map $\partial_1 : H \rightarrow G$.

## 3.2 Weak 3-groups with $\Pi_2 = 0$

The strictification of a general weak 3-group is involved. Here we first discuss the special case where $\Pi_2 = 0$. Physically, this case can arise from gauging a finite subgroup of 0-form

symmetry in 4d in presence of mixed 't Hooft anomaly [67]. In this case, as $\ker(\partial_1)/\mathrm{im}(\partial_2) = \Pi_2 = 0$ in (3.6), the following sequence is exact:

$$1 \to \Pi_3 \to L \xrightarrow{\partial_2} H \xrightarrow{\partial_1} G \to \Pi_1 \to 1 \,. \tag{3.10}$$

Thus the equivalence class is characterized by a degree 4 cohomology class $c \in H^4(B\Pi_1; \Pi_3)$. We can also assume $c$ is additive with respect to the first argument, which holds if $\Pi_1$ is product of finitely generated abelian groups and $U(1)$-factors. Let us again denote $A := \mathrm{Hom}(\Pi_1; \Pi_3)$. Then we have a degree 3 cocycle $Z \in H^3(B\Pi_1; A)$, which is given by

$$Z(g_2, g_3, g_4)(g_1) := c(g_1, g_2, g_3, g_4) \,. \tag{3.11}$$

Now $Z$ determines a 2-group, and we can translate it into a crossed module as in the section 2.2:

$$1 \to A \to H \xrightarrow{\partial_1} G \xrightarrow{p} \Pi_1 \to 1 \,. \tag{3.12}$$

In addition, we fix a trivial exact sequence

$$1 \to \Pi_3 \to L \to A \to 1 \tag{3.13}$$

where $L := \Pi_3 \times A$. Similarly, the $G$-action on $L$ factorizes through $\Pi_1$. Explicitly, if $g \in G$

$$g \triangleright (b, \gamma) := (b + \gamma(p(g)), \gamma) \tag{3.14}$$

where $(b, \gamma) \in L = \Pi_3 \times A$ and $\Pi_3$ is additive.

At last, the boundary map $\partial_2$ is given by composition $L \to A \to H$. Hence we obtain a 2-crossed module with trivial Peiffer lifting. Triviality of Peiffer lifting is completely due to $\Pi_2 = 0$.

It is easy to verify that this 2-crossed module has the desired Postnikov class. Apparently, similar procedure works for $H^{n+2}(B\Pi_1; \Pi_{n+1})$ provided $\Pi_1$ satisfies our assumption (i.e. it is a product of finitely generated abelian group and $U(1)$ factors).

**Example:** $\Pi_1 = \mathbb{Z}_2$, $\Pi_2 = 0$, $\Pi_3 = \mathbb{Z}_2$   In this case, $A = \mathrm{Hom}(\mathbb{Z}_2; \mathbb{Z}_2) = \mathbb{Z}_2$, and the elements of $A$ are

$$\begin{aligned} f_0 \in A : \quad & f_0(x) := 0 \,, \\ f_1 \in A : \quad & f_1(x) := x \,. \end{aligned} \tag{3.15}$$

Let us construct the crossed module (3.12) which is a strictification of 2-group ($\Pi_1 = \mathbb{Z}_2$, $A = \mathbb{Z}_2$, $0$, $Z$):

$$1 \to \mathbb{Z}_2 \to \mathbb{Z}_4 \xrightarrow{\partial_1} \mathbb{Z}_4 \xrightarrow{p} \mathbb{Z}_2 \to 1 \,. \tag{3.16}$$

As in the strictification of 2-group in section 2.3, we take $G = H = \mathbb{Z}_4$, $\partial_1 : a \to 2a$, $p : (\text{mod } 2)$. Now there are two different group actions $\triangleright : G \to \text{Aut}(H)$, leading to different $Z \in H^3(B\Pi_1; A)$ and finally different Postnikov classes $c \in H^4(B\Pi_1; \Pi_3)$ for the weak 3-group:

1. Trivial group action $g \triangleright h = h$, giving rise to the trivial element $0 \in H^3(B\Pi_1; A) = \mathbb{Z}_2$. In this case, from (3.11), since $Z(g_2, g_3, g_4) = f_0$ for all $g_2, g_3, g_4 \in \Pi_1$, we have $c(g_1, g_2, g_3, g_4) = 0$ for all $g_1, g_2, g_3, g_4 \in \Pi_1$, hence the weak 3-group $(\mathbb{Z}_2, 1, \mathbb{Z}_2, \text{id.}, 0, c_0)$ has a trivial Postnikov class $c = c_0 \in H^4(B\Pi_1; \Pi_3) = \mathbb{Z}_2$.

2. Non-trivial group action $g \triangleright h = (2g + 1)h$, giving rise to the non-trivial element $\beta \in H^3(B\Pi_1; A) = \mathbb{Z}_2$. In this case, we have

$$Z(g_2, g_3, g_4) = \begin{cases} f_1 & g_2 = g_3 = g_4 = 1 \\ f_0 & \text{other cases} \end{cases}. \tag{3.17}$$

Hence

$$c(g_1, g_2, g_3, g_4) = \begin{cases} 1 & g_1 = g_2 = g_3 = g_4 = 1 \\ 0 & \text{other cases} \end{cases}. \tag{3.18}$$

This corresponds to the weak 3-group $(\mathbb{Z}_2, 1, \mathbb{Z}_2, \text{id.}, 0, c_1)$ with non-trivial Postnikov class $c_1 \in H^4(B\Pi_1; \Pi_3) = \mathbb{Z}_2$. Note that in this case $B\mathbb{G}_2 \equiv B\Pi_1$ since $\Pi_2 = 0$.

Finally, for either of these two cases, from (3.13) we have $L = \mathbb{Z}_2 \times \mathbb{Z}_2$, and the group action $\triangleright : G \to \text{Aut}(L)$ is defined as (3.14):

$$g \triangleright (a, b) = (a + f_b(g \bmod 2), b). \tag{3.19}$$

## 3.3 An algebraic model of weak 3-groups with $\Pi_1 = 0$

In this section, we provide an algebraic model of weak 3-groups with $\Pi_1 = 0$, which is more convenient for computations.

Before that, we need to recall the notion of quadratic functions/forms.

**Definition 3.1.** *Given $A$ and $B$ are abelian groups, a function on $A$ (valued in $B$) is said to be quadratic if there is a bilinear function $F : A \times A \to B$ such that*

$$f(x) = F(x, x), \quad x, y \in A. \tag{3.20}$$

The key point is following result by Eilenberg and Maclane [68]

**Theorem 1.** *The cohomology class $H^4(B^2\Pi_2; \Pi_3)$ is in one to one correspondence with $\Pi_3$-valued quadratic functions on $\Pi_2$.*

Given an abelian group $A$, there is an associated group $\Gamma(A)$ called the universal quadratic group [11] (unique) with a (non-unique) map $\gamma : A \to \Gamma(A)$ such that for any quadratic function $f$ on $A$ (values in $B$), there is a unique homomorphism $\tilde{f} : \Gamma(A) \to B$ with $f = \tilde{f} \circ \gamma$.

In this sense, we only have to consider the universal quadratic groups.

We have following theorem:

**Theorem 2.**  *1. If $A = \mathbb{Z}_r$ with even $r$, then $\Gamma(A) = \mathbb{Z}_{2r}$ and $\gamma(1) = 1$.*

*2. If $A = \mathbb{Z}_r$ with $r$ odd, then $\Gamma(A) = \mathbb{Z}_r$ with $\gamma(1) = 1$.*

*3. for finite abelian group $A = \bigoplus_i^n A_i$, we have*

$$\Gamma(A) = \bigoplus_i \Gamma(A_i) \bigoplus_{i<j} A_i \otimes A_j \tag{3.21}$$

*where $\otimes$ stands for the tensor product of $\mathbb{Z}$-modules.*

**Example 1.** *Let's consider $A = \mathbb{Z}_4$ and $B = \mathbb{Z}_4$. In this case, the bilinear forms $F_k : A \times A \to B$ are given by $F_k(x,y) = kxy$ $(k = 0, 1, 2, 3)$. The associated quadratic form $f_k : A \to B$ are classified as $f_k(x) = kx^2$ $(k = 0, 1, 2, 3)$.*

Next, we strictify the weak 3-group with $\Pi_1 = 0$ to semi-strict 3-group, i.e. the 2-crossed module. We claim that the choice

$$(G, H, L, \partial_1, \partial_2, \triangleright, \{-, -\}) = (0, \Pi_2, \Pi_3, 0, 0, \mathrm{id.}, F) \tag{3.22}$$

will do the job, where $F$ is the bilinear form defined by (3.20). Note that this choice ensures that $H$ and $L$ are abelian groups. Especially this construction will satisfy the axioms of Peiffer lifting listed in (3.4), as we will check below.

The first axiom that $\partial_2\{h_1, h_2\} = h_1 h_2 h_1^{-1} (\partial_1 h_1) \triangleright h_2^{-1}$ follows trivially because both sides are 1. This follows from that $\partial_2 = 0$ and $H$ is abelian.

The second axiom states that $g \triangleright \{h_1, h_2\} = \{g \triangleright h_1, g \triangleright h_2\}$, which is also obviously true since $G = 1$ by our choice so it acts trivially on $H, L$.

The third axiom $\{\partial_2 l, \partial_2 l_2\} = l_1 l_2 l_1^{-1} l_2^{-1}$ is true because $\partial_2 = 0$ and $L$ is abelian.

The fourth property and fifth axiom is ensured by bilinearity of $F$ and abelian nature of $H$ and $L$.

The sixth, the last axiom $\{\partial_2 l, h\}\{h, \partial_2 l\} = l(\partial_1 h) \triangleright l^{-1}$ is ensured by $\partial_1 = 0 = \partial_2$.

So the choice $(0, \Pi_2, \Pi_3, 0, 0, \mathrm{id.}, F)$ does give us a 2-crossed module, which is semi-strict 3-group.

# 4 Strictification on higher gauge fields

In most of the physics literature, people prefer to work with weak 2-groups as they describe "physical" global symmetries. However, in mathematics literature, people prefer to construct 2-gauge theories with strict 2-groups, e.g. [54]. Based on the fact that strict version of 2-groups is equivalent to weak 2-groups [53], we should expect that gauge theories based on strict or weak 2-groups should be somehow equivalent as well. In this section, we obtain an explicit relation between "strict gauge fields" and "weak gauge fields". We derive the Green-Schwarz shift in [7, 9]. Furthermore, we will consider this equivalence at the level of observables, and extend the discussions to the cases of 3-groups. In section 4.1 we focus on the cases of discrete 2-groups,

## 4.1 The dictionary of discrete gauge fields

In this section we will perform the strictification on the level of gauge field content. In the following discussion, we will use Čech cohomology to formulate bundles.

For simplicity, we work with discrete groups. Given following crossed module extension:

$$1 \to \Pi_2 \xrightarrow{i} H \xrightarrow{\partial} G \xrightarrow{p} \Pi_1 \to 1 \tag{4.1}$$

with the group action $\rhd : G \to \mathrm{Aut}(H)$ and the associated Postnikov class $\beta \in H^3(B\Pi_1; \Pi_2)$. We will collect these data $(G, H, \partial, \rhd)$ and denote it as the 2-group $\mathcal{G}$. We introduce the following notations:

1. $M$ is our $n$-dimensional spacetime.

2. $a, b$ are "weak" gauge fields on $M$ with gauge groups $\Pi_1^{[0]}$ and $\Pi_2^{[1]}$ respectively, where the superscript means that $\Pi_2$ acts as a 1-form symmetry group and $\Pi_1$ is a 0-form symmetry group. We will model $a$ as a 1-cochain and $b$ as a 2-cochain. Similarly $A$ and $B$ will be the gauge fields of $G$ and $H$ respectively and again, $A$ is a 1-cochain, $B$ is a 2-cochain.

3. The Čech differential will be denoted as $\delta$, and the differential twisted by $A$ will be $\delta_A$.

4. $B\mathcal{G}$ will be the classifying space of our 2-group [69].

5. Some notations of crossed module extension will be the same as section 2.1.

Gauge fields of discrete gauge group can play many roles, including classifying map, or transition functions of the bundle (equivalently, lattice gauge field defined on the nerve of

trivialization charts), see e.g. [5] for more details. Hence we will not distinguish the gauge field $a$ and the classifying map $M \to B\Pi_1$ for the $\Pi_1$ bundle over $M$.

We have the following commutative diagram:

$$
\begin{array}{ccc}
M & \xrightarrow{\ f\ } & B\mathcal{G} \xleftarrow{\ Bi\ } B^2\Pi_2 \\
& {}^{a}\searrow & \downarrow{\scriptstyle Bp} \\
& & B\Pi_1
\end{array}
\tag{4.2}
$$

The right hand part is the Postnikov tower associated to the 2-group $\mathcal{G}$. It means that whenever we have a 2-bundle $f : M \to B\mathcal{G}$, we automatically get a principal $\Pi_1$ bundle $a : M \to B\Pi_1$, which implies that

$$
\delta a = 1_{\Pi_1} \, .
\tag{4.3}
$$

This is the cocycle condition of the transition function. However, we should not expect $\delta b = 1_{\Pi_2}$ since the image of the map $f$ does not fall into $B^2\Pi_2$ in general. Let us choose a lift of $s : \Pi_1 \to G$ that satisfies $p \circ s = 1$, and define $A = s(a)$. In fact, our construction does not depend on the choice of the lift. Different choices of $s$ result in a "1-form" gauge transformation, as will be clear in section 4.2.

We apply the Čech differential $\delta$ on $A = s(a)$:

$$
p(\delta A) = \delta p(A) = \delta a = 1_{\Pi_1} \, ,
\tag{4.4}
$$

hence $\delta A = \partial B$ for some $B \in H$ (note that $\ker(p) = \mathrm{im}(\partial)$). This is the exactly the flat condition of fake curvature described in [54].

On the other hand we have $A = s(a) = a^*(s)$, where $a^*$ is the pullback by the map $a$. Hence $\delta A = a^*(\delta s) = a^*(\partial F)$, where $F$ is the one in (2.9). Using

$$
\partial(a^*F) = \partial(B) \, ,
\tag{4.5}
$$

we finally get

$$
b = B^{-1}(a^*F) \in \ker\partial \, .
\tag{4.6}
$$

Note that there exists the 2-truncation condition for our case $\delta_A B = 1_H$ [54], since we have a 2-group rather than a 3-group (there is no 3-morphism). Now we apply the twisted differential $\delta_A$ on both sides of (4.6):

$$
\delta_A b = \delta_A(a^*F) = a^*(\delta_A F) \, .
\tag{4.7}
$$

Recall that $\delta_A F = \beta \in H^3(B\Pi_1; \Pi_2)$ (2.9). Furthermore, since $b \in \Pi_2$, which is an element of the central subgroup of $H$, the group action by $A$ on $b$ will descend to an action by $a = p(A)$. Hence we can write

$$
\delta_a b = a^*\beta \, .
\tag{4.8}
$$

This is the Green-Schwarz shift occured in [7]. We demand the equation (4.8) to be covariant under the gauge transformation of $a$, i.e. after $a \to a\delta\lambda$, we need to impose $b \to b\gamma$, where $\gamma$ satisfies

$$\delta_a\gamma = (\delta\lambda)^*F\,. \tag{4.9}$$

This equation determines the gauge transformation of weak gauge fields and explain how the gauge transformations in [9] arise.

## 4.2 Gauge transformations and consistency

In this section, we discuss the gauge transformations of gauge field $A$ and $B$. In particular, we will write down a discrete analogy of the gauge transformations for Lie 2-gauge theory in [54].

We will work with Čech cohomology, and we need to choose trivialization charts $\{U_i\}_{i\in\mathcal{I}}$ ($\mathcal{I}$ is the set of indices). We define $A_{ij}$ as the locally constant gauge field $A$ on $U_i \cap U_j$, similarly $B_{ijk}$ is that of $B$ on $U_i \cap U_j \cap U_k$.

Let us summarize the dictionary between strict and weak gauge fields derived in section 4.1:

$$\begin{aligned} A_{ij} &= s(a_{ij})\,, \\ \partial B_{ijk} &= (\delta A)_{ijk}\,, \\ b_{ijk} &= B_{ijk}^{-1}F(a_{ij}, a_{jk})\,. \end{aligned} \tag{4.10}$$

Here $s : \Pi_1 \to G$ is a section and see (2.9) for the definition of $F$.

### 4.2.1 0-form gauge transformations

First, we expect

$$A_{ij} \to \lambda_i A_{ij} \lambda_j^{-1} \tag{4.11}$$

where $\lambda_i$ is a $G$-valued function on $U_i$. To preserve $\delta A = \partial B$, note that $(\delta A)_{ijk} = A_{ij}A_{jk}A_{ik}^{-1}$, we have

$$(\delta A)_{ijk} \to \lambda_i(\delta A)_{ijk}\lambda_i^{-1}\,. \tag{4.12}$$

Hence we expect

$$(\partial B)_{ijk} \to \lambda_j(\partial B)_{ijk}\lambda_i^{-1} = \partial(\lambda_i \triangleright B_{ijk})\,. \tag{4.13}$$

Thus we deduce that under the gauge transformation $A_{ij} \to \lambda_i A_{ij}\lambda_j^{-1}$, $B$ transforms as $B_{ijk} \to B'_{ijk} = (\lambda_j \triangleright B_{ijk})\rho_{ijk}$, where $\rho_{ijk} \in \ker(\partial) = \Pi_2$. However, this $\rho$ does not play a significant role and can be omitted here.

Let us check that the 2-truncation condition $\delta_A B = 1_H$ is well-defined (covariant) under the gauge transformation with parameter $\lambda$. Note that both $A$ and $B$ will undergo the gauge

transformation, not only $B$:

$$\delta_{A'}(B')_{ijkl} = (A'_{ij} \triangleright B'_{jkl})B'^{-1}_{ikl}B'^{-1}_{ikl}B'_{ijk}.  \tag{4.14}$$

It simplifies to

$$\delta_{A'}(B')_{ijkl} = (\lambda_i)(\triangleright(A_{ij} \triangleright B_{jkl})B^{-1}_{ikl}B^{-1}_{ikl}B_{ijk}) = \lambda_i \triangleright (\delta_A B) = 1_H,  \tag{4.15}$$

and the 2-truncation condition is indeed gauge invariant.

In summary, under the 0-form gauge transformation, we have

$$A_{ij} \to \lambda_i A_{ij} \lambda_j^{-1}  \tag{4.16}$$

$$B_{ijk} \to \lambda_i \triangleright B_{ijk}  \tag{4.17}$$

for a $G$-valued function $\lambda_i$ defined on $U_i$.

### 4.2.2   1-form gauge transformation

Before starting the discussions, we emphasize that it is necessary to assume $\mathrm{im}(\partial)$ is a central subgroup of $G$ in order to have a well-defined

$$\delta A = \partial B.  \tag{4.18}$$

We apply $\delta$ on both sides of (4.18), and get

$$1_G = \partial(\delta B).  \tag{4.19}$$

Meanwhile we have $\delta_A B = 1_H$, so

$$\partial(\delta_A B) = 1_G  \tag{4.20}$$

trivially. We compare (4.19) and (4.20) and deduce that

$$\partial(A_{ij} \triangleright B_{ijk}) = \partial B_{ijk}.  \tag{4.21}$$

Again it is equivalent to $A_{ij}\partial(B_{ijk}) = \partial(B_{ijk})A_{ij}$, which holds under the assumption that $\mathrm{im}(\partial)$ is central in $G$, for arbitary $A_{ij}$ and $B_{ijk}$. This justifies the use of Bockstein homomorphisms.

For the 1-form gauge transformation, we observe that $\delta_A B = 1_H$ is preserved by $B_{ijk} \to B_{ijk}(\delta_A\Lambda)_{ijk}$, since $\delta_A^2 = 1$ ($\Lambda_{ij} \in H$ here). We perform such a transformation in $\partial B_{ijk} = (\delta A)_{ijk}$. Note $\partial \circ \delta_A = \delta \circ \partial$, hence the full 1-form gauge transformation is given by

$$\begin{aligned} A_{ij} &\to A_{ij}\partial\Lambda_{ij} \\ B_{ijk} &\to B_{ijk}(\delta_A\Lambda)_{ijk} \end{aligned}  \tag{4.22}$$

for $\Lambda \in H$.

**Remark 1.** *Consider $A = s(a)$ as in the dictionary, let us choose another section $s'$. Note that $s'(a) = s(a)\partial\Lambda$ where $\Lambda \in H$. Hence $A' = s'(a) = s(a)\partial\Lambda = A\partial\Lambda$, and changing the choice of section amounts to a 1-form gauge transformation.*

## 4.3 Dictionary on observables

Now we discuss the definition of observables in terms of the strict gauge fields $(A, B)$. For simplicity we work on a spacetime lattice, i.e. picking up a good cover of the spacetime manifold and taking its nerve[2]. On this lattice, the 1-form gauge field $A$ is defined on links. Given a link $e$ we will write $A_e$ for the gauge field on it. Similarly, given a 2-simplex $\sigma$, there is a 2-form gauge field defined on it $\sigma$.

Given a loop $\gamma$ on the lattice and a representation $\rho$ of group $G$, we can define the Wilson loop operator

$$W_\gamma := \mathrm{Tr}(\prod_{e \in \gamma} \rho(A_e)) \,. \tag{4.23}$$

Now let us check gauge invariance of (4.23). Under the 0-form gauge transformation

$$A_{ij} \to \lambda_i A_{ij} \lambda_j^{-1} \,. \tag{4.24}$$

It is easy to see the Wilson loop is invariant. On the other hand, to make the Wilson line (4.23) invariant under the 1-form gauge transformation

$$A_{ij} \to A_{ij}\partial(\Lambda_{ij}) \,, \tag{4.25}$$

we need to impose that $\mathrm{im}\partial \subset \ker(\rho)$. In this case, $\rho$ induces a well-defined map $\tilde{\rho}$ on $G/\mathrm{im}\partial = \Pi_1$. Hence the gauge invariant Wilson loops of the theory is labelled by representations of $\Pi_1$.

Now we consider surface operators. Given a closed surface $\Sigma$ and a representation $\eta$ of $H$, we define the surface operator by

$$W_\Sigma := \mathrm{Tr}(\prod_{\sigma \in \Sigma} \eta(B_\sigma)) \,. \tag{4.26}$$

To ensure the Wilson surface operator is invariant under the 0-form gauge transformation

$$B_{ijk} \to \lambda_i \triangleright B_{ijk} \,, \tag{4.27}$$

we impose that $\eta$ is a $G$-invariant representation of $H$, that is

$$\eta(g \triangleright h) = \eta(h) \tag{4.28}$$

---

[2]A good cover means that for $\{U_i\}$ a cover of M, each $U_i \cong \mathbb{R}^n$ and each intersection $\cap_i U_i \cong \mathbb{R}^n$. Also, we do always have such a choice of good cover.

for any $g \in G$, $h \in H$. With such constraint, Wilson surface is manifestly invariant under the 0-form gauge transformation.

For the 1-form gauge transformation

$$B_{ijk} \to B_{ijk}(\delta_A \Lambda)_{ijk} \,, \tag{4.29}$$

we note that $\eta(\delta_A \Lambda) = \delta \eta(\Lambda)$ since $\eta$ is $G$-invariant. Hence $\Lambda$ will contribute a factor

$$\prod_{\sigma \in \Sigma} (\delta \eta(\Lambda))_{\sigma} = 1 \tag{4.30}$$

by "Stokes" theorem. In conclusion, gauge invariant Wilson surface operators correspond to $G$-invariant representations of $H$.

One can also construct gauge invariant operator using the fake curvature $\mathfrak{F} = dA - \partial B$ (in the additive form). We take a surface $\Sigma$ with boundary $\partial \Sigma$, and define the gauge invariant operator

$$\begin{aligned}
W_{\Sigma,\rho} &= \exp\left(\int_\Sigma \rho(\mathfrak{F})\right) \\
&= \exp\left(\int_{\partial\Sigma} \rho(A)\right) \exp\left(\int_\Sigma \rho(\partial(B))\right) \,.
\end{aligned} \tag{4.31}$$

$\rho$ is an arbitary representation of $G$.

In the multiplicative notations, we can write an equivalent form

$$W_{\Sigma,\rho} = \mathrm{Tr}(\prod_{e \in \partial\Sigma} \rho(A_e)) \mathrm{Tr}(\prod_{\sigma \in \Sigma} \rho(\partial(B_\sigma))^{-1}) \,. \tag{4.32}$$

## 4.4 Lie 2-gauge theory

Now we briefly discuss the case of Lie 2-groups, and we will derive the famous Green-Schwarz shift [9] algebraically.

As before, we have the exact sequence

$$1 \to \Pi_2 \to H \xrightarrow{\partial} G \xrightarrow{p} \Pi_1 \to 1 \tag{4.33}$$

and a section map $s : \Pi_1 \to G$ st. $p \circ s = \mathrm{id}$.. Here $\Pi_{1,2}$ are both Lie groups.

Now we start from a $\Pi_1$-bundle and derive Čech cohomology. We denote $a$ an $\mathrm{Lie}(\Pi_1)$-valued 1-form (1-form gauge field), where Lie denotes the functor mapping Lie groups to their corresponding Lie algebras. As a principal $\Pi_1$-bundle, we should obviously have

$$g_{ij}g_{jk}g_{ki} = 1_{\Pi_1} \,. \tag{4.34}$$

Taking section s on both sides ($\widetilde{g}_{ij} \equiv s(g_{ij})$),

$$\widetilde{g}_{ij}\widetilde{g}_{jk}\widetilde{g}_{ki} = s(1_{\Pi_1}) \equiv \Omega_{ijk} . \tag{4.35}$$

Since $p \circ s = $ id., we clearly have $\Omega_{ijk} \in \ker(p) = \operatorname{im}(\partial)$. Thus we can assign an $\omega_{ijk}$ st. $\Omega_{ijk} = \partial(\omega_{ijk})$.

Consider the $\mathfrak{g}$-valued 1-form $A$ induced by $\Pi_1$ connection $a$ through $A = \underline{s}(a)$ where the underline denotes the differential of a map. The induced gauge transformation of $A$ is therefore

$$A_i - A_j = \widetilde{g}_{ij}^{-1}\mathrm{d}\widetilde{g}_{ij} \tag{4.36}$$

To arrive at the relation for $\omega_{ijk}$, we permute the labels of (4.36) $(ij, jk, ki)$ and sum them up, to get

$$0 = \underline{\partial}(\omega_{ijk}^{-1}\mathrm{d}\omega_{ijk}) \tag{4.37}$$

Thus the $\mathfrak{h}$-valued 1-form $\omega_{ijk}^{-1}\mathrm{d}\omega_{ijk} \in \ker(\underline{\partial}) = \operatorname{Lie}(\Pi_2)$.

Then let's consider the derivation of Čech 3-cocycle $\beta$. This would require an intersection of four $U_i$ patches, namely we can calculate $\widetilde{g}_{ij}\widetilde{g}_{jk}\widetilde{g}_{kl}$ in two different orders of composition and the result should agree in $G$:

$$\Omega_{ijl}\Omega_{jkl} = \Omega_{ijk}\Omega_{jkl} . \tag{4.38}$$

So the corresponding $\omega$s should have the following relation,

$$\omega_{ijl}\omega_{jkl} = \omega_{ijk}\omega_{jkl}\beta_{ijkl} \tag{4.39}$$

where $\beta_{ijkl} \in \ker(\partial) = \Pi_2$ carries the information of Postnikov class. We can check that by taking $\partial$ on both sides of (4.39) we clearly have (4.38).

To view $\beta_{ijkl}$ as an element in group cohomology $H^3_{grp}(\Pi_1; \Pi_2)$, we simply take $\beta_{ijkl}$ : $\Pi_1 \times \Pi_1 \times \Pi_1 \to \Pi_2, (g_{ij}, g_{jk}, g_{kl}) \mapsto \beta_{ijkl}$, and the cocycle condition can be checked by pentagon identity. To view $\beta$ as Čech cohomology $\check{H}^3(\Pi_1; \underline{\Pi_2})$, we naturally take the indices as intersection of elements of $\{U_i\}$ and $\beta$ as the map from 3 Čech simplexes into $\Pi_1$. Note that in the crossed module $p^*\beta = 0$, we can describe the bundle with strict 2-group $(G, H, \partial, \rhd)$ in the way of [54].

To consider the gauge transformation, we make the following correspondence, $a$ is the $\operatorname{Lie}(\Pi_1)$-valued 1-form gauge field, $B$ as a $\operatorname{Lie}(\Pi_2)$-valued 2-form gauge field and $\beta$ the 3-form gauge field on $B\Pi_1$. Consider the classifying map $f : M \to B\Pi_1$, we directly derive from Čech cohomology that

$$\mathrm{d}B = f^*\beta \tag{4.40}$$

and the gauge transformation rules can be derived from that.

An example would be the Chern-Simons form for $\Pi_1 = \Pi_2 = U(1)$. For $\beta \in H^3(BU(1); U(1)) = \mathbb{Z}$, its pullback corresponds to the $\mathbb{Z}$-graded Chern-Simons form, namely,

$$\mathrm{d}B = \frac{\kappa}{2\pi} a \wedge \mathrm{d}a \tag{4.41}$$

with corresponding gauge transformation when $a \mapsto a + \mathrm{d}\lambda$,

$$B' = B + \frac{\kappa}{2\pi} \lambda \wedge \mathrm{d}a \,. \tag{4.42}$$

## 4.5   Strictification of 3-gauge fields

In this subsection, we present a similar construction for 3-group gauge fields. We focus on two special cases which are discussed in section 3, which are 3-groups with either $\Pi_1 = 0$ or $\Pi_2 = 0$.

**The case of $\Pi_1 = 0$.**   This case is completely in analogue with the 2-group case with degree shift by one. More precisely, let us fix the exact sequences

$$1 \to \Pi_3 \to L \to \mathrm{im}(\partial_2) \to 1 \tag{4.43}$$

and

$$1 \to \mathrm{im}(\partial_2) \to H \to \Pi_2 \to 1 \,. \tag{4.44}$$

Given $b \in H^2(M, \Pi_2)$ for spacetime manifold $M$ which satisfies $\delta b = 1_{\Pi_2}$ where $\delta$ is the Čech differential. Then one chooses a section $s : \Pi_2 \to H$ which may not be a homomorphism. So $\delta s(b) \neq 1_H$ in general. Nonetheless, we can easily verify that $\delta s(b) \in \mathrm{im}(\partial_2)$, so one can further lift

$$\delta(s(b)) = \partial_2(l) \tag{4.45}$$

for some $l \in L$. Note that $\delta^2 = 0$, hence we obtain

$$\partial_2(\delta l) = 1_H \,. \tag{4.46}$$

It means that $\delta l = b^* \gamma \in H^4(M, \Pi_3)$ where $\gamma \in H^4(B^2\Pi_2; \Pi_3)$ is the Postnikov cocycle.

**The case of $\Pi_2 = 0$.**   In the case of $\Pi_2 = 0$, let us fix the exact sequences

$$\begin{aligned} 1 &\to \Pi_3 \to L \to \mathrm{im}(\partial_2) \to 1 \\ 1 &\to \ker(\partial_1) \to H \to G \to \Pi_1 \to 1 \,. \end{aligned} \tag{4.47}$$

Consider the second exact sequence in (4.47), one starts with $a \in H^1(M, \Pi_1)$ in this case which satisfies $\delta a = 1_{\Pi_1}$ as usual. We fix a set theoretic section $s : \Pi_1 \to G$ and we have

$$\delta(s(a)) = \partial_1(b) \tag{4.48}$$

as before. Taking Čech differential of (4.48), we obtain

$$\partial_1(\delta b) = 1 \tag{4.49}$$

which means that there exists $l \in L$ such that $\partial_2(l) = \delta(b)$. We take yet another Čech differential, we obtain $\partial_2(\delta l) = 1$. So it defines a Cech cohomology class $\gamma = \delta(l) \in H^4(M; \Pi_3)$, which is the pullback of Postnikov cocycle in $H^4(B\Pi_1; \Pi_3)$ by the classifying map $a : M \to B\Pi_1$.

## 5 Higher-representations

### 5.1 Automorphism 2-representation

The mathematical theory of higher representations of higher groups is not fully established. In this paper, for the physical application purpose, we would construct 2-representations of strict and weak 2-groups using the automorphism 2-group of algebras.

**Definition 5.1.** *Given an algebra A, the* automorphism 2-group *[70] of the algebra* $\mathcal{A}ut(A)$ *is defined to be*

$$H = A^\times = \{\text{invertible elements in } A\}$$
$$G = \text{Aut}(A) \tag{5.1}$$
$$\text{Ad.} : A^\times \to \text{Aut}(A), \ a \mapsto \text{Adj}_a \ .$$

This definition gives a crossed extension sequence,

$$1 \to Z(A^\times) \to A^\times \to \text{Aut}(A) \to \text{Out}(A) \to 1 \ . \tag{5.2}$$

Therefore we define a 2-representation of a 2-group on $A$ as follows.

**Definition 5.2.** *Given an algebra A, $\mathcal{A}ut(A)$ is the automorphism 2-group, we define a representation of a strict 2-group $\mathcal{G}$ on A to be a strict intertwiner [71]*

$$\mathcal{R} : \ \mathcal{G} \to \mathcal{A}ut(A) \ , \tag{5.3}$$

*see figure 1.*

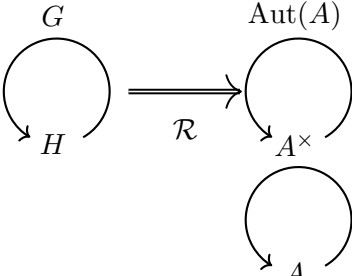

Figure 1: This figure shows how we construct an automorphism 2-representation. We choose a suitable algebra $A$, and its automorphism 2-group $\mathcal{A}ut(A)$ can be calculated by definition. Then we build an intertwiner $\mathcal{R}$ to embed the 2-group structure into $\mathcal{A}ut(A)$.

This could be considered as a "homomorphism" (with respect to the group laws, $\partial$ map and action) from a strict 2-group to an automorphism 2-group of an algebra. Note that this is not a weak equivalence, since we do not require the $\Pi_1$ and $\Pi_2$ of the two crossed modules to be the same.

Algebraically, a 2-representation breaks down to building a commutative diagram preserving the strict 2-group action $\triangleright$, namely,

$$
\begin{array}{ccc}
H & \xrightarrow{\ \partial\ } & G \\
{\scriptstyle t_H}\downarrow & & \downarrow{\scriptstyle t_G} \\
A^\times & \xrightarrow{\ \text{Ad.}\ } & \mathrm{Aut}(A)
\end{array}
\qquad\qquad
t_G(g)\big(t_H(h)\big) = t_H(g \triangleright h)
\tag{5.4}
$$

thus we can denote a 2-representation by $(t_H, t_G, A)$. We define the automorphism 2-representation to be faithful if both $t_H$ and $t_G$ are injective group homomorphisms.

During the process of constructing automorphism 2-representations, we may sometimes take considerations of the subgroup of our choices, since the entirety could be massive and redundant. For example, when we take $A = \mathbb{C}[K]$, we sometimes only consider

$$
\mathrm{im}(t_G) \subseteq \mathrm{Aut}(K) \subseteq \mathrm{Aut}(A) \ .
\tag{5.5}
$$

We will write out the maps explicitly if anything may cause confusion.

The physical meaning of the algebra $A$ here is typically the fusion algebra of line operators.

For example, in the case of pure $U(1)$ 1-form symmetry of Maxwell theory, these line operators are Wilson lines of electromagnetic fields. They are described by the algebra $A = \mathbb{C}[\widehat{U(1)}]$ where $\mathbb{C}[K]$ stands for the group algebra of discrete group $K$ and $\widehat{K} := \mathrm{Hom}(K, U(1))$ is the Pontryagin dual. More concretely, let's denote the Wilson line with fundamental charge as $x$, then

$$
A = \mathbb{C}[x, x^{-1}]
\tag{5.6}
$$

Note that only finite sums are allowed. In this situation, $\mathrm{Aut}(A) = \mathbb{C}^\times \rtimes \mathbb{Z}_2$. Given $k \in \mathbb{C}^\times$, the automorphism acts as $x \to kx$. On the other hand, for $1 \neq \sigma \in \mathbb{Z}_2$, we have $\sigma(x) = x^{-1}$, which is actually charge conjugation.

Thus, the automorphism 2-representation of $BU(1)$ on $A$ is labelled by $\mathrm{Hom}(U(1), \mathbb{C}^\times) = \mathbb{Z}$, which is the charge of Wilson lines, as expected.

More generally, for Wilson lines in $K$-gauge theory, we have $A = \mathbb{C}[\widehat{K}]$.

In the 2-representation, both $\mathrm{Aut}(A)$ and $A_0^\times$ shall admit an action on the algebra $A$, the former by natural action, and the latter by left multiplication. However, if we choose the algebra $A$ to be a group algebra $A = \mathbb{C}[K]$ as we do in most cases, there can be another type of action given by Pontryagin dual. We denote the duality map by

$$\widehat{\ } : K \to \widehat{K} = \mathrm{Hom}_{\mathrm{Grp}}(K, U(1)) \ , \ \ k \mapsto \hat{k} \ . \tag{5.7}$$

In this notation, the two types of actions of the 2-group $\mathcal{G}$ on the algebra $A$ can also be constructed as

$$t_G(g) \rhd \sum_{k \in K} f(k) \ k = \sum_{k \in K} f(k) \ t_G(g)(k) \ , \tag{5.8}$$

$$t_H(h) \cdot \sum_{k \in K} f(k) \ k = \sum_{k \in K} f(k) \cdot \hat{k}(t_H(h)) \ k \ . \tag{5.9}$$

In the following, we will call the first type of action in (5.8), in which $H$ acts by left multiplication, the **natural $H$-action**. The second type of action in (5.9), in which $H$ acts by a phase determined by Pontryagin dual of each element, is called the **Wilsonian $H$-action**, as the 2-matter operator would correspond to a linear combination of a collection of Wilson loops with coefficients.

### 5.1.1 Example: $\Pi_1 = \Pi_2 = \mathbb{Z}_2$

Here we give a concrete example of faithful automorphism 2-representation for the case of $\Pi_1 = \Pi_2 = \mathbb{Z}_2$. We present both cases of Postnikov classes $\beta = 0, 1 \in H^3_{\mathrm{grp}}(\mathbb{Z}_2; \mathbb{Z}_2) \cong \mathbb{Z}_2$. In the construction below, we use the scheme of taking the algebra $A$ to be group algebra $\mathbb{C}[K]$, and roughly takes $\mathrm{Aut}(K) \subseteq \mathrm{Aut}(\mathbb{C}[K])$. It does not cause any problem, since we can always embed $\mathrm{Aut}(K)$ back into $\mathrm{Aut}(\mathbb{C}[K])$.

**Trivial $\beta$:** With the above formulation, one simple example we can show is for

$$1 \to \mathbb{Z}_2 \xrightarrow{\ i\ } \mathbb{Z}_4 \xrightarrow{\ \partial\ } \mathbb{Z}_4 \xrightarrow{\ p\ } \mathbb{Z}_2 \to 1 \tag{5.10}$$

and the strict action $\rhd$ being trivial, we choose the algebra of representation to be $\mathbb{C}[D_4]$. The

notation for group elements of $D_4$ is $D_4 = \langle a, x | a^4 = e, (a^n x)^2 = e \; \forall n \rangle$. Thus we have

$$
\begin{array}{ccc}
\mathbb{Z}_4 & \xrightarrow{\;\partial = \times 2\;} & \mathbb{Z}_4 \\
{\scriptstyle t_H} \downarrow & & \downarrow {\scriptstyle t_G} \\
D_4 & \xrightarrow[\text{Ad.}]{} & D_4
\end{array}
\tag{5.11}
$$

where $\text{Aut}(D_4) = D_4$; $t_H(1) = a$, from this one can see that $\text{Ad}_{a^2} = \text{id.}$ is exactly what we would expect for the diagram to commute; $t_G(1) = \kappa$ is the representative of outer automorphism, acting as $\kappa(x) = ax, \kappa(a) = a$, and acts on $\text{Im}(t_H)$ trivially.

**Non-Trivial $\beta$:** Suppose now we are trying to represent the $\Pi_1 = \Pi_2 = \mathbb{Z}_2$ case with a non-trivial Postnikov class non-trivial. The closest automorphism 2-group we can naively imagine would be

$$
(\mathbb{Z}_4, \mathbb{Z}_2 = \text{Aut}(\mathbb{Z}_4), \triangleright, \partial)
$$
$$
1 \triangleright 1 = 3 \; , \; 1 \triangleright 3 = 1 \; , \; \partial = \times 2 \; .
\tag{5.12}
$$

But this 2-representation is not faithful. The construction we attempt to build now is to properly expand the algebra (and in this simple case, a group $K$) to accommodate the $G$. Here is how we can do it in the example of 2.3 with non-trivial action.

We can slightly decorate to the automorphism 2-group that fixes the deficiency of order-4-elements in the automorphism group. The proposal is

$$
\begin{array}{ccc}
\mathbb{Z}_4 & \xrightarrow{\qquad\partial = \times 2\qquad} & \mathbb{Z}_4 \\
{\scriptstyle t_H} \downarrow & & \downarrow {\scriptstyle t_G} \\
K = \mathbb{Z}_4 \times S_4 & \xrightarrow[\text{Ad.}]{} & \mathbb{Z}_2 \times S_4 \subset \text{Aut}(K) \; ,
\end{array}
\tag{5.13}
$$

where

$$
t_H(1) = (1, (13)(24)) \; , \; t_G(1) = ((1 \leftrightarrow 3), \text{Ad}_{(1234)})
\tag{5.14}
$$

In this way, we successfully made $t_G$ injective by decorating some group with order-4 automorphism element.

One may ask if we can choose an abelian $K$. The answer is no, since we hope for both $t_H$ and $t_G$ to be faithful, and $\text{im}\partial \neq \{e_G\}$, $t_G \circ \text{im}\partial \neq \{\text{id.}_{\text{Aut}(K)}\}$. But an abelian $K$ will most certainly have $\text{Ad.}(K) = \{\text{id.}_{\text{Aut}(K)}\}$.

## 5.2  Relation to 2-representations of weak 2-groups

Here we review the 2-representations for weak 2-groups, see e.g. [45], and observe how it could be related with automorphic 2-representations for strict 2-groups.

For a weak 2-group $(\Pi_1, \Pi_2, \rho, \beta)$, a 2-representation is labeled by $(n, \sigma, \chi_i, c_i)$, $(i = 1, \ldots, n)$. The object for the 2-representation is $n$ line operators $L_i(\gamma)$, $i = 1, \ldots, n$. $\sigma :$ $\Pi_1 \to S_n$ is a permutation of the line operators.

An element $g \in \Pi_1$ acts on $L_i(\gamma)$ as

$$g \cdot L_i(\gamma) = L_{\sigma_g(i)}(\gamma), \tag{5.15}$$

where $\sigma_g(i) \in \{1, \ldots, n\}$.

$\chi_i : \Pi_2 \to U(1)^n$ is a $n$-dimensional unitary representation of $\Pi_2$, which obeys

$$\chi_i(a)\chi_i(b) = \chi_i(ab). \tag{5.16}$$

It is also called the character of the 2-representation.

An element $a \in \Pi_2$ acts on $L_i(\gamma)$ as

$$a \cdot L_i(\gamma) = \chi_i(a)L_i(\gamma). \tag{5.17}$$

The functions $\chi$ also satisfy the constraint

$$\chi_i(a) = \chi_{\sigma_g(i)}(\rho_g(a)). \tag{5.18}$$

Finally, we have $c_i : \Pi_1 \times \Pi_1 \to U(1)^n$ $(i = 1, \ldots, n)$, which describes the additional phase factor when two $\Pi_1$ symmetry generators act on the same line operator, see Figure 14 of [45]. They are required to satisfy

$$(\delta_\sigma c)_i(g, h, k) = \chi_i(\beta(g, h, k)). \tag{5.19}$$

Hence for a split weak 2-group with $\beta = 0$, $c$ is an element of $H^2_\sigma(B\Pi_1, U(1)^n)$.

Two 2-representations $(n, \sigma, \chi, c)$ and $(n, \sigma', \chi', c')$ are equivalent if there exists a permutation $\tau \in S_n$ such that

$$\sigma' = \tau \circ \sigma \circ \tau^{-1}, \quad [c'] = [\tau \cdot c], \quad \chi' = \tau \cdot \chi. \tag{5.20}$$

How does the weak 2-representation relate to the automorphism 2-representation for strict 2-groups? One way is to construct the following commutative diagram.

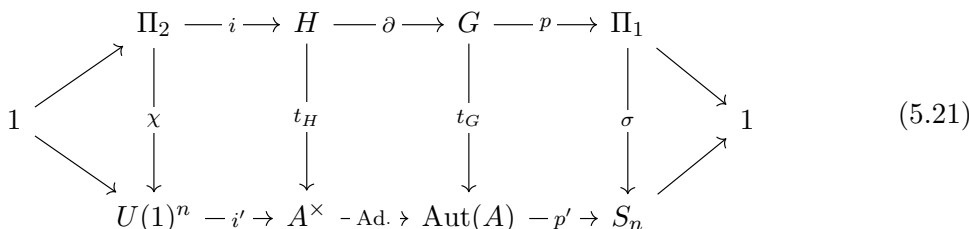

$$\tag{5.21}$$

By carefully choosing the algebra $A$, and respectively the homomorphisms $t_{G,H}$, we shall ensure that $(t_H, t_G, A)$ forms an automorphism 2-representation, and the lower row forms a crossed module.

Notably, the role of $c$ is played by $F$ in the procedure of obtaining the Postnikov class from strict 2-group in section 2.1. More precisely, we should have

$$i'(c(g_1, g_2)) = t_H(F(g_1, g_2)) \,, \tag{5.22}$$

which in terms re-generate (2.9) mapped into the lower crossed module in (5.21).

### 5.2.1 Example: $\Pi_1 = \Pi_2 = \mathbb{Z}_2$

Let us shed some concreteness into the construction relating strict and weak 2-representations through the simplest discrete example. Unless specified, we will still adopt the abbreviation by taking group algebra and only consider the automorphism of groups. In this example, we do not consider one particular outer automorphism of $U(1)$,

$$\alpha_i : U(1) \to U(1) \,, \ e^{i\theta} \mapsto e^{-i\theta} \,. \tag{5.23}$$

**Trivial $\beta$:** In the case where the crossed module is

$$1 \to \mathbb{Z}_2 \xrightarrow{i} \mathbb{Z}_4 \xrightarrow{\partial} \mathbb{Z}_4 \xrightarrow{p} \mathbb{Z}_2 \to 1 \tag{5.24}$$

with the action $\rhd$ trivial, one can have myriad choices of weak 2-representations. In the follows, we will discuss some of these weak 2-representations $(n, \sigma, \chi_i, c_i)$ and relate them with automorphism 2-representations.

1. When $n = 1$, $\sigma$ has to be trivial, and we can choose $\chi_1$ to be either $\chi_1(a) = 1$ or $\chi_1(a) = (-1)^a$. $c_1 \in C^2(\mathbb{Z}_2, U(1))$ is a choice independent of $\chi_1$.

   In the strict description, we choose $A^\times = U(1)$, $t_G(1) = \mathrm{id}._A$, hence the actual $\mathrm{Aut}(A)$ does not matter. There are 2 choices for $t_H$, $t_H(1) = \pm 1$ if $\chi_1(a) = 1$; $t_H(1) = e^{\pm \frac{\pi i}{2}}$ if $\chi_1(a) = (-1)^a$. Any of the above choices yields trivial Postnikov class and gives trivial $c$ up to coboundary.

2. When $n = 2$, there are two choices for $\sigma$:

   (a) $\sigma$ is trivial, and one may choose $\chi_1$, $\chi_2$, $c_1$, $c_2$ independently. The representation is a direct sum $(1, 1, \chi_1, c_1) \oplus (1, 1, \chi_2, c_2)$.

   In the strict description, one correspondingly choose $A^\times = U(1)^2$, $t_H = (\pm 1, \pm 1)$ being the direct sum of previous results, and $t_G(1) = \mathrm{id}$.

(b) $\sigma(0) = \text{id}$, $\sigma(1) = (12)$. In this case, because of the condition (5.18), we have to impose

$$\chi_1(a) = \chi_2(a) \quad \forall a. \tag{5.25}$$

Hence we have either $\chi_1(a) = \chi_2(a) = 1$ or $\chi_1(a) = \chi_2(a) = (-1)^a$.

The choices of $c_1$, $c_2$ are independent of $\chi_1, \chi_2$, they need to satisfy

$$c_{\sigma_g(i)}(h,k)c_i(g,hk)c_i^{-1}(g,h)c_i^{-1}(gh,k) = 1 \quad (i = 1,2). \tag{5.26}$$

In this case we can still choose $A^\times = U(1)^2$, and take $S_2 \subset \text{Aut}(A)$ to be where $\text{im}(t_G)$ resides. Then we can still have 2 choices for $t_H$, $t_H(1) = (\pm 1, \pm 1)$ if $\chi_1(a) = 1$; $t_H(1) = (e^{\pm \frac{\pi i}{2}}, e^{\pm \frac{\pi i}{2}})$ if $\chi_1(a) = (-1)^a$. For $t_G$, we take $t_G(1) = (12)$. One can check that this choice will make the diagram commutative while preserving the action.

**Non-trivial $\beta$:** Now suppose the crossed module is same as before,

$$1 \to \mathbb{Z}_2 \xrightarrow{i} \mathbb{Z}_4 \xrightarrow{\partial} \mathbb{Z}_4 \xrightarrow{p} \mathbb{Z}_2 \to 1 \tag{5.27}$$

but the action $\triangleright$ is non-trivial, thus yielding the Postnikov class non-trivial. The situation becomes more interesting.

1. When $n = 1$, $\sigma$ has to be trivial, and we can choose $\chi_1$ to be either $\chi_1(a) = 1$ or $\chi_1(a) = (-1)^a$.

   (a) If $\chi_1(a) = 1$, $c_1 \in C^2(\mathbb{Z}_2, U(1))$. The strict automorphic representation is same as before, since we do not allow any information of action (and Postnikov class) to appear.

   (b) If $\chi_1(a) = (-1)^a$, $c_1$ satisfies

   $$c_1(h,k)c_1(g,hk)c_1^{-1}(g,h)c_1^{-1}(gh,k) = \chi_1(\beta(g,h,k)). \tag{5.28}$$

   However, the above equation has no solution, hence when this weak 2-group has a non-trivial $\beta$, there is no one-dimensional non-trivial representation.

2. When $n = 2$, if $\sigma$ is trivial, the 2-representation is a direct sum of two 1-dimensional 2-representations. And we do not repeat the details here.

   However, if $\sigma(1) = (12)$, from (5.18) we have to impose

   $$\chi_1(a) = \chi_2(a) \quad \forall a. \tag{5.29}$$

   There are two choices for $\chi_i$:

(a) If $\chi_1(a) = \chi_2(a) = 1$, the choices of $\chi_i$ satisfy the same equation of (5.26), hence the strict automorphic 2-representation has the same structure.

(b) If $\chi_1(a) = \chi_2(a) = (-1)^a$, the equations (5.19) for $c_i$ become

$$c_1(0,0) = c_1(0,1) = c_2(1,0),$$
$$c_2(0,0) = c_2(0,1) = c_1(1,0), \tag{5.30}$$
$$c_2(1,1)c_1(1,0)c_1^{-1}(1,1)c_1^{-1}(0,1) = -1.$$

There are eight choices of $(c_1, c_2)$. Using a construction resembling the non-trivial Postnikov class case of section 5.1.1, we give a commutative diagram

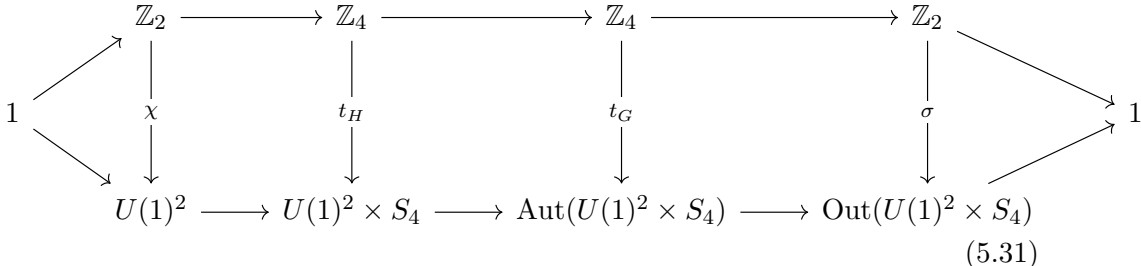

$$\tag{5.31}$$

where we take

$$t_H(1) = \left( \begin{pmatrix} e^{\pm \frac{\pi i}{2}} & 0 \\ 0 & e^{\pm \frac{\pi i}{2}} \end{pmatrix}, (1234) \right) , \ t_G(1) = ((12),(1234)) \in S_2 \times S_4 \subset \mathrm{Aut}(U(1)^2 \times S_4)$$

$$i_A : U(1)^2 \to U(1)^2 \times S_4 , \ \begin{pmatrix} -1 & 0 \\ 0 & -1 \end{pmatrix} \mapsto \left( \begin{pmatrix} -1 & 0 \\ 0 & -1 \end{pmatrix}, (13)(24) \right)$$

$$\sigma(1) = (12) \in \mathrm{Out}(U(1)^2 \times S_4) \ , \ \ \sigma(1) \cdot \begin{pmatrix} a & 0 \\ 0 & b \end{pmatrix} = \begin{pmatrix} b & 0 \\ 0 & a \end{pmatrix} , \ \forall a,b \in U(1) .$$

$$\tag{5.32}$$

One can verify that this diagram is indeed commutative and is consistent with the group action.

With this construction, we can uncover the information of $[c]$ in 2-representation by choosing $\mathrm{im}(t_H)$, where each diagonal element $e^{\pm \frac{\pi i}{2}}$ can take the $\pm$ independently. If $t_H(1) \propto \mathrm{diag}(1,1)$, then the swapping generated by $t_G(1)$ and $\sigma(1)$ acts trivially; if $t_H(1) \propto \mathrm{diag}(1,-1)$, $t_G(1)$ acts non-trivially on $\mathrm{im}(t_H)$, rendering the automorphism 2-representation faithful, while $\sigma$ acting on $\mathrm{im}(\chi)$ is still trivial by definition. It's the two classes of choices gives different $[c]$ information: the $t_H(1) \propto \mathrm{diag}(1,1)$ case cannot generate $c$ satisfying Eq.(5.30) from $F$; while the $t_H(1) \propto \mathrm{diag}(1,-1)$ can generate a non-trivial Postnikov class in the 2-representation of our choice, in other words, it faithfully represent the Postnikov class information.

## 5.3 3-representations

In this section, we discuss the 3-representations of 3-group $(G, H, L, \partial_1, \partial_2, \triangleright, \{,\})$ in the special case of $\Pi_1 = G/\mathrm{im}(\partial_1) = 0$.

When $\Pi_1 = 0$, we can choose $G$ to be trivial in the semi-strict 3-group. In this case, the 3-group is essentially characterized by a 2-group structure $(H, L, \partial_2, \triangleright')$ as described by (3.5). Therefore, we can still use the automorphism representation for 2-groups to construct the representation of 3-groups. We will now see for the simplest case how this works.

**Example:** $\Pi_2 = \Pi_3 = \mathbb{Z}_4$. In this example, we consider the case when $\Pi_1 = 0$ and $\Pi_2 = \Pi_3 = \mathbb{Z}_4$. With the choice of $\partial_2 = \times 2$, to determine the 2-group $(\Pi_2, \Pi_3, \partial_2, \triangleright')$, we long to see what choices of Peiffer lifting could give trivial or non-trivial actions $\triangleright'$. The results of calculation is presented in Table 1. We can construct both the case of trivial action and the case of non-trivial action with suitable choices of the quadratic form $F$, which correspond to elements of $H^4(B^2\mathbb{Z}_4; \mathbb{Z}_4)$. With the 2-group structure determined, we can extradite this case back to section 5.1.1 where the 2-representation of this particular 2-group is explicitly constructed.

| $F$ | $\triangleright'$ | $F$ | $\triangleright'$ |
|---|---|---|---|
| $\begin{pmatrix} 0 & 0 & 0 & 0 \\ 0 & 0 & 0 & 0 \\ 0 & 0 & 0 & 0 \\ 0 & 0 & 0 & 0 \end{pmatrix}$ | trivial | $\begin{pmatrix} 0 & 0 & 0 & 0 \\ 0 & 2 & 0 & 2 \\ 0 & 0 & 0 & 0 \\ 0 & 2 & 0 & 2 \end{pmatrix}$ | trivial |
| $F$ | $\triangleright'$ | $F$ | $\triangleright'$ |
| $\begin{pmatrix} 0 & 0 & 0 & 0 \\ 0 & 1 & 2 & 3 \\ 0 & 2 & 0 & 2 \\ 0 & 3 & 2 & 1 \end{pmatrix}$ | non-trivial | $\begin{pmatrix} 0 & 0 & 0 & 0 \\ 0 & 3 & 2 & 1 \\ 0 & 2 & 0 & 2 \\ 0 & 1 & 2 & 3 \end{pmatrix}$ | non-trivial |

Table 1: This table shows the choice of quadratic forms $F : \mathbb{Z}_4 \times \mathbb{Z}_4 \to \mathbb{Z}_4$ and the resulting action $\triangleright' : \mathbb{Z}_4 \to \mathrm{Aut}(\mathbb{Z}_4) \cong \mathbb{Z}_2$ in the sub 2-group structure. In this table, $F = (F)_{4\times4}$ means that $F(i, j) = F_{i+1,j+1}$ for $i, j \in \mathbb{Z}_4$.

# 6 Higher gauge theory in path space

## 6.1 Path space formalism

In this section, we formulate the 2-group gauge theory in the path space following [54]. We denote by $M$ the base manifold, $(G, H, \partial, \rhd)$ the Lie 2-group, and the Lie algebras take basis

$$
\begin{aligned}
\mathfrak{g} &= \mathrm{Span}(T^a, \ a = 1, \cdots, \dim(\mathfrak{g})) \\
\mathfrak{h} &= \mathrm{Span}(S^a, \ a = 1, \cdots, \dim(\mathfrak{h}))
\end{aligned}
\tag{6.1}
$$

and we follow the mathematicians' notation of gauge group and holonomy as in [54] (i.e. a Wilson line is $\exp\left(\int_\gamma A\right)$ instead of $\exp\left(i \int_\gamma A\right)$). For two points $s, t \in M$, $\mathcal{P}_s^t(M)$ denotes the **path space**,

$$
\mathcal{P}_s^t(M) = \{X : [0,1] \to M, \ X(0) = s, \ X(1) = t\} .
\tag{6.2}
$$

We also define the evaluation map

$$
e_\sigma : \mathcal{P}_s^t \to M, \ X \mapsto X(\sigma) ,
\tag{6.3}
$$

which would induce a natural pullback,

$$
e_\sigma^* : \Omega^p(M) \to \Omega^p(\mathcal{P}_s^t(M)), \ \omega \mapsto e_\sigma^*(\omega) \equiv \omega(\sigma) .
\tag{6.4}
$$

The specific components are

$$
\omega(\sigma) \equiv e_\sigma^*(\omega) = \omega_{\mu_1 \cdots \mu_p} \frac{\partial X^{(\mu_1, \sigma)}}{\partial X^{(\nu_1, \rho_1)}} \cdot \frac{\partial X^{(\mu_2, \sigma)}}{\partial X^{(\nu_2, \rho_2)}} \cdots \frac{\partial X^{(\mu_p, \sigma)}}{\partial X^{(\nu_p, \rho_p)}} dX^{(\nu_1, \rho_1)} \wedge \cdots X^{(\nu_p, \rho_p)} .
\tag{6.5}
$$

Given that

$$
\frac{\partial X^{(\mu_i, \sigma)}}{\partial X^{(\nu_i, \rho_i)}} = \delta_{\nu_i}^{\mu_i} \delta(\sigma - \rho_i) ,
\tag{6.6}
$$

the components are explicitly

$$
\omega(\sigma) = \omega_{\mu_1 \cdots \mu_p}(X(\sigma)) \, dX^{(\mu_1, \sigma)} \wedge \cdots \wedge dX^{(\mu_p, \sigma)} .
\tag{6.7}
$$

As such, the exterior differential $\mathbf{d}$ on the path space $\mathcal{P}_s^t(M)$ is written as

$$
\mathbf{d} = dX^{(\mu, \sigma)} \wedge \frac{\delta}{\delta X^{(\mu, \sigma)}} .
\tag{6.8}
$$

For a given line $X$, there is a vector field generating reparameterizations,

$$
K(X) = \frac{dX}{d\sigma} .
\tag{6.9}
$$

Given a family of forms $\{\omega_i\}$ on $M$, we introduce an integral

$$
\begin{aligned}
\Omega_{\{\omega_i\},(\alpha,\beta)} &\equiv \oint_{X|_\alpha^\beta} (\omega_1, \cdots, \omega_n) \\
&\equiv \int_{\alpha \leq \sigma_i \leq \sigma_{i+1} \leq \beta} \iota_K \omega_1(\sigma_1) \wedge \cdots \wedge \iota_K \omega_n(\sigma_n)
\end{aligned}
\tag{6.10}
$$

If each $\deg(\omega_i) = p_i + 1$, then $\deg(\Omega_{\{\omega_i\}}) = \sum_i p_i$.

Now we introduce the line holonomy,

$$
\begin{aligned}
W_A[X](\sigma_1, \sigma_2) &\equiv \mathcal{P} \exp\left( \int_{X|_{\sigma_1}^{\sigma_2}} A \right) \\
&\equiv \sum_{n=0}^{\infty} \oint_{X|_{\sigma_1}^{\sigma_2}} (A^{a_1}, \cdots, A^{a_n}) \cdot T^{a_1} \cdots T^{a_n}
\end{aligned}
\tag{6.11}
$$

The parallel transport of $\mathfrak{g}$-valued element is

$$
\begin{aligned}
T^A(\sigma) &\equiv T^{W_A[X]}(\sigma) = W_A^{-1}[X](\sigma, 1) T(\sigma) W_A[X](\sigma, 1) \\
S^A(\sigma) &\equiv S^{W_A[X]}(\sigma) = W_A^{-1}[X](\sigma, 1) \triangleright S(\sigma) .
\end{aligned}
\tag{6.12}
$$

Now we abbreviate the integral for $\mathfrak{g}$-valued forms on $M$,

$$
\begin{aligned}
\oint_A (\omega_1, \cdots, \omega_n) &\equiv \oint (\omega_1^{W_A}, \cdots, \omega_n^{W_A}) \\
&= \int_{\alpha \leq \sigma_i \leq \sigma_{i+1} \leq \beta} \iota_K(W_A^{-1}[X](\sigma_1, 1) \omega_1(\sigma_1) W_A[X](\sigma_1, 1)) \wedge \cdots \\
&\qquad \wedge \iota_K(W_A^{-1}[X](\sigma_n, 1) \omega_n(\sigma_n) W_A[X](\sigma_n, 1))
\end{aligned}
\tag{6.13}
$$

and similarly for the $\mathfrak{h}$-valued forms, we substitute the adjoint with 2-group action.

More precisely, for $\oint_A(a)$ for an $\mathfrak{h}$-valued 1-form $a$, the explicit expression is

$$
\begin{aligned}
a \in \mathfrak{h} \otimes \Omega^1(M) &\mapsto a(\sigma) \equiv e_\sigma^*(a) \in \mathfrak{h} \otimes \Omega^1(\mathcal{P}_s^t(M)) \\
&\mapsto W_A^{-1}[X](\sigma, 1) \triangleright a(\sigma) \in \mathfrak{h} \otimes \Omega^1(\mathcal{P}_s^t(M)) \\
&\mapsto \iota_K(W_A^{-1}[X](\sigma, 1) \triangleright a(\sigma)) \in \mathfrak{h} \\
&\mapsto \oint_A (a) \equiv \int_0^1 d\sigma \, \iota_K(W_A^{-1}[X](\sigma, 1) \triangleright a(\sigma)) \in \mathfrak{h} .
\end{aligned}
\tag{6.14}
$$

Notably, the integral $\oint_A$ helps us to see how a correspondence

$$
\Omega^p(M) \leftrightarrow \Omega^{p-1}(\mathcal{P}(M))
\tag{6.15}
$$

takes place, the important thing here is that a 1-form connection $\mathcal{A}$ in path space shall correspond to a 2-form field $B$ in the real space, and a first-order derivative $\mathbf{d}$ of the path

space correspond to an area derivative in the real space. Notably, there is the following formula for $\omega \in \Omega^k(M)$,

$$\mathbf{d} \oint_A (\omega) = - \oint_A (\mathrm{d}_A \omega) - (-1)^{\deg(\omega)} \oint_A (T^a \triangleright \omega, F_A^a) \ . \tag{6.16}$$

Given $A$ a $\mathfrak{g}$-valued 1-form and $B$ an $\mathfrak{h}$-valued 2-form on $M$, we define the path space 1-form

$$\mathcal{A}_{(A,B)} \equiv \oint_A (B) = \int_0^1 \mathrm{d}\sigma \ \iota_K (W_A^{-1}[X](\sigma,1) \triangleright B(\sigma)) \ \in \Omega^1(\mathcal{P}_s^t(M)) \otimes \mathfrak{h} \tag{6.17}$$

and the path space holonomy is thus ($\Sigma$ in $\mathcal{P}_s^t(M)$ is a curve from $X_0$ to $X_1$, which sweep through a surface in $M$)

$$\mathcal{W}_\mathcal{A}(\Sigma) \equiv \mathcal{P} \exp\left(\int_\Sigma \mathcal{A}\right) \ , \tag{6.18}$$

which admits 2 kinds of gauge transformations.

The first kind of gauge transformation is for a given $\phi \in \Omega^0(M, G)$, when the path space is a loop space $\mathcal{P}_x^x(M)$,

$$\mathcal{W}_{\mathcal{A}_{(A',B')}}(\Sigma) = \phi(x) \triangleright \mathcal{W}_{\mathcal{A}_{(A,B)}}(\Sigma)$$
$$A' = \phi A \phi^{-1} + \phi \mathrm{d}\phi^{-1} \tag{6.19}$$
$$B' = \phi \triangleright B \ .$$

The second kind of gauge transformation is for a given $a \in \Omega^1(M, \mathfrak{h})$, (infinitesimal version)

$$\mathcal{W}_{\mathcal{A}_{(A',B')}}(\Sigma|_{X_0}^{X_1}) = (1 - \epsilon \oint_A (a))_{X_0} \mathcal{W}_{\mathcal{A}_{(A,B)}}(\Sigma|_{X_0}^{X_1})(1 + \epsilon \oint_A (a))_{X_1}$$
$$A' = A + \epsilon \ \partial(a) \tag{6.20}$$
$$B' = B - \epsilon \ \mathrm{d}_A a$$

the finite version of the second kind is denoted $a_\gamma$ in the above.

Now we see precisely what should $\Phi$ the matter field be. We would like to have a term looking like

$$(\mathbf{d} + \mathcal{A}) \cdot \Phi \ , \tag{6.21}$$

so $\Phi \in \Omega^0(\mathcal{P}_s^t(M), V)$, where $\mathcal{A}$ admits a representation on $V$. Also, for the 0-form action $g$ to be well-defined, we should assert that $\Phi \in \Omega^0(\mathcal{P}_x^x(M), V)$.

## 6.2 Path space derivative

In this subsection we provide more detailed explanations of the exterior derivative in the path space.

In the above, the basis for differential are denoted as $\mathrm{d}X^{(\mu,\sigma)}$, which could be depicted as infinitesimally dragging the line segment at $\sigma$ along the $\mu$ direction, pictorially shown as Figure 2.

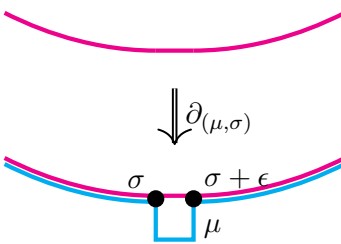

Figure 2: Here we demonstrate the picture of path space derivative. The above shows the original line in magenta, the below shows both the original line in magenta and the deformed line in cyan. The line in cyan is formed by a deviation from $\sigma$ to $\sigma + \epsilon$ along the $\mu$ direction.

Formally speaking, we consider two vectors $t^\mu$ and $n^\mu$, which are tangent vector of the line and the normal vector that generates the shift of line segment at $\sigma$. The area of the zone swept through by the line segment is therefore

$$\delta A^{\mu\nu} = n^\mu t^\nu - n^\nu t^\mu . \tag{6.22}$$

We can re-write this in the path space language, suppose $\delta X^\mu(\sigma)$ denotes such a deviation and plays the role of normal vector, and $K = \mathrm{d}X/\mathrm{d}\sigma$ plays the role of tangent vector, thus

$$\delta A^{\mu\nu} = \int_0^1 \mathrm{d}\sigma \; \delta X^{[\mu} K^{\nu]} \Big|_\sigma . \tag{6.23}$$

When we take $\delta X^\mu(\sigma)$ to be localized in a certain point $\gamma(\sigma_0)$, namely

$$\delta X^\mu(\sigma) = \delta v^\mu \delta(\sigma - \sigma_0) \tag{6.24}$$

one can decompose $v^\mu = v_\mathrm{n}^\mu + v_\mathrm{t}^\mu$ into components normal and tangent to the chosen path. By doing this, one finds out that only normal component contributes to the area, while tangent components generate reparameterization. To sum up, the localized path space derivative goes back to the area derivative described in [61].

## 6.3 Pure 1-form gauge theory

In this subsection, we observe that this formulation derives the mean string field theory formulation in [61], when $\Pi_1 = G = 0$.

Consider the case of gauge theory with pure 1-form symmetry described by a Lie group $H = \Pi_2$. The covariant derivative written in explicit formula is

$$\mathbf{d} + \mathcal{A} = \mathbf{d} + \int_0^1 \mathrm{d}\sigma \; \iota_K(B) . \tag{6.25}$$

The exact meaning of $\mathbf{d}$ in the path space can be perceived from using the (6.16) . In the absence of group $G$, (6.16) reads

$$\mathbf{d} \int_0^1 \mathrm{d}\sigma \, \iota_K(\omega) = - \int_0^1 \mathrm{d}\sigma \, \iota_K(\mathrm{d}\omega) . \tag{6.26}$$

Consider the action on the matter field,

$$(1 - \epsilon \oint_A (a)) \mapsto \exp\left(-\int_0^1 \mathrm{d}\sigma \, \iota_K(a)\right) = \exp\left(-\oint(a)\right),$$
$$\mathbf{d}\exp\left(-\oint(a)\right) = \exp\left(\oint(a)\right) \cdot \oint(\mathrm{d}a) . \tag{6.27}$$

While its action on the 2-form gauge field $B$ gives

$$\mathcal{A} = \oint(B) \mapsto \mathcal{A}' = \oint(B - \mathrm{d}a) . \tag{6.28}$$

Since the $G$-action is trivial, we omit the $A$ in $\oint_A$. Thus, a matter field $\Phi$ in the path space transforming as

$$\Phi \mapsto \exp\left(-\oint(a)\right) \cdot \Phi \tag{6.29}$$

generates the covariant term

$$\begin{aligned}
(\mathbf{d} + \mathcal{A}) \cdot \Phi \mapsto &(\mathbf{d} + \mathcal{A}') \cdot \exp\left(-\oint(a)\right)\Phi \\
=&(\mathbf{d}\exp\left(-\oint(a)\right))\Phi + \exp\left(-\oint(a)\right)\mathbf{d}\Phi + \oint(B + \mathrm{d}a)\exp\left(-\oint(a)\right)\Phi \\
=&\exp\left(-\oint(a)\right) \cdot [\mathbf{d}\Phi + \oint(B + \mathrm{d}a - \mathrm{d}a)\Phi] \\
=&\exp\left(-\oint(a)\right)(\mathbf{d} + \mathcal{A}) \cdot \Phi .
\end{aligned} \tag{6.30}$$

The resulting covariant derivative on $\Phi$ exactly matches the one in [61] (with a difference of conventions). Along this line, we can choose a unitary representation for $H$, and construct the kinetic terms and mass terms which lead to a Landau-Ginzburg Lagrangian for the 1-form symmetry $H$.

## 6.4 Discrete gauge fields

For the discrete case, we take a triangulation of the spacetime manifold $M$, with each vertex labelled by $i$. A path on $M$ is defined to be a ordered set of vertex labels

$$\gamma = [i_1, i_2, \ldots, i_n] \tag{6.31}$$

with each link $(i_k, i_{k+1})$ being a 1-simplex in the triangulation. A 1-form field $A \in \Omega^1(M, \mathfrak{g})$ is translated as an assignment of each link with a $G$ element, with

$$A_{ij} = A_{ji}^{-1} . \tag{6.32}$$

Likewise, a 2-form field $B \in \Omega^2(M, \mathfrak{h})$ is translated as an assignment of each 2-simplex with an $H$ element.

As we previously described, the path space derivative is generated by deviation of a path on a certain point, which should be translated as

$$\Delta_{k,j} : \gamma = [i_1, \ldots, i_n] \mapsto \gamma' = [i_1, \ldots, i_k, j, i_{k+1}, \ldots, i_n] \tag{6.33}$$

where $(i_k, j, i_{k+1})$ forms a 2-simplex [3]. Thus, a path space 1-form should be defined as

$$\mathcal{A}_{\gamma, \Delta_{k,j}(\gamma)} \in H , \quad \mathcal{A}_{\gamma_1, \gamma_2} = \mathcal{A}_{\gamma_2, \gamma_1}^{-1} . \tag{6.34}$$

Following the continuous version (6.17), we could define the discrete version of path space 1-form as

$$\mathcal{A}_{\gamma, \Delta_{k,j}(\gamma)} = (W_A[i_{k+1}, \ldots, i_n])^{-1} \triangleright B_{i_{k+1}, j, i_k} . \tag{6.35}$$

For multiple such line segments placed together, the 2-holonomy should be an ordered product

$$\mathcal{W}_{\mathcal{A}} = \prod_{k_i \leq k_{i+1}} \mathcal{A}_{\gamma, \Delta_{k_i, j_i}(\gamma)} . \tag{6.36}$$

Under the 0-form gauge transformation, the 1-form connection $\mathcal{A}$ of path space transforms as

$$(W_A[i_{k+1}, \ldots, i_n])^{-1} \mapsto g_{i_n} \cdot (W_A[i_{k+1}, \ldots, i_n])^{-1} \cdot g_{i_{k+1}}^{-1} , \quad B_{i_{k+1}, j, k} \mapsto g_{i_{k+1}} \triangleright B_{i_{k+1}, j, k}$$
$$\Rightarrow \mathcal{A}_{\gamma, \Delta_{k,j}(\gamma)} \mapsto g_{i_n} \triangleright \mathcal{A}_{\gamma, \Delta_{k,j}(\gamma)} , \tag{6.37}$$

For 1-form transformation, we only have to do a direct translation. But if we take the assumption we had before in section 4.2.2, $\mathcal{A}$ shall become

$$\mathcal{A}_{\gamma, \Delta_{k,j}(\gamma)} \mapsto \mathrm{Ad}_H(\Lambda_{i_{n-1} i_n}^{-1}) \circ A_{i_{n-1} i_n}^{-1} \triangleright \ldots \mathrm{Ad}_H(\Lambda_{i_{k+1} i_{k+2}}^{-1}) \circ A_{i_{k+1} i_{k+2}}^{-1} \triangleright (B_{i_{k+1} j i_k}(\delta_A \Lambda)_{i_{k+1} j i_k}) \tag{6.38}$$

for computational convenience, if we take the assumption that $H$ is abelian [4], we get

$$\mathcal{A}_{\gamma, \Delta_{k,j}(\gamma)} \mapsto W_A^{-1}[i_{k+1}, \ldots, i_n] \triangleright B_{i_{k+1} j i_k} \cdot W_A^{-1}[i_{k+1}, \ldots, i_n] \triangleright (\delta_A \Lambda)_{i_{k+1} j i_k}$$
$$= \mathcal{A}_{\gamma, \Delta_{k,j}(\gamma)} \cdot W_A^{-1}[i_{k+1}, \ldots, i_n] \triangleright (\delta_A \Lambda)_{i_{k+1} j i_k} \tag{6.39}$$

---

[3] To make things consistent, we induce an identification of $[i_1, \ldots, i_{k-1}, i_k, j, i_k, i_{k+1} \ldots, i_n]$ $\cong$ $[i_1, \ldots, i_{k-1}, i_k, i_{k+1}, \ldots, i_n]$.

[4] Note that by assuming $H$ is abelian, we do not mean that the chosen group algebra of the automorphism 2-representation should also be abelian. Suppose that the automorphism 2-representation is faithful and $H$ is abelian, we should still have $\mathrm{Im}(t_H)$ is abelian, and the formulas we introduce here will still hold.

Note that in the discrete scenario, we still require that the fake curvature should vanish, meaning that

$$\delta A = \partial(B) \tag{6.40}$$

as Čech cocycles.

We can also see at what the 1-form transformation looks like in the discrete case. A 1-form transformation is defined as ($\epsilon = 1/N$)

$$\lim_{N \to \infty} (1 - \epsilon \oint_A (a))(1 - \epsilon \oint_{A+\epsilon\partial(a)} (a))(1 - \epsilon \oint_{A+2\epsilon\partial(a)} (a)) \ldots (1 - \epsilon \oint_{A+\frac{N-1}{N}\partial(a)} (a)) \tag{6.41}$$

and in the discrete language, for $a \in \Omega^1(M) \otimes \mathfrak{h}$, $a$ gives an $H$ element on each link, then for a given line $\gamma = [i_1, \ldots, i_n]$,

$$\exp\left(\oint_A (a)\right) \mapsto \prod_{k=n-1}^{1} W_A^{-1}[i_{k+1}, \ldots, i_n] \triangleright a[i_k, i_{k+1}] . \tag{6.42}$$

Thus the consistency condition is

$$\exp\left(-\oint_A (a)\right)_{\gamma_1} \mathcal{W}_{\mathcal{A}}(\Sigma_{\gamma_1,\gamma_2}) \exp\left(\oint_A (a)\right)_{\gamma_2} = \mathcal{W}_{\mathcal{A}'}(\Sigma_{\gamma_1,\gamma_2}) , \tag{6.43}$$

which in the case of minimal deformation $\Delta_{k,j}$,

$$
\begin{aligned}
\text{LHS} =& W_A^{-1}[i_{k+1}, \ldots, i_n] \triangleright B_{i_{k+1},j,i_k} \cdot [W_A^{-1}[i_{k+1}, \ldots, i_n] \triangleright a_{i_k,i_{k+1}}]^{-1} \cdot W_A^{-1}[i_{k+1}, \ldots, i_n] \triangleright a_{j,i_{k+1}} \\
& \cdot W_A^{-1}[j, i_{k+1}, \ldots, i_n] \triangleright a_{i_k,j} \\
=& \mathcal{A}_{\gamma, \Delta_{k,j}(\gamma)} \cdot W_A^{-1}[i_{k+1}, \ldots, i_n] \triangleright (A_{i_{k+1},j} \triangleright a_{j,i_k}^{-1} a_{i_{k+1},i_k} a_{i_{k+1},j}^{-1}) \\
=& \mathcal{A}_{\gamma, \Delta_{k,j}(\gamma)} \cdot W_A^{-1}[i_{k+1}, \ldots, i_n] \triangleright (\delta_A(a^{-1})) = \text{RHS( taking } a = -\Lambda \text{ )} .
\end{aligned}
\tag{6.44}
$$

is indeed correct.

## 6.5 Discrete 2-matter

In this section, we build the non-local matter fields in the discrete case. In this section, we suppose the 2-group is represented by an automorphism 2-group,

$$1 \to Z(\Upsilon^\times) \to \Upsilon^\times \to \text{Aut}(\Upsilon) \to \text{Out}(\Upsilon) \to 1 \tag{6.45}$$

naturally, the matter field should take value in the algebra $\Upsilon$. Note that we do not require any abelian property of the $\Upsilon$ in the 2-representation we choose.

Suppose we would like the matter $\Phi$ to be a scalar field in the **path space**, then due to the $\oint_A$ construction, we should build $\Phi$ by an $\Upsilon$-valued 1-form $\phi$ in the **real spacetime**. We

could take the construction [5]

$$\Phi_\gamma = \sum_{k=1}^{n} W_A^{-1}[i_{k+1}, \ldots, i_n] \rhd \phi_{i_k, i_{k+1}} \ , \tag{6.47}$$

where under 0-form gauge transformation

$$\phi_{ij} \mapsto g_i \rhd \phi_{ij} \ , \tag{6.48}$$

$\Phi$ is covariant under this construction,

$$\Phi_\gamma \mapsto g_{i_n} \rhd \Phi_\gamma \ . \tag{6.49}$$

Now we should consider how $\mathbf{d}_\mathcal{A}\Phi(\Lambda)$ transforms under 0-form gauge transformation. Note that

$$(\mathbf{d}_\mathcal{A}\Phi)_{\gamma_1, \gamma_2} = (\mathcal{A}_{\gamma_1, \gamma_2} \cdot \Phi_{\gamma_2})\Phi_{\gamma_1}^{-1} \mapsto g_{i_n} \circ (\mathbf{d}_\mathcal{A}\Phi)_{\gamma_1, \gamma_2} \tag{6.50}$$

where $\circ$ is the adjoint action of element of $H$ on $A = \Upsilon^\times$. Now let's consider the transformation of $\mathbf{d}_\mathcal{A}\Phi$ under 1-form gauge transformation,

$$\Phi_\gamma \mapsto \exp\left(\oint_A (a)\right)_\gamma \Phi_\gamma$$

$$(\mathbf{d}_\mathcal{A}\Phi)_{\gamma_1, \gamma_2} \mapsto \left\{ \left[ \mathcal{A}_{\gamma_1, \gamma_2} \cdot (W_A^{-1}[i_{k+1}, \ldots, i_n] \rhd (\delta_A a)) \right] \circ \left[ \exp\left(\oint_A (a)\right)_{\gamma_2} \Phi_{\gamma_2} \right] \right\}$$

$$\cdot \left\{ \exp\left(\oint_A (a)\right)_{\gamma_1} \Phi_{\gamma_1} \right\}^{-1} \tag{6.51}$$

$$= \exp\left(\oint_A (a)\right)_{\gamma_1} \cdot \mathcal{A}_{\gamma_1, \gamma_2} \cdot \Phi_{\gamma_2} \cdot \Phi_{\gamma_1}^{-1} \cdot \exp\left(-\oint_A (a)\right)_{\gamma_1}$$

$$= \exp\left(\oint_A (a)\right)_{\gamma_1} \cdot (\mathbf{d}_\mathcal{A}\Phi)_{\gamma_1, \gamma_2} \cdot \exp\left(-\oint_A (a)\right)_{\gamma_1}$$

where $\exp(\oint_A(a))$ is defined in the discrete case by (6.42), and we see this is indeed covariant under both 0-form and 1-form gauge transformation.

## 6.6   Continuous 2-matter

The discrete version shed light on how we can define the continuous 2-matter.

---

[5]There can be another valid construction by substituting sum (in the algebra) with multiplication (in the algebra),

$$\Lambda_\gamma = \prod_{k=1}^{n} W_A^{-1}[i_{k+1}, \ldots, i_n] \rhd \phi_{i_k, i_{k+1}} \ , \tag{6.46}$$

but as we will see, $\Phi$ is a better choice for discussing 2-matter for higher gauge theories.

Given an $\Upsilon$(the algebra of 2-representation)-valued 1-form $\phi \in \Omega^1(M, \Upsilon)$, one can build the 2-matter in path space by

$$\Phi_\gamma \equiv \oint_A (\phi) = \int_0^1 d\sigma \; \iota_K(W_A^{-1}(\sigma, 1) \circ \phi(\sigma)) \in \Omega^0(\mathcal{P}_s^t(M), \Upsilon) . \tag{6.52}$$

Here $\circ$ denotes the $G = \mathrm{Aut}(\Upsilon)$ element acting on $\Upsilon$-element. But in general, we merely require the 2-matter to be

$$\Phi \in \Omega^0(\mathcal{P}_s^t(M), \Upsilon) . \tag{6.53}$$

The gauge transformation for $\phi$

$$\phi_\mu(x) \mapsto g(x) \circ \phi_\mu(x) , \; W_A(\sigma, 1) \mapsto g(\gamma(1))W_A g^{-1}(\gamma(\sigma)) . \tag{6.54}$$

induces the transformation

$$\Phi_\gamma \mapsto g(\gamma(1)) \circ \Phi_\gamma . \tag{6.55}$$

The 1-form gauge transformation is non-local, since it's related to a path. We write the infinitesimal transformation as

$$\Phi_\gamma \mapsto (1 + \epsilon \oint_A (a))_\gamma \circ \Phi_\gamma = (1 + \epsilon \oint_A (a))_\gamma \cdot \Phi_\gamma \tag{6.56}$$

where $\circ$ denotes the $H = \Upsilon_0^\times$ element acting on $\Upsilon$-element by adjoint action, and $\cdot$ denotes the multiplication of $\Upsilon$-elements. Similarly, we can deduce the 1-form gauge transformation

$$
\begin{aligned}
\mathbf{d}_{\mathcal{A}}\Phi =& \mathbf{d}\Phi + \mathcal{A} \circ \Phi \\
\mapsto& \mathbf{d}[(1 + \epsilon \oint_A (a)) \circ \Phi] + [\mathcal{A}'(1 + \epsilon \oint_A (a))] \circ \Phi \\
=& (1 + \epsilon \oint_A (a)) \circ (\mathbf{d}\Phi) \\
& + (\epsilon \mathbf{d}\oint_A (a)) \circ \Phi + \{[(1 + \epsilon \oint_A)(\mathbf{d} + \mathcal{A})(1 - \epsilon \oint_A (a))](1 + \epsilon \oint_A (a))\} \circ \Phi \\
=& (1 + \epsilon \oint_A (a)) \circ (\mathbf{d}_{\mathcal{A}}\Phi) + \mathcal{O}(\epsilon^2) .
\end{aligned}
\tag{6.57}
$$

Thus we have proved this covariant derivative term is indeed covariant under both 0-form and 1-form gauge transformation. Notably, for the continuous case, we do not require $H$ to be abelian.

## 6.7   Towards $n$-matter

In this part, we consider a Lie 3-group with $\Pi_1 = 0$. In this case we have showed that the structure of 3-group is essentially encoded a 2-group, see section 5.3). Hence we can also

describe the 3-representation of the 3-group by the language of 2-representation as depicted in section **??**. The corresponding spacetime gauge fields are

$$C \in \Omega^3(M, \mathfrak{l}) \ , \ \ B \in \Omega^2(M, \mathfrak{h}) \ . \tag{6.58}$$

Since we turn off the zero form part $G$ and its 1-form gauge field $A$ in the 3-group, it is within our purview to write

$$\begin{aligned}
\mathcal{C} &\equiv \oint(C) \in \Omega^2(\mathcal{P}(M), \mathfrak{l}) \, , \\
\mathcal{B} &\equiv \oint(B) \in \Omega^1(\mathcal{P}(M), \mathfrak{h}) \, .
\end{aligned} \tag{6.59}$$

We could observe an acquainted visage on the path space! We could once again define

$$\mathscr{D} \equiv \oiint_{\mathcal{B}}(\mathcal{C}) \in \Omega^1(\mathcal{P}(\mathcal{P}(M)), \mathfrak{l}) \tag{6.60}$$

where $\mathcal{P}(\mathcal{P}(M)) = \mathcal{P}^2(M)$ is the surface space of $M$. There should also be a fake-curvature condition

$$\mathbf{d}^{(1)}\mathcal{B} - \partial(\mathcal{C}) = 0 \ , \tag{6.61}$$

where $\mathbf{d}^{(k)}$ denotes the exterior derivative in the $k-$path space $\mathcal{P}^k(M)$. Thus the 3-curvature constructed is

$$\mathcal{Z} = \mathbf{d}^{(1)}_{\mathcal{B}}\mathcal{C} \in \Omega^3(\mathcal{P}(M), \mathfrak{l}) \ , \ \ \mathscr{F}_{\mathcal{B},\mathcal{C}} = \oiint_{\mathcal{B}}(\mathcal{Z}) \in \Omega^2(\mathcal{P}^2(M), \mathfrak{l}) \tag{6.62}$$

Likewise, given a 2-representation on algebra $\Upsilon$, we can similarly define a matter field $\Psi \in \Omega^0(\mathcal{P}^2(M), \Upsilon)$, which is a brane field, s.t. $\mathbf{d}^{(2)}_{\mathscr{D}}\Psi$ is covariant under both 1-form and 2-form gauge transformations.

This could be easily generalized to $n$-group with only the $n$-th and $(n-1)$-th categorical layer non-trivial. This generalizes the result in [72].

## 7  Landau-Ginzburg model of higher-group symmetries

### 7.1  2-group gauge theory with 2-matter

With the formalism constructed in the last sections, we summarize the 2-group covariant terms as follows,

$$Z = \mathbf{d}_A B \ , \ \ \mathcal{F}_\mathcal{A} = \oint_A(Z) \ , \ \mathbf{d}_\mathcal{A}\Phi \ , \ \Phi \tag{7.1}$$

and the vanishing condition for the fake curvature,

$$\partial(B) - F_A = \mathrm{d}Z + A \wedge^\triangleright Z = 0 \tag{7.2}$$

which could be inserted into the Lagrangian through Lagrange multipliers.

With this regard, suppose both $\Upsilon_0^\times$ and $\text{Aut}(\Upsilon)$ act on $\Upsilon$ unitarily, we propose an action of the following form,

$$Z = \int [\mathscr{D}A][\mathscr{D}B][\mathscr{D}\lambda][\mathscr{D}\phi]$$
$$\exp\left\{ i\int_{\mathcal{P}(M)} \left[ -\frac{1}{2g^2}|\mathcal{F}_\mathcal{A}|^2 + \frac{1}{L(C)}(\mathbf{d}_\mathcal{A}\Phi)^\dagger (\mathbf{d}_\mathcal{A}\Phi) + V(\Phi, \Phi^\dagger) \right] [dC] + i\int_M \lambda^{(d-2)} \wedge (\partial(B) - F_A) \right\}.$$

(7.3)

$[dC]$ denotes the integration measure of the path space.

For 3-group with $\Pi_1 = 0$, we can build a similar 3-group gauge theory with 3-matter utilizing the 3-gauge-covariant terms constructed in section(6.7).

### 7.1.1 Spontaneously symmetry breaking and area law

In absence of gauge fields, i.e. $A = B = 0$, above Lagrangian reduces to following form

$$S = \int_{\mathcal{P}(M)} \left[ \frac{1}{L(C)}(\mathbf{d}\Phi)^\dagger (\mathbf{d}\Phi) + V(\Phi, \Phi^\dagger) \right] [dC].$$

(7.4)

This Lagrangian has following global symmetries:

1. $\Phi \to g \circ \Phi$, where $g \in G$. This action is induced by the automorphic 2-representation on $\Upsilon$-valued 1-form $\phi$ (Recall that 2-representation contains a map $G \to \text{Aut}(\Upsilon)$).

2. $\Phi_\gamma \to e^{i\theta \oint_\gamma a} \circ \Phi_\gamma$, where $a$ is a $\mathfrak{h}$-valued 1-form. This action is induced by the 2-representation $H \to \Upsilon^\times$.

The equation of motion is

$$\star\mathbf{d} \star \left( \frac{1}{L(C)}\mathbf{d}\Phi \right) - \frac{\delta V}{\delta\Phi^\dagger} = 0,$$

(7.5)

where $L(C)$ is the length of the path (especially, it is not a constant on the path space) and $\star$ is supposed to be the Hodge start operator on path space $\mathcal{P}(M)$. In components, we have

$$\partial_{\mu,\sigma} \left( \frac{1}{L(C)}\partial^{\mu,\sigma}\Phi \right) - \frac{\delta V}{\delta\Phi^\dagger} = 0$$

(7.6)

The contraction of index $\mu$ is usual Einstein convention, however, $\sigma$ should be thought as a continuous index, hence the contraction of $\sigma$ involves an integration over $\sigma$.

Now, let us focus on loops for now, thus we can talk about the area $A(C)$ bounded by the loop $C$. Notice that $\mathbf{d}A$ should be thought as a 1-form on $\mathcal{P}(M)$. Given a tangent vector, i.e. a deformation vector $\delta X^{\mu,\sigma}$ (they are really vector fields defined along the curve $C$), we have

$$\mathbf{d}A\left( \frac{\delta}{\delta X^{\mu,\sigma}} \right) = K^\mu(\sigma).$$

(7.7)

Let us recall that $K^\mu(\sigma) = \frac{dX^\mu(\sigma)}{d\sigma}$ is the tangent vector of the curve. Below, we fix the parameter $\sigma$ to be the arc length $s$ of path $C$. Hence, we can define

$$|\mathbf{d}A|^2 := \partial_{\mu,\sigma} A \partial^{\mu,\sigma} A = \int_0^{L(C)} |K|^2 ds = L(C) \, . \tag{7.8}$$

Since $|K|^2 = 1$ if $\sigma = s$. Intuitively, above equation means, the infinitesimal variation of area is proportional to length.

Now we focus on the simplest potential $V(\Phi^\dagger \Phi) = r|\Phi|^2$. Without higher order terms, $r$ should be positive to make sure that $V$ is bounded from below. Besides, we take the ansatz $\Phi = e^{S[A]}$ where $S[A]$ is a functional of $A$ on the path space.

The equation of motion reduces to

$$S'[A]^2 + \partial_{\mu,\sigma}\left(\frac{1}{L(C)} S'[A]\partial^{\mu,\sigma} A\right) = r \tag{7.9}$$

where we have used (7.8) in the first term. If we only consider loops with large areas, for regular curves, we should expect that $L(C) \propto \sqrt{A}$, so the second term is as worst as $S'[A]/\sqrt{A}$, hence it can be omitted safely.

To sum up, we have

$$S[A] = -\sqrt{r}A + O(A^{\frac{1}{2}}) \, . \tag{7.10}$$

In the lowest order, we can see $\Phi$ decays with area-law.

Now suppose that $r < 0$, in that case we have to incorporate the $u|\Phi|^4$ term, making the equation of motion Eq.(7.6)

$$\partial_{\mu,\sigma}\left(\frac{1}{L(C)}\partial^{\mu,\sigma}\Phi\right) = r\Phi + 2u|\Phi|^2\Phi \, . \tag{7.11}$$

One can consider a static stable solution of the form

$$|\Phi| = \sqrt{\frac{-r}{2u}} \, . \tag{7.12}$$

An easy observation is that one can choose a special direction s.t. the 0-form part of the symmetry is broken. Let's take the example of $\Upsilon = \mathbb{C}(U(1)^n)$, thus $\mathrm{Aut}(\Upsilon) = S_n$, $\Upsilon^\times = U(1)^n$. Under this specific 2-representation, one can choose

$$\Phi = \sqrt{\frac{-r}{2u}}(1,0,\ldots,0) \tag{7.13}$$

Thus, the $S_n$ symmetry breaks down to $S_{n-1}$ symmetry. Yet one is not limited to make only this choice, for example,

$$\Phi = \sqrt{\frac{-r}{2nu}}(1,\ldots,1) \tag{7.14}$$

which preserves the entire $S_n$ permutation symmetry is also plausible. For the 1-form symmetry spontaneously broken phase, one can fix a local counter term on $\Phi$,

$$\Phi \mapsto \exp\left( ic \int_\gamma \mathrm{d}x \right) \Phi \tag{7.15}$$

s.t. the $\langle \Phi \rangle = const$ phase is equivalent to the $\langle \Phi \rangle = \exp(-aL[C])$ perimeter law phase.

We can also formulate the above discussion in the discrete scenario. For simplicity, let us assume that the spacetime is discretized into square lattices with uniform edge length 1, and we work in the loop space of the system. Thus a loop is defined as[6]

$$C = [i_1, \ldots, i_{L[C]-1}, i_{L[C]}] \tag{7.16}$$

s.t. $i_{L(C)} = i_1$ and each $i_k i_{k+1}$ is a link in the plaquettes. In this sense, a minimal deformation of a loop at site $i_k \in C = [i_1, \ldots, i_{L[C]-1}, i_{L(C)}]$ towards the direction $\mu$ becomes

$$\Delta_{i_k, \mu} C = [i_1, \ldots, i_k, j_k, j_{k+1}, i_{k+1}, \ldots, i_{L[C]-1}, i_{L[C]}] = C' \tag{7.17}$$

s.t. $i_k, j_k, j_{k+1}, i_{k+1}$ forms a plaquette spanning the $\mu$ direction and the $i_k i_{k+1}$ direction. Note that the sequence described in (7.17) may be redundant, and can be reduced up to thin homotopy equivalence.

One can classify the loop deformation into 3 types, (we always assume that $i_k i_{k+1}$ is orthogonal to the $\mu$ direction), see figure 3 for visual demonstration:

- Suppose $j_{k+1} \neq i_{k+2}$, $i_{k-1} \neq j_k$ and $j_{k+1} \neq i_{k+2}$ we have

$$\Delta_{i_k, \mu} L[C] = +2 \ , \ \ \Delta_{i_k, \mu} A[C] = \pm 1 \ . \tag{7.18}$$

- Suppose $j_{k+1} = i_{k+2}$, then the sequence $[\ldots i_k j_k j_{k+1} i_{k+1} i_{k+2} \ldots] = [\ldots i_k j_k i_{k+2} \ldots]$, rendering

$$\Delta_{i_k, \mu} L[C] = 0 \ , \ \ \Delta_{i_k, \mu} A[C] = \pm 1 \ . \tag{7.19}$$

- Suppose $i_{k-1} = j_k$ and $j_{k+1} = i_{k+2}$, then $[\ldots i_{k-1} i_k j_k j_{k+1} i_{k+1} i_{k+2} \ldots] = [\ldots i_{k-1} i_{k+2} \ldots]$, resulting in

$$\Delta_{i_k, \mu} L[C] = -2 \ , \ \ \Delta_{i_k, \mu} A[C] = \pm 1 \ . \tag{7.20}$$

We could recover (7.8) up to a normalization factor,

$$\left| \mathbf{d}A[C] \right|^2 = \sum_{i_k \in C \ , \ \mu} g^\mu{}_\mu \left| \Delta_{i_k, \mu} A[C] \right|^2 = (D-2)L[C] \ . \tag{7.21}$$

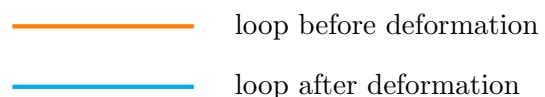

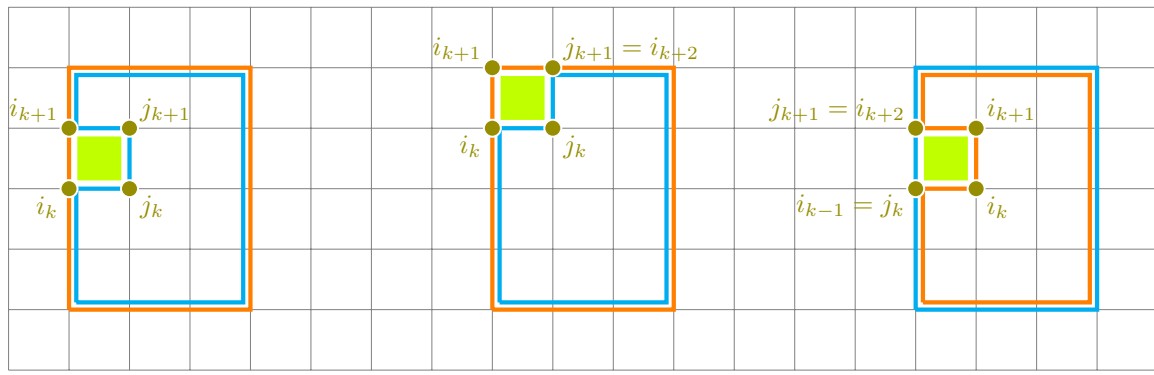

Figure 3: From left to right, this figure demonstrates three different cases of loop deformation described above. The orange lines are the loops before deformation and cyan lines are loops after deformation. The lime-coloured region signifies the change of area.

To write down the equation of motion, we need to explicitly define how the differential operators acts on the functions. We define

$$\Delta_{i_k,\mu} f[C] \equiv f[\Delta_{i_k,\mu} C] - f[C] \ . \tag{7.22}$$

For example,

$$\Delta_{i_k,\mu}\left(\frac{1}{L[C]}\right) = \begin{cases} -\dfrac{2}{L[C](L[C]+2)}, & \Delta L[C] = 2 \\[2mm] 0, & \Delta L[C] = 0 \\[2mm] \dfrac{2}{L[C](L[C]-2)}, & \Delta L[C] = -2 \end{cases} \tag{7.23}$$

Suppose $V(\Phi) = r|\Phi|^2$ $(r > 0)$, the equation of motion is

$$(D-2)S'[A]^2 + \Delta_{i_k,\mu}\Big(\frac{1}{L[C]}S'[A]\Delta^{i_k,\mu}A[C]\Big) = r \ , \tag{7.24}$$

where the $(D-2)$ comes from the normalization in (7.21). Taking that the loop is large and roughly rectangular, we can assume $L[C] \propto \sqrt{A}$ and omit the second term as we previously analyzed. Thus

$$S[A[C]] \propto \sqrt{\frac{r}{D-2}}A + \mathcal{O}(A^{\frac{1}{2}}) \ . \tag{7.25}$$

The result unabated if we do a discrete analysis.

---

[6]A loop is defined up to thin homotopy equivalence when we consider the mapping into the algebra. When we count length, we choose the configuration with minimal length.

### 7.1.2 A concrete example: $G = H = \mathbb{Z}_4$

In this section we make precise of the simplest non-trivial example of 2-group gauge theory with the framework we just developed.

Suppose we have a 2-group $\mathcal{G} = (\mathbb{Z}_2, \mathbb{Z}_2, \alpha = \text{id}, \beta \neq 0)$, this weak 2-group admits a strictification $(\mathbb{Z}_4, \mathbb{Z}_4, \triangleright, \partial)$, where $\partial = \times 2$, $(1) \triangleright (k) = 4 - k \mod 4$ as established in section 2.3. The next thing to do is automorphic 2-representation, which is discussed in section 5.1.1. In this simple case, we merely use the 2-representation for completeness of the procedure, since after 2-representation, we choose a group representation s.t. only the

$$\text{im}(t_H) \cong \mathbb{Z}_4 \ , \ \text{im}(t_G) \cong \mathbb{Z}_4 \tag{7.26}$$

parts are left non-trivial.

With the chosen representation, the 2-group globally symmetric action takes the form of (7.4), with space discretized and $\partial_{(\mu,\sigma)}$ substituted by $\Delta_{k,j}$ defined as (6.33),

$$S = \int_{\mathcal{P}(M)} \left[ \frac{1}{L(C)} (\mathbf{d}\Phi)^\dagger (\mathbf{d}\Phi) + V(\Phi) \right] [dC] . \tag{7.27}$$

In this example, there are two ways of $H = \mathbb{Z}_4$ action on the vector $\Phi$ as stated in section 5.1. The natural action is given by

$$1 \circ_n \begin{pmatrix} a \\ b \\ c \\ d \end{pmatrix} = \begin{pmatrix} d \\ a \\ b \\ c \end{pmatrix} , \tag{7.28}$$

and the Wilsonian action is given by

$$1 \circ_W \begin{pmatrix} a \\ b \\ c \\ d \end{pmatrix} = \begin{pmatrix} a \\ ib \\ -c \\ -id \end{pmatrix} . \tag{7.29}$$

With discrete 2-global symmetry $(\mathbb{Z}_4, \mathbb{Z}_4, \times 2, \triangleright)$, the potential term $V(\Phi)$ to the lowest orders shall include

$$|\Phi|^2 \ , \ |\Phi|^4 \ . \tag{7.30}$$

Only for the natural action can we admit $\Phi^4$ term into the potential.

Now suppose the spacetime dimension $D > 2$, we study the 2-matter field configuration

$$\Phi[C] = e^{S[A[C]]} \Phi_0 = e^{S[A[C]]} \begin{pmatrix} a \\ b \\ c \\ d \end{pmatrix} = e^{S[A[C]]} (a\mathbf{0} + b\mathbf{1} + c\mathbf{2} + d\mathbf{3}) \tag{7.31}$$

where we have chosen a canonical representation for group $H$, and the group algebra structure should be

$$\Phi_1 \cdot \Phi_2 = \begin{pmatrix} a_1 \\ b_1 \\ c_1 \\ d_1 \end{pmatrix} \cdot \begin{pmatrix} a_2 \\ b_2 \\ c_2 \\ d_2 \end{pmatrix} = \begin{pmatrix} a_1 a_2 + b_1 d_2 + d_1 b_2 + c_1 c_2 \\ b_1 a_2 + a_1 b_2 + c_1 d_2 + d_1 c_2 \\ c_1 a_2 + c_2 a_1 + b_1 b_2 + d_1 d_2 \\ d_1 a_2 + d_2 a_1 + b_1 c_2 + b_2 c_1 \end{pmatrix} , \tag{7.32}$$

one can verify that this multiplication is associative. Later we will use

$$\begin{pmatrix} a \\ b \\ c \\ d \end{pmatrix}^3 = \begin{pmatrix} a^3 + 3(ac^2 + b^2c + cd^2) + 6abd \\ d^3 + 3(a^2b + bc^2 + b^2d) + 6acd \\ c^3 + 3(a^2c + ab^2 + ad^2) + 6bcd \\ b^3 + 3(a^2d + bd^2 + c^2d) + 6abc \end{pmatrix} \tag{7.33}$$

Since we assume that $a, b, c, d$ do not depend on the path space, the variation only concerns the $e^{S[A[C]]}$ factor. In the following, we always assume $\Phi_0 \neq 0$.

The result of $V(\Phi) = r|\Phi|^2$ $(r > 0)$ is given previously by (7.25). In the following, we will discuss the case of $r < 0$ for this 2-group, and analyze the configurations of spontaneous symmetry breaking.

A potential admitting broken symmetry can take the form of[7]

$$V_1(\Phi) = r|\Phi|^2 + u|\Phi|^4 \ (r < 0 \ , \ u > 0) \ , \tag{7.34}$$

for both choices of $H$-action. The analysis is identical with the previous analysis. A static solution with

$$|\Phi| = \sqrt{\frac{-r}{2u}} \tag{7.35}$$

would satisfy the equation of motion.

Also one can consider the potential (when taking the natural $H$-action)

$$V_2(\Phi) = r|\Phi|^2 + u|\Phi|^4 + s \operatorname{Tr}_\Upsilon(\Phi\Phi^*\Phi\Phi^*) \ (r < 0 \ , \ u, s > 0) \ , \tag{7.36}$$

where

$$\operatorname{Tr}_\Upsilon \begin{pmatrix} a \\ b \\ c \\ d \end{pmatrix} = \operatorname{Tr}_\Upsilon(\Phi) = a + b + c + d \tag{7.37}$$

$$\operatorname{Tr}_\Upsilon(\Phi\Phi^*\Phi\Phi^*) = (a + b + c + d)^2 (\bar{a} + \bar{b} + \bar{c} + \bar{d})^2 . \tag{7.38}$$

The static solution shall satisfy

$$r\Phi + 2u|\Phi|^2\Phi + 2s \operatorname{Tr}_\Upsilon(\Phi)^2 \operatorname{Tr}_\Upsilon(\Phi^*) \begin{pmatrix} 1 \\ 1 \\ 1 \\ 1 \\ 1 \end{pmatrix} = 0 . \tag{7.39}$$

---

[7]Note that here we take $|\Phi|^2 = |a|^2 + |b|^2 + |c|^2 + |d|^2$

When considering the Wilsonian $H$-action, we can add another $|\Phi^2|$ term into $V_1(\Phi)^8$, resulting in the potential

$$V_3(\Phi) = r|\Phi|^2 + w|\Phi^2| + u|\Phi|^4 \tag{7.40}$$

and the equation of motion

$$r\Phi + 2u|\Phi|^2\Phi + w|\Phi^2|' = 0 . \tag{7.41}$$

In the following we will analyze the configurations and the symmetries they preserve and break with either the natural $H$ action or the Wilsonian $H$ action.

**With Natural $H$ Action and $V_1(\Phi)$**

Note that as stated before, the solution could both preserve and break the $G$-action symmetry. Namely, the $\Phi$ configurations of the form

$$\Phi = \sqrt{\frac{-r}{2u}} \begin{pmatrix} a \\ b \\ c \\ d \end{pmatrix} , \quad |a|^2 + |b|^2 + |c|^2 + |d|^2 = 1 , \quad b = d \tag{7.42}$$

preserves the $G$ symmetry, while configurations with $b \neq d$ breaks the $G$ symmetry.

For $H$ symmetry acting on $\Phi$ by multiplying an element in the group, the only configuration preserving the complete $H = \mathbb{Z}_4$ symmetry is

$$\Phi = \sqrt{\frac{-r}{8u}} e^{i\theta} \begin{pmatrix} 1 \\ 1 \\ 1 \\ 1 \end{pmatrix} . \tag{7.43}$$

This configuration preserves the entire 2-group symmetry, both strict and weak.

The configuration preserving $\mathbb{Z}_2 \subset H$ is

$$\Phi = \sqrt{\frac{-r}{2u}} \begin{pmatrix} a \\ b \\ a \\ b \end{pmatrix} , \quad 2|a|^2 + 2|b|^2 = 1 , \quad a, b \neq 0 . \tag{7.44}$$

This configuration preserves the sub-2-group symmetry $(\mathbb{Z}_2, \mathbb{Z}_2, \times 2, \mathrm{id})$, which has $\Pi_1 = \Pi_2 = \mathbb{Z}_2$, a trivial Postnikov class and trivial action in the weak 2-group language.

---

[8]It is also plausible to add $|\Phi^4|$ term, but doing so will significantly increase the complexity of the situation without providing new insights.

**With Natural $H$-Action and $V_2(\Phi)$**

There can be several non-trivial solutions, one type of which is

$$\Phi = c \cdot \begin{pmatrix} 1 \\ 1 \\ 1 \\ 1 \end{pmatrix} , \tag{7.45}$$

making (7.39)

$$(r + 8u|c|^2 + 128s|c|^2)c = 0 , \tag{7.46}$$

solved by

$$\Phi = \sqrt{\frac{-r}{8u + 128s}}e^{i\theta} \begin{pmatrix} 1 \\ 1 \\ 1 \\ 1 \end{pmatrix} , \quad \Phi = 0 , \quad \left(\forall \theta \in [0, 2\pi)\right). \tag{7.47}$$

These solutions preserve the entire 2-group symmetry.

Another type of solution is

$$\Phi_1 = b \begin{pmatrix} -2 \\ 1 \\ 0 \\ 1 \end{pmatrix} , \quad \Phi_2 = b \begin{pmatrix} 0 \\ 1 \\ -2 \\ 1 \end{pmatrix} , \tag{7.48}$$

with

$$|b| = \sqrt{-\frac{r}{12u}} . \tag{7.49}$$

From the strict 2-group perspective, these two solutions preserve the $G$-action symmetry and breaks the $H$-symmetry completely, the preserved sub-2-group is $(\mathbb{Z}_2, 0, 0, \text{id})$. From the weak 2-group perspective, the only remaining symmetry is the $\mathbb{Z}_2$ 0-from symmetry, rendering $(\mathbb{Z}_2, 0, \text{id}, 0)$.

The third type of solution we introduce here is

$$\Phi = \begin{pmatrix} a \\ b \\ a \\ b \end{pmatrix} , \quad a \neq b . \tag{7.50}$$

Notice that now (7.39) reads

$$\left(r + 4u(|a|^2 + |b|^2)\right) \begin{pmatrix} a \\ b \\ a \\ b \end{pmatrix} = -16s(a + b)^2(\bar{a} + \bar{b}) \begin{pmatrix} 1 \\ 1 \\ 1 \\ 1 \end{pmatrix} , \tag{7.51}$$

which is never going to hold for $a \neq b$, unless certain terms equal to zero. Here the obvious choice is $a = -b$, rendering $|a| = \sqrt{-r/(8u)}$, the final configuration becomes

$$\Phi = \sqrt{\frac{-r}{8u}} e^{i\theta} \begin{pmatrix} 1 \\ -1 \\ 1 \\ -1 \end{pmatrix} , \quad \forall \theta \in [0, 2\pi) . \tag{7.52}$$

This configuration preserves the $G$ symmetry and $\mathbb{Z}_2 \subset H = \mathbb{Z}_4$ symmetry. From the strict 2-group perspective it preserves $(\mathbb{Z}_2, \mathbb{Z}_2, 0, \mathrm{id})$ symmetry. From the weak 2-group perspective, it preserves the $(\mathbb{Z}_2, \mathbb{Z}_2, \mathrm{id}, 0)$ symmetry, the 0-form and 1-form symmetry groups are preserved respectively, but the Postnikov class is trivialized.

**With Wilsonian $H$-Action and $V_1(\Phi)$**

The analysis procedure is largely the same. The $\Phi$ configurations can be classified as follows,

1. Configuration

$$\Phi = \sqrt{\frac{-r}{2u}} e^{i\theta} \begin{pmatrix} 1 \\ 0 \\ 0 \\ 0 \end{pmatrix} \tag{7.53}$$

    preserves the entire 2-group symmetry from both strict and weak perspective;

2. Configuration

$$\Phi = \sqrt{\frac{-r}{2u}} \begin{pmatrix} a \\ 0 \\ b \\ 0 \end{pmatrix} , \quad |a|^2 + |b|^2 = 1 , \quad a, b \neq 0 \tag{7.54}$$

    preserves the $(\mathbb{Z}_2, \mathbb{Z}_2, \times 2, \mathrm{id})$ sub-2-group symmetry, as before, the $\Pi_1 = \Pi_2 = \mathbb{Z}_2$ are preserved but the Postnikov class is rendered trivial;

3. Configuration

$$\Phi = \sqrt{\frac{-r}{2u}} \begin{pmatrix} a \\ b \\ c \\ b \end{pmatrix} , \quad |a|^2 + 2|b|^2 + |c|^2 = 1 , \quad b \neq 0 \tag{7.55}$$

    preserves only $G$ symmetry action, thus the remaining 2-group symmetry is $(\mathbb{Z}_2, 0, 0, \mathrm{id})$. From the weak perspective, only $\Pi_1 = \mathbb{Z}_2$ survives.

**With Wilsonian $H$-Action and $V_3(\Phi)$**

Adding $|\Phi^2|$ term into the potential drastically changes the structure of the equation of motion. Here we analyze a few types of solutions.

1. The first type of solution is

$$\Phi = \sqrt{\frac{-(r+w)}{2u}} e^{i\theta} \begin{pmatrix} 1 \\ 0 \\ 0 \\ 0 \end{pmatrix} \ , \ r + w < 0, u > 0 \tag{7.56}$$

   which preserves the entire 2-group symmetry.

2. The second type of solution is

$$\Phi = \sqrt{\frac{-(r+w)}{2u}} e^{i\theta} \begin{pmatrix} 0 \\ 1 \\ 0 \\ 0 \end{pmatrix} \ , \ \Phi = \sqrt{\frac{-(r+w)}{2u}} e^{i\theta} \begin{pmatrix} 0 \\ 0 \\ 0 \\ 1 \end{pmatrix} \ , \ r + w < 0, u > 0 \tag{7.57}$$

   which completely breaks the 2-group symmetry.

3. The third type of solution is

$$\Phi = \sqrt{-\frac{r+\sqrt{2}w}{4u}} e^{i\theta} \begin{pmatrix} 0 \\ 1 \\ 0 \\ 1 \end{pmatrix} \ , \ \Phi = \sqrt{-\frac{r+\sqrt{2}w}{4u}} e^{i\theta} \begin{pmatrix} 0 \\ 1 \\ 0 \\ -1 \end{pmatrix} \ , \ r + \sqrt{2}w < 0, u > 0 \ . \tag{7.58}$$

   The first solution preserves the $G$-symmetry and breaks the entire $H$-symmetry, preserving the weak 2-group $(\mathbb{Z}_2, 0, 0, 0)$, while the second solution completely breaks the 2-group symmetry.

In summary, we discussed various phases of SSB of the weak 2-group symmetry $(\mathbb{Z}_2, \mathbb{Z}_2, \text{id.}, \beta \neq 0)$ in the strict formulation, and interestingly we found that such a non-split 2-group symmetry can be spontaneously broken to a split 2-group $(\mathbb{Z}_2, \mathbb{Z}_2, \text{id.}, 0)$ with trivial Postnikov class!

### 7.1.3 Approaching continuous 2-groups

In this section we consider the Landau-Ginzberg model of the 2-group

$$1 \rightarrow U(1) \xrightarrow{i} U(1) \times \mathbb{Z}_N \xrightarrow{\partial} \mathbb{Z}_N.\mathbb{Z}_N \xrightarrow{p} \mathbb{Z}_N \rightarrow 1 \tag{7.59}$$

as described in section 2.4. It is a 2-group $(G = \mathbb{Z}_N.\mathbb{Z}_N, H = U(1) \times \mathbb{Z}_N, \partial, \triangleright)$ with

$$\partial(e^{2\pi i a}, b) = (b, 0) \ , \ (a, b) \triangleright (e^{2\pi i c}, d) = (e^{2\pi i (c + \frac{bd}{N})}, d) \tag{7.60}$$

and the addition in $G = \mathbb{Z}_N.\mathbb{Z}_N$ is

$$(a, b) + (c, d) = \begin{cases} (a + c, b + d) & (b + d < N) \\ (a + c + m, b + d) & (\text{otherwise}) \end{cases} \ . \tag{7.61}$$

Here we will approach the $U(1) \times \mathbb{Z}_N$ continuous group by considering a sequence of $\{\mathbb{Z}_M \times \mathbb{Z}_N\}_M$ with increasing $M$ to infinity. In other words, we consider the limit of $\{(G = \mathbb{Z}_N.\mathbb{Z}_N, H = \mathbb{Z}_M \times \mathbb{Z}_N, \partial, \triangleright)\}_M$ with

$$\partial(a,b) = (b,0) \ , \ (a,b) \triangleright (c,d) = \left(c + \frac{bd}{N}, d\right) \ . \tag{7.62}$$

For the action to be well-defined, we require that $\gcd(M,N) = 1$.

We can build an automorphism 2-representation for such a discrete 2-group $(\mathbb{Z}_N.\mathbb{Z}_N, \mathbb{Z}_M \times \mathbb{Z}_N, \partial, \triangleright)$ and build the Landau-Ginzberg theory as we previously performed. We consider the 2-matter field $\Phi$ to take value in $\mathbb{C}[\mathbb{Z}_M \times \mathbb{Z}_N]$.

The question is what kind of terms can appear in the potential $V(\Phi)$. Of course $|\Phi|^2$ and terms proportional to that can appear, and all the $\Phi^k$ terms do not survive the limit $M \to \infty$, since such term depends explicitly on $M$ and $N$ to satisfy 2-group global symmetry in the Lagrangian.

Therefore, we can consider the previously considered potential

$$V_1(\Phi) = r|\Phi|^2 + u|\Phi|^4 \ , \ (r < 0 \ , \ u > 0) \tag{7.63}$$

with static solution satisfying

$$|\Phi| = \sqrt{\frac{-r}{2u}} \ . \tag{7.64}$$

Passing to the $M \to \infty$ limit, this should become

$$\Phi = \sum_{k=0}^{N-1} \int_0^{2\pi} f(\theta, k) \left[e^{i\theta}, k\right] \, \mathrm{d}\theta$$
$$\sum_{k=0}^{N-1} \int_0^{2\pi} |f(\theta, k)|^2 \, \mathrm{d}\theta = \frac{-r}{2u} \ , \tag{7.65}$$

where $[e^{i\theta}, k]$ signifies group element.

### With Natural $H$-Action

There are several possible types of solutions. The first type is uniform distribution of coefficients,

$$f(\theta, k) = \sqrt{\frac{-r}{4\pi u N}} e^{i\alpha} \ , \ \alpha \in \mathbb{R} \ . \tag{7.66}$$

This configuration preserves the entire 2-group symmetry. Also we can observe another type of solution

$$f(\theta, k) = \sqrt{\frac{-r}{4\pi u N}} e^{2\pi i P\theta} \ , \ P \in \mathbb{Z}_{>0} \tag{7.67}$$

which breaks $H = U(1) \times \mathbb{Z}_N$ down to $H' = \mathbb{Z}_P \times \mathbb{Z}_N$, and the $G = \mathbb{Z}_N.\mathbb{Z}_N$ is broken down to $G' = \mathbb{Z}_N.\mathbb{Z}_{N/Q}$, where $Q$ is the smallest positive integer satisfying

$$Q|N \ , \ \frac{QP}{N} \in \mathbb{Z} \ , \tag{7.68}$$

and we will denote $K = N/Q$. While the additive rule signifying the Postnikov class is unchanged, the Postnikov class would generically be changed. Since the weak symmetries after the SSB are given by $\Pi'_1 = \mathbb{Z}_K$, $\Pi'_2 = \mathbb{Z}_P$, the group cohomology accommodating the Postnikov class is $H^3_{\mathrm{grp}}(\mathbb{Z}_K, \mathbb{Z}_P) \cong \mathbb{Z}_{\gcd(P,K)}$.

Since all groups are subgroups of the entire 2-group symmetry while preserving the algebraic structure of $i$, $\partial$ and $p$, each step of the calculation of the Postnikov class can be directly transformed by substituting the group elements with sub-group elements (*id est*, cosets).

More precisely, the exact sequence after the SSB is

$$1 \to \mathbb{Z}_P \xrightarrow{i} \mathbb{Z}_P \times \mathbb{Z}_N \xrightarrow{\partial} \mathbb{Z}_N.\mathbb{Z}_K \xrightarrow{p} \mathbb{Z}_Q \to 1 \,, \tag{7.69}$$

where we still have

$$i: \ a \to (a,0) \ , \ \partial: \ (a,b) \to (b,0) \ , \ p: \ (c,d) \to d \,, \tag{7.70}$$

$$(a,b) \rhd (c,d) = (c + bd \ (\mathrm{mod} \ P), d) \tag{7.71}$$

and the group operation on $\mathbb{Z}_N.\mathbb{Z}_K$:

$$(a,b) + (c,d) = \begin{cases} (a+c, b+d) & b+d < \frac{N}{Q} \\ (a+c+m, b+d) & \text{otherwise} \end{cases} \,. \tag{7.72}$$

As a result the Postnikov class inherited from $m \in \mathbb{Z}_N \cong H^3_{\mathrm{grp}}(\mathbb{Z}_N, U(1))$ is

$$[m] = m \ \mathrm{mod} \ \gcd(P,K) \in \mathbb{Z}_{\gcd(P,K)} \cong H^3_{\mathrm{grp}}(\mathbb{Z}_K, \mathbb{Z}_P) \,. \tag{7.73}$$

Therefore, such configuration completely trivializes the Postnikov class if $m|\gcd(P,K)$.

**With Wilsonian $H$-Action**

With Wilsonian $H$ action, the only configuration that preserves the $H$-symmetry is to concentrate on the identity. Consider the sequence of $\mathbb{C}[\mathbb{Z}_M \times \mathbb{Z}_N]$, the coefficients are given by

$$f(p,k) = \sqrt{\frac{-r}{2u}} e^{i\alpha} \delta_{p,0} \delta_{k,0} \,. \tag{7.74}$$

The reason we fall back to finite group case is that for $U(1) \times \mathbb{Z}_N$, we would encounter the difficulty of defining a square root of Dirac Delta function. Since only identity element enjoys

a non-zero coefficient, it automatically preserves the $G$-symmetry, rendering the entire 2-group symmetry preserved from both strong/weak category perspective.

The previously considered

$$f(\theta, k) = \sqrt{\frac{-r}{4\pi u N}} e^{i\alpha} \tag{7.75}$$

solution, however, breaks the entire $H$ symmetry while preserving the entire $G$ symmetry. In weak 2-group perspective, it preserves $(\Pi_1 = \mathbb{Z}_N.\mathbb{Z}_N, 0, 0, 0)$.

Then, the previously considered

$$f(\theta, k) = \sqrt{\frac{-r}{4\pi u N}} e^{2\pi i P \theta} \ , \ \ P \in \mathbb{Z}_{>0} \tag{7.76}$$

solution with

$$Q|N \ , \ \frac{QP}{N} \in \mathbb{Z} \ , K = N/Q \tag{7.77}$$

would preserve only $(\Pi_1 = \mathbb{Z}_N.\mathbb{Z}_K, 0, 0, 0)$.

## 7.2 Screening center 2-form symmetry in 2-group gauge theory

In the scenario of screening 1-form symmetry, if one can construct a Wilson line with end points (i.e. if there is a field $\phi$ in the same representation with the Wilson line),

$$\phi(x)^\dagger \mathcal{P}[e^{i\int_x^y A}]\phi(y) \ , \tag{7.78}$$

which becomes gauge invariant under

$$\phi(x) \mapsto e^{-i\alpha(x)}\phi(x) \ , \ \ \mathcal{P}[e^{i\int_x^y A}] \mapsto e^{-i\alpha(x)}\mathcal{P}[e^{i\int_x^y A}]e^{i\alpha(y)} \ , \tag{7.79}$$

then we say the matter field screens the 1-form symmetry. In many cases, the 1-form symmetry acting on the Wilson lines are the center 1-form symmetry, i.e., $G^{(1)} = Z(G^{(0)})$.

We can build an analog mechanism in 2-group gauge theories. In pure 2-group gauge theories without matter, consider a closed Wilson surface

$$\mathcal{W}_\mathcal{A}(\Sigma_{\gamma_1, \gamma_2}) = \mathcal{P} \exp\left( i \int_{\Sigma_{\gamma_1, \gamma_2}} \mathcal{A} \right) \tag{7.80}$$

where $\gamma_1 = \gamma_2$ and thus the surface $\Sigma_{\gamma_1, \gamma_2}$ resembles a spindle. This operator also possess a center symmetry. Suppose there is a 2-form field $\lambda \in \Omega^2(M, Z(\mathfrak{h}))$ valued in the center of the Lie algebra $\mathfrak{h}$, since automorphisms shall preserve the center, we have

$$\oint_A (\lambda) = \int_0^1 d\sigma \ \iota_K \left( W_A^{-1}[X](\sigma, 1) \rhd \lambda(\sigma) \right) \in \Omega^1(\mathcal{P}(M), Z(\mathfrak{h})) \ . \tag{7.81}$$

Thus we arrive at

$$[\oint_A (B), \oint_A (\lambda)]_{\mathfrak{h}} = [\mathcal{A}, \oint_A (\lambda)]_{\mathfrak{h}} = 0 \ .$$

Following the previous formalism of center 1-form symmetry in a 0-form symmetry gauge theory, we have a center 2-form symmetry for a 2-group gauge theory.

Now we can elaborate on the idea of screening a 2-form center symmetry with the presence of 2-matter. Since under $H$-gauge transformation, the 2-matter field $\Phi$ transforms as (6.56), we can also define a gauge-invariant term

$$\Phi_{\gamma_1}^{\dagger} \mathcal{W}_{\mathcal{A}}(\Sigma_{\gamma_1,\gamma_2}) \Phi_{\gamma_2} \ . \tag{7.82}$$

As a result, in a 2-group gauge theory with matter, the center 2-form symmetry can be screened.

## 7.3 Higgs mechanism

The Higgs mechanism of our effective model does not differ much to the Higgs mechanism of the ordinary gauge theory. Suppose that we take the form of potential and the ansatz of $\Phi$ matter field to be

$$V(\Phi^{\dagger}\Phi) = \frac{1}{4}\lambda \Big( \Phi^{\dagger}\Phi - \frac{v^2}{2} \Big)^2 \tag{7.83}$$

$$\Phi(\gamma) = \frac{1}{\sqrt{2}}(v + \rho(\gamma)) \exp\Big( \frac{i\chi(\gamma)}{v} \Big) \ . \tag{7.84}$$

With this assumption, the action becomes

$$\begin{aligned}
\mathcal{L} = &-\frac{1}{2}\partial^{(\mu,\sigma)}\rho\partial_{(\mu,\sigma)}\rho - \frac{1}{2}\Big(1 + \frac{\rho}{v}\Big)^2 (\partial_{(\mu,\sigma)}\chi - ev\mathcal{A}_{(\mu,\sigma)})(\partial^{(\mu,\sigma)}\chi - ev\mathcal{A}^{(\mu,\sigma)}) \\
&- \frac{1}{4}\lambda\Big(v^2\rho^2 - v\rho^3 - \frac{1}{4}\rho^4\Big) - \frac{1}{4}\mathcal{F}^{(\mu,\sigma)(\nu,\sigma')}\mathcal{F}_{(\mu,\sigma)(\nu,\sigma')} \ .
\end{aligned} \tag{7.85}$$

Just as in QFT, here we can do a gauge transformation to make

$$(\partial_{(\mu,\sigma)}\chi - ev\mathcal{A}_{(\mu,\sigma)})^2 \mapsto e^2v^2\mathcal{A}^2 \ . \tag{7.86}$$

The Lagrangian becomes

$$\begin{aligned}
\mathcal{L} = &-\frac{1}{2}\partial^{(\mu,\sigma)}\rho\partial_{(\mu,\sigma)}\rho - \frac{1}{4}\lambda\Big(v^2\rho^2 - v\rho^3 - \frac{1}{4}\rho^4\Big) \\
&- \frac{1}{4}\mathcal{F}^{(\mu,\sigma)(\nu,\sigma')}\mathcal{F}_{(\mu,\sigma)(\nu,\sigma')} - \frac{1}{2}\Big(1 + \frac{\rho}{v}\Big)^2 e^2v^2\mathcal{A}^2 \ .
\end{aligned} \tag{7.87}$$

We can observe that the Higgs mechanism of 2-gauge theories with 2-matter resembles that of ordinary gauge theories. With proper gauge-fixing, one can have a massive boson in the path space.

# 8 Discussions

In this paper, we provided a Lagrangian formulation of 2-matter charged under 2-group symmetries in the path space, and discussed the spontaneous symmetry breaking of 2-group symmetries under such Landau-Ginzburg model. A key technique is to consider the strictification of weak 2-group symmetry, and construct the 2-matter $\Phi$ living in an automorphism 2-representation. Using different Landau-Ginzburg potential $V(\Phi)$, we can realize different symmetry breaking patterns including the one from a non-split 2-group to a split 2-group. In the future, it would be interesting to further investigate the dynamics of the 2-group gauge theory defined in the path space, and it is also worthy to extend the discussions to other 2-representations and 2-groups.

For weak 3-groups, we only provided the strictification procedure for special scenarios with either $\Pi_1 = 0$ or $\Pi_2 = 0$. Hence a more complete discussion of the strictification of general weak 3-groups would be subject to future work. The discussions of 3-representations of 3-groups are also quite limited, and we hope to further investigate the structures of 3-representations in a more detailed manner, for instance, using automorphism 3-groups $\mathcal{A}ut(\mathcal{G})$ for 2-groups $\mathcal{G}$. As a goal of this line of research, we hope to formulate higher representations of higher groups in a more clear, algebraic language.

We have shown that when generalized into $n$-groups and $n$-matter, the brane fields naturally appear into the higher gauge theory with higher-matter (for higher-form symmetry case, it was discussed in [72]). A natural question would be to build an algebraic and physical model to formulate such brane-field mechanism for general $n$-group symmetries and further, generic higher-categorical symmetries.

Another important direction is to apply this formulation to physical systems with higher-group global symmetries, and discuss the SSB of these symmetries, such the lattice models in [33,39] which possess higher-group symmetries and realize topological error correction codes. It would also be interesting to further investigate concrete continuous QFT models with higher-group symmetries, and see how to encode its SSB structure in the path space Landau-Ginzburg theory. There could also be interesting interplay with supersymmetry if we also consider the fermionic 2-group gauge theory discussed in [73].

We briefly discussed the screening of center 2-form symmetry in 2-group gauge theory, without specifying the particular 2-representation. It is known in the usual gauge theory that matter under different representations shall provide different screenings of the center 1-form symmetries. The role of different higher-representations in screening such center higher-from symmetries shall be subject to future research.

The Higgs mechanism of field theory in path space is also to be developed. Future research may consider the formulation of Goldstone bosons and their counting in path space, and relate such theories with physical models.

**Acknowledgments.** We thank Lakshya Bhardwaj, Shi Cheng, Clay Cordova, Ho Tat Lam, Sakura Schafer-Nameki, Urs Schreiber, Yi Zhang for discussions. Ran Luo and Yi-Nan Wang are supported by National Natural Science Foundation of China under Grant No. 12175004, by Peking University under startup Grant No. 7100603667, and by Young Elite Scientists Sponsorship Program by CAST (2022QNRC001, 2023QNRC001). Research of Ruizhi Liu at Perimeter Institute is supported in part by the Government of Canada through the Department of Innovation, Science and Industry Canada and by the Province of Ontario through the Ministry of Colleges and Universities.

## A    Triviality of Postnikov class after the strictification

We present a general proof that given exact sequence

$$1 \to \Pi_2 \to H \to G \xrightarrow{p} \Pi_1 \to 1 \tag{A.1}$$

the Postnikov class satisfies $p^*\beta = 0$.

Let us consider following lemma case first.

**Lemma 1.** *Given a principal $G$ bundle ($G$ is assumed to be a 1-group) $P \xrightarrow{p} M$, then the pullback bundle $p^*P \to P$ is trivial.*

*Proof.* Recall that for principal bundles, they are trivial iff they admit a global section (in contrast to vector bundles, they always have trivial zero section). Set theoretically, the pullback bundle is constructed as $P \times_\pi P := \{(x,y) \in P \times P | \pi(x) = \pi(y)\}$, there is a canonical diagonal map $\Delta : P \to P \times_\pi P$ given by $x \to (x,x)$. As a result, the pullback bundle admits a global section and hence trivial. $\qquad\square$

Let us go back to 2-group case. Consider the Postnikov tower of the bottom layer, as indicated in following diagram.

$$
\begin{array}{ccc}
B^2\Pi_2 & \longrightarrow & B\mathcal{G} \\
& & \downarrow{\scriptstyle p} \\
& & B\Pi_1
\end{array}
$$

The classifying space of 2-group $\mathcal{G}$ is a total space of a principal $B^2\Pi_2$ bundle over $B\Pi_1$. As a result, it is labelled by a particular class in Čech cohomology $\check{H}^1(B\Pi_1; B^2\Pi_2)$. We have known that this is Postnikov class $\beta$.

Now the Postnikov class is trivial iff the bundle itself is trivial. We have argued that $p^*B\mathcal{G}$ is trivial, as a result $p^*\beta = 0$.

This has confirmed that Postnikov class is necessarily zero after strictification.

There is another algebraic proof based on decomposition of 2-groups:

$$
\begin{array}{ccccccccc}
1 & \longrightarrow & \Pi_2 & \xrightarrow{\ i\ } & H & \xrightarrow{\ \partial\ } & \mathrm{im}\partial & \longrightarrow & 1 \\
 & & \downarrow & & \downarrow & & \downarrow & & \\
1 & \longrightarrow & \mathrm{im}\partial & \xrightarrow{\ i\ } & G & \xrightarrow{\ p\ } & \Pi_1 & \longrightarrow & 1
\end{array}
\tag{A.2}
$$

We view the bottom line as a group extension and top line as a sequence of coefficients. Hence we have Bockstein homomorphism:

$$
\mathrm{Bock} : H^2_{grp}(\Pi_1; \mathrm{im}\partial) \to H^3_{grp}(\Pi_1; \Pi_2)
\tag{A.3}
$$

This is completely valid at least when $H$ is abelian. We denote the extension class of the bottom line as $\alpha \in H^2_{grp}(\Pi_1; \mathrm{im}\partial)$, so

$$
p^*(\mathrm{Bock}(\alpha)) = \mathrm{Bock}(p^*\alpha) = 0
\tag{A.4}
$$

(This is due to the naturality of Bock.)

# B   More examples of strictification of weak 2-groups

## B.1   $\Pi_1 = \mathbb{Z}_N$, $\Pi_2 = \mathbb{Z}_M$

We discuss the case for a general $\Pi_1 = \mathbb{Z}_N$, $\Pi_2 = \mathbb{Z}_M$. In this case, $H^3(B\mathbb{Z}_N; \mathbb{Z}_M)$ is generated by the $N$-torsion subgroup of $\mathbb{Z}_M$, which consists of elements $\{a \in \mathbb{Z}_M | Na = 0 \ (\mathrm{mod}\ M)\}$. Hence we have

$$
H^3(B\mathbb{Z}_N; \mathbb{Z}_M) = \mathbb{Z}_{\gcd(N,M)}.
\tag{B.1}
$$

The general exact sequence is

$$
1 \to \mathbb{Z}_M \xrightarrow{\ i\ } \mathbb{Z}_{MK} \xrightarrow{\ \partial\ } \mathbb{Z}_{NK} \xrightarrow{\ p\ } \mathbb{Z}_N \to 1
\tag{B.2}
$$

The maps are

$$
i: a \to Ka, \quad \partial: a \to Na \ (\mathrm{mod}\ NK), \quad p: (\mathrm{mod}\ N).
\tag{B.3}
$$

The section $s : \mathbb{Z}_N \to \mathbb{Z}_{NK}$ is chosen as $s(g) \equiv g$, and the map $f : \mathbb{Z}_N \times \mathbb{Z}_N \to \mathbb{Z}_{NK}$ is

$$f(g,h) = \begin{cases} 0 & (g+h < N) \\ N & (g+h \geq N) \end{cases}. \tag{B.4}$$

The map $F : \mathbb{Z}_N \times \mathbb{Z}_N \to \mathbb{Z}_{MK}$ is

$$F(g,h) = \begin{cases} 0 & (g+h < N) \\ 1 & (g+h \geq N) \end{cases}. \tag{B.5}$$

In order to have a non-trivial Postnikov class, it is required that the short exact sequence

$$1 \to \ker(\partial) \xrightarrow{\partial} \mathbb{Z}_{NK} \xrightarrow{p} \mathbb{Z}_N \to 1 \tag{B.6}$$

is a non-split central extension ($H^2(B\mathbb{Z}_N; \mathbb{Z}_K) \neq 0$), such that $s$ cannot be taken as a homomorphism.

The Postnikov class element $c : \mathbb{Z}_N \times \mathbb{Z}_N \times \mathbb{Z}_N \to \mathbb{Z}_M$ and the action $\rhd : \mathbb{Z}_N \to \mathrm{Aut}(\mathbb{Z}_M)$ depends on the choice of action $\rhd : \mathbb{Z}_{NK} \to \mathrm{Aut}(\mathbb{Z}_{MK})$.

If the action $\rhd : \mathbb{Z}_{NK} \to \mathrm{Aut}(\mathbb{Z}_{MK})$ is trivial, then both the Postnikov class and the action of $\mathbb{Z}_N$ on $\mathbb{Z}_M$ will be trivial. To write down all the non-trivial actions, we first need a mathematical fact that $\mathrm{Aut}(\mathbb{Z}_{MK})$ is generated by the units of $\mathbb{Z}_{MK}$, which form a set $\{p \in \mathbb{Z}_{MK} | \gcd(p, MK) = 1\}$. Each element $p$ corresponds to the following action:

$$a \rhd_p b = (p^a \ (\mathrm{mod} \ MK)) \cdot b. \tag{B.7}$$

Furthermore, to satisfy the conditions (2.1) of a strict 2-group, it is also required that

$$\begin{aligned} p &= 1 \ (\mathrm{mod} \ K), \\ p^N &= 1 \ (\mathrm{mod} \ MK). \end{aligned} \tag{B.8}$$

For each $p \in \{0, 1, \ldots, MK - 1\}$ satisfying (B.8), there is a well-defined strict 2-group $(\mathbb{Z}_{NK}, \mathbb{Z}_{MK}, \partial, \rhd_p)$ that gives rise to different weak 2-groups $(\mathbb{Z}_N, \mathbb{Z}_M, \rho, \beta)$.

In particular, when $p = 1 \ (\mathrm{mod} \ M)$, the action $\rhd : \mathbb{Z}_N \to \mathbb{Z}_M$ is trivial, and we get an element in the untwisted cohomology $H^3(B\mathbb{Z}_N, \mathbb{Z}_M)$. On the other hand, if $p \neq 1 \ (\mathrm{mod} \ M)$, the action $\rhd : \mathbb{Z}_N \to \mathbb{Z}_M$ is non-trivial, and we get an element in the twisted cohomology $H^3_\rho(B\mathbb{Z}_N, \mathbb{Z}_M)$.

Actually, all classes in (B.1) can be constructed in this way, as we will describe below.

First, we have three constraints on the choice of $p$ in (B.7): two of them are given by (B.8) and the other is $\gcd(p, MK) = 1$.

For general $K$, we have $p = 1 \ (\mathrm{mod} \ K)$, hence we can write

$$p = wK + 1 \ (\mathrm{mod} \ MK), w = 1, 2, ..., M - 1 \tag{B.9}$$

The untwisted condition is $p = 1 (\mathrm{mod}\, M)$, which amounts to say $M | wK$. We require

$$(wK+1)^N - 1 = NwK + \frac{N(N-1)}{2}(wK)^2 + ... = 0 \,(\mathrm{mod}\, MK) \tag{B.10}$$

Note $M | wK$, above equation is actually equivalent to $M | Nw$. Now we write $M = md$, $N = nd$ where $d = \gcd(M, N)$ and $\gcd(m, n) = 1$. The condition is $m | nw$ hence $m | w$.

We can now write an admissible $w$ as $w = ms$, $s = 0, 1, .., d - 1$. But we need $M | wK$ (untwisted condition), which now simplifies to $d | sK$ for any $s = 0, 1, 2, ..., d - 1$. Hence we have $d | K$ by taking $s = 1$. So we have $K \geq d = \gcd(M, N)$.

Let us fix $K = d$ and choose any admissible $w$, one can compute the Postnikov class on $(g_1, g_2, g_3)$, $0 \leq g_1, g_2, g_3 < N$

$$c(g_1, g_2, g_3) = \begin{cases} wg_1, & g_2 + g_3 \geq N \\ 0, & g_2 + g_3 < N \end{cases} \tag{B.11}$$

This class has some general features:

1. It's linear in the first argument $g_1$.

2. We can check, using GAP package that it's indeed a nontrivial cohomology class parameterized by group actions.

Let's comment on twisted case, i.e., there is nontrivial group action of $\Pi_1$ on $\Pi_2$. We still have (B.9) and (B.10) in this case. Given $M, N$, one can always check if there is any $p \neq 1 \,(\mathrm{mod}\, M)$ satisfying these equations, however, it's unlikely to obtain a general solution in this case. We will be content with several examples described below:

1. $M = N = 2$. This is the case in section 2.3. The action $\mathbb{Z}_2 \to \mathrm{Aut}(\mathbb{Z}_2)$ is always trivial. The minimal $K$ is

$$\mathrm{min.}(K) = \begin{cases} 1 & (\text{trivial } \beta) \\ 2 & (\text{non-trivial } \beta) \end{cases} \tag{B.12}$$

2. $M = N = 3$. In this case there is no non-trivial action of $\mathbb{Z}_3 \to \mathrm{Aut}(\mathbb{Z}_3)$.

   The smallest $K$ with a non-trivial $H^2(B\mathbb{Z}_3; \mathbb{Z}_K)$ is $K = 3$. The actions $\rhd_p$ satisfying (B.8) are $p = 1, 4, 7$, which one-to-one corresponds to the three elements in $H^3(B\mathbb{Z}_3; \mathbb{Z}_3) = \mathbb{Z}_3$.

   Hence for any non-trivial Postnikov class $\beta$, the smallest $K = 3$.

$$\mathrm{min.}(K) = \begin{cases} 1 & (\text{trivial } \beta) \\ 3 & (\text{non-trivial } \beta) \end{cases} \tag{B.13}$$

3. $N = 2$, $M = 4$. In this case there exists a non-trivial action of $\mathbb{Z}_4 \to \text{Aut}(\mathbb{Z}_4) = \mathbb{Z}_2$. $H^3(B\mathbb{Z}_2; \mathbb{Z}_4) = \mathbb{Z}_2$ and we denote its generator by $\alpha$.

$$\text{min.}(K) = \begin{cases} 1 & \text{(trivial } \beta) \\ 2 & \text{(non-trivial } \beta \text{ or non-trivial } \rho) \end{cases} \tag{B.14}$$

4. $M = N = 4$. In this case there exists a non-trivial action of $\mathbb{Z}_4 \to \text{Aut}(\mathbb{Z}_4) = \mathbb{Z}_2$. $H^3(B\mathbb{Z}_4; \mathbb{Z}_4) = \mathbb{Z}_4$ and we denote its generator by $\alpha$.

The smallest $K$ with a non-trivial $H^2(B\mathbb{Z}_4; \mathbb{Z}_K)$ is $K = 2$. For $K = 2$, the actions $\rhd_p$ satisfying (B.8) are $p = 1, 3, 5, 7$. For $p = 5$, the induced action of $\mathbb{Z}_4 \to \text{Aut}(\mathbb{Z}_4)$ is trivial, and the Postnikov class $\beta = 2\alpha$. For $p = 3$ or $p = 7$, the induced action of $\mathbb{Z}_4 \to \text{Aut}(\mathbb{Z}_4)$ is non-trivial.

On the other hand, we can also choose $K = 4$, and the actions $\rhd_p$ satisfying (B.8) are $p = 1, 5, 9, 13$. For these cases, the induced action of $\mathbb{Z}_4 \to \text{Aut}(\mathbb{Z}_4)$ is always trivial. The Postnikov class $\beta = \frac{1}{4}(p - 1)\alpha$.

As a conclusion, for a given weak 2-group $(\mathbb{Z}_4, \mathbb{Z}_4, \rho, \beta)$, the smallest $K$ is

$$\text{min.}(K) = \begin{cases} 1 & \text{(trivial } \beta) \\ 2 & \text{(non-trivial } \rho \quad \text{or trivial } \rho, \ \beta = 2\alpha) \\ 4 & \text{(trivial } \rho, \ \beta = (2k+1)\alpha) \end{cases} \tag{B.15}$$

## B.2 $\Pi_1 = \mathbb{Z}_N$, $\Pi_2 = \bigoplus_{i=1}^m \mathbb{Z}_{M_i}$

In this case, we note following identity in cohomology groups:

$$H^3(\Pi_1; \bigoplus_{i=1}^m \mathbb{Z}_{M_i}) \simeq \bigoplus_{i=1}^m H^3(\Pi_1; \mathbb{Z}_{M_i}) \tag{B.16}$$

It means, essentially summands in $\Pi_2$ are decoupled, we can treat them separately. As a result, we can obtain following crossed module extension:

$$1 \to \bigoplus_{i=1}^m \mathbb{Z}_{M_i} \to \mathbb{Z}_{M_j K} \oplus \bigoplus_{i \neq j} \mathbb{Z}_{M_i} \to \mathbb{Z}_{NK} \to Z_N \to 1 \tag{B.17}$$

The boundary map $\partial : \mathbb{Z}_{M_j K} \oplus \bigoplus_{i \neq j} \mathbb{Z}_{M_i} \to \mathbb{Z}_{NK}$ is given by

$$\partial(x_1, x_2, ...x_j..., x_m) := Nx_j, \ (\text{mod } NK) \tag{B.18}$$

where $x_j \in \mathbb{Z}_{M_j K}$ and $x_i \in \mathbb{Z}_{M_i}$ for $i \neq j$ and $K$ can be chosen as $K = \gcd(M_j, N)$ as we described in sec B.1. These extensions realized all classes in (B.17).

## B.3    $\Pi_1 = \bigoplus_{i=1}^n \mathbb{Z}_{N_i}$, $\Pi_2 = \mathbb{Z}_M$

For simplicity, we will begin with $n = 2$ case, i.e., $\Pi_1 = \mathbb{Z}_{N_1} \times \mathbb{Z}_{N_2}$.

We define a group

$$\Gamma_\alpha(N_1, N_2, K) := \langle a, b, x | x^K = a^{N_1} = b^{N_2} = 1, ax = xa, bx = xb, bab^{-1} = ax^\alpha \rangle \quad \text{(B.19)}$$

where $K \in \mathbb{Z}^{>0}$ is a free parameter to be specified later. The group can be viewed as some kind of generalization of finite Heisenberg group. Note $bab^{-1} = ax^\alpha$ imposes a strong constraint on $\alpha$: taking $N_1$-th power of both sides, we find:

$$x^{\alpha N_1} = 1 \quad \text{(B.20)}$$

which means $K | \alpha N_1$. Analogously, $K | \alpha N_2$, Bezout's identity shows

$$K | \alpha \gcd(N_1, N_2) \quad \text{(B.21)}$$

Denote $K = kd$ where $d := \gcd(N_1, N_2, K)$, we now obtain admissible values of $\alpha$:

$$\alpha = 0, k, ..., (d-1)k < kd = K \quad \text{(B.22)}$$

This group satisfies following exact sequence:

$$0 \to \mathbb{Z}_K \overset{i}{\hookrightarrow} \Gamma_\alpha(N_1, N_2, K) \overset{p}{\longrightarrow} \mathbb{Z}_{N_1} \times \mathbb{Z}_{N_2} \to 1 \quad \text{(B.23)}$$

where we write $Z_K$ additively and $\mathbb{Z}_{N_1} \times \mathbb{Z}_{N_2}$ multiplicatively. The maps are $i(a) = x^a$, $\forall a \in \mathbb{Z}_K$, $p$ is a surjection whose kernel is generated by $x$, i.e., $p(x) = 1$. Sometimes, we will denote $p(a)$ as $a$ and $p(b)$ as $b$ if there is no confusion.

We now construct the crossed module extension. Note the cohomology class is calculated by Künneth formula

$$H^3(\mathbb{Z}_{N_1} \times \mathbb{Z}_{N_2}; \mathbb{Z}_M) \simeq \bigoplus_{i+j=3} H^i(\mathbb{Z}_{N_1}; H^j(\mathbb{Z}_{N_2}; \mathbb{Z}_M)) \quad \text{(B.24)}$$

Note following result:

$$H^k(B\mathbb{Z}_m; \mathbb{Z}_n) \simeq \mathbb{Z}_{\gcd(m,n)}, \ k > 0 \quad \text{(B.25)}$$

Now, we have

$$H^3(B(\mathbb{Z}_{N_1} \times \mathbb{Z}_{N_2}); \mathbb{Z}_M) \simeq \mathbb{Z}_{\gcd(N_1,M)} \oplus \mathbb{Z}_{\gcd(N_2,M)} \oplus \mathbb{Z}^2_{\gcd(M,N_1,N_2)} \quad \text{(B.26)}$$

where $\mathbb{Z}^2_{\gcd(M,N_1,N_2)} = \mathbb{Z}_{\gcd(M,N_1,N_2)} \oplus \mathbb{Z}_{\gcd(M,N_1,N_2)}$. The first two terms above correspond to those extensions in which $\mathbb{Z}_{N_1}$ or $\mathbb{Z}_{N_2}$ decouple in the sense of following sequences:

$$1 \to \mathbb{Z}_M \to \mathbb{Z}_{MK} \to \mathbb{Z}_{N_1 K} \times \mathbb{Z}_{N_2} \to \mathbb{Z}_{N_1} \times \mathbb{Z}_{N_2} \to 1 \quad \text{(B.27)}$$

where the map $\partial : \mathbb{Z}_{MK} \to \mathbb{Z}_{N_1 K} \times \mathbb{Z}_{N_2}$ is given by $\partial(a) = (N_1 a, 0)$. Symmetrically, we have the same construction involving $\mathbb{Z}_{N_2}$ but $\mathbb{Z}_{N_1}$ decouples.

So we only concentrate on mixing case, i.e., corresponding to $\mathbb{Z}^2_{\gcd(M, N_1, N_2)}$ in the cohomology group. We will argue following extension can realize all the mixing classes:

$$0 \to \mathbb{Z}_M \xrightarrow{i} \mathbb{Z}_{MK} \xrightarrow{\partial} \Gamma_\alpha(N_1, N_2, K) \xrightarrow{\pi} \mathbb{Z}_{N_1} \times \mathbb{Z}_{N_2} \to 1 \tag{B.28}$$

(See (B.19) for the definition of $\Gamma_\alpha$.)

Before diving into details, note following commutative diagram:

where $l = 0, 1, 2, ..., \gcd(K, N_1, N_2)$ and the morphism $\phi_l$ is given by $\phi_l(x) = x'^l$ where $x$ and $x'$ are generators of the centers of $\Gamma_\alpha$ and $\Gamma_{l\alpha}$ respectively. By definition, we know the sequence (B.28) is (weakly) equivalent to

$$0 \to \mathbb{Z}_M \xrightarrow{i} \mathbb{Z}_{lMK} \xrightarrow{\partial} \Gamma_{l\alpha}(N_1, N_2, K) \xrightarrow{\pi} \mathbb{Z}_{N_1} \times \mathbb{Z}_{N_2} \to 1 \tag{B.30}$$

hence, they correspond to the same element in cohomology class $H^3(\Pi_1; \Pi_2)$. For this reason, we can limit us with minimal choice: $\alpha = \frac{K}{\gcd(K, N_1, N_2)}$.

Let's discuss the group action in this case. It is determined by a homomorphism $\Gamma_\alpha \to \mathbb{Z}^\times_{MK}$, where $\mathbb{Z}^\times_{MK}$ is the group of units in $\mathbb{Z}_{MK}$ (i.e., invertible elements under ring multiplication). Since $\Gamma_\alpha$ is generated by $x, a, b$, we only need to determine their images in $\mathbb{Z}^\times_{MK}$. We will denote the image of $a, b, x$ as $p_1, p_2, p_3$ respectively. They are subject to group relations from $\Gamma_{alpha}$, that is

$$p_i^{N_i} = 1 \,(\mathrm{mod}\, MK), \; i = 1, 2 \tag{B.31}$$

Note $x = [b, a] = bab^{-1}a^{-1}$ and $\mathbb{Z}^\times_{MK}$ is abelian, hence $p_3 = 1$, i.e., the center $\mathbb{Z}_K$ must act trivially. And $p_i$'s have to satisfy equivariance (2.1) and Peiffer identity (2.2), explicitly,

$$p_i = 1 \,(\mathrm{mod}\, K), \; i = 1, 2 \tag{B.32}$$

And we assume the weak 2-group is untwisted, hence

$$p_i = 1 \,(\mathrm{mod}\, M), \; i = 1, 2, \tag{B.33}$$

We can combine (B.32) and (B.33), write

$$p_i = w_i \frac{MK}{\gcd(M,K)} + 1 \, (\mathrm{mod} \, MK) \tag{B.34}$$

where $w_i = 0, 1, 2, ..., \gcd(M,K) - 1$. We now solve (B.31). It's easy to see $MK | (\frac{MK}{\gcd(M,K)})^n$ if $n \geq 2$, so if we expand (B.31), we see it is equivalent to

$$\gcd(M,K) | (w_i N_i) \tag{B.35}$$

Let $d_i := \gcd(M, K, N_i)$ and $m_i := \frac{\gcd(M,K)}{d_i}$. We have $w_i = s_i m_i$, $s_i = 0, 1, 2, ..., d_i - 1$. There is no more constraint on group actions.

Let's summarize the discussion on group actions: group $\Gamma_\alpha$ acts on $\mathbb{Z}_{MK}$ as

$$a \triangleright l = (\frac{s_1 MK}{\gcd(M, K, N_1)} + 1) l \, (\mathrm{mod} \, MK) \tag{B.36}$$

$$b \triangleright l = (\frac{s_2 MK}{\gcd(M, K, N_2)} + 1) l \, (\mathrm{mod} \, MK) \tag{B.37}$$

where $s_i = 0, 1, ..., \gcd(M, K, N_i) - 1, i = 1, 2$. There is no further constraint on the group action.

Let's compute the Postnikov cocyle. Following the notation of section B.1, we denote $x = (x_1, x_2) \in \mathbb{Z}_{N_1} \times \mathbb{Z}_{N_2}$ (in additive notation) and $s((1, 0)) = a$, $s((0, 1)) = b$, hence

$$f(x, y) = x^{\alpha x_1 y_2} \tag{B.38}$$

We then lift $f$ to $F(x, y) = \alpha x_1 y_2$. According to (2.9), we have

$$c(x, y, z) = \alpha y_1 z_2 (\sum_{i=1}^{2} \frac{M s_i x_i}{\gcd(M, K, N_i)}), \; s_i = 0, 1, ..., \gcd(M, K, N_i) - 1 \tag{B.39}$$

We can argue that these classes are indeed mixing case. Note $c$ vanishes if it's evaluated on $(x, y, z)$ with $x_1 = y_1 = z_1 = 0$ or $x_2 = y_2 = z_2 = 0$. It cannot happen in non-mixing case (i.e. first two summands in (B.26)) unless the class is trivial.

In fact, if $K = \gcd(M, N_1, N_2)$, then we set $\alpha = 1$ (see the discussion below (B.30)). In this case $s_1, s_2 \in \{0, 1, 2, ..., \gcd(M, N_1, N_2) - 1\}$ in one-to-one correspondence with $\mathbb{Z}^2_{\gcd(M, N_1, N_2)}$. Obviously, $K$ can't be smaller than $\gcd(M, N_1, N_2)$, otherwise, $\gcd(M, N_i, K) \leq K < \gcd(M, N_i, N_2)$, $i = 1, 2$. In that situation, one cannot realize all (mixing) cohomology classes in (B.26).

There is no difficulty to generalize above construction to any finite abelian group (which is product of cyclic groups after all). So one immediate corollary of our construction is that: if one starts with finite abelian $\Pi_1$, $\Pi_2$ (without twist), one can always assume that $H$ in (2.6) is also finite abelian (a cyclic group actually).

## B.4 Examples with non-abelian Lie groups

Let us consider the field theory examples with a weak 2-group symmetry $(\Pi_1, \Pi_2, \mathrm{id.}, \beta)$. $\Pi_1$ is the 0-form symmetry group, which is non-simply-connected. We can write $\Pi_1 = G/(\mathrm{im}\partial)$, where $G$ is a simply-connected Lie group and $\mathrm{im}\partial$ is a subgroup of the center of $G$. $\Pi_2$ is the 1-form symmetry group of the theory.

In the terminology of [26], we have the identification

$$\Pi_1 = \mathcal{F} , \ \Pi_2 = \mathcal{O} , \ G = F , \ H = \mathcal{E} , \ \mathrm{im}\partial = \mathcal{Z} , \tag{B.40}$$

and these groups fit into the following commutative diagram

$$
\begin{array}{ccccccccc}
1 & \longrightarrow & \Pi_2 & \xrightarrow{i} & H & \longrightarrow & \mathrm{im}\partial & \longrightarrow & 1 \\
& & \downarrow & & \downarrow{\scriptstyle\partial} & & \downarrow & & \\
1 & \longrightarrow & \mathrm{im}\partial & \longrightarrow & G & \xrightarrow{p} & \Pi_1 & \longrightarrow & 1
\end{array}
\tag{B.41}
$$

Here $H = \mathcal{E} \subset Z(F) \times Z(G_{\mathrm{gauge}})$ is the maximal subgroup of the product of flavor center $Z(F)$ and gauge center $Z(G_{\mathrm{gauge}}) \supset \mathcal{O}$ that acts trivially on matter fields. For example see the context of 5D SCFTs with M-theory geometric construction [26, 32].

For example, Let us consider the 5d rank-1 SCFT with IR gauge theory description $SU(2)_0$. In [26] it is shown that the theory has a weak 2-group symmetry with $\Pi_1 = SO(3)$, $\Pi_2 = \mathbb{Z}_2$. We first use the following commutative diagram

$$
\begin{array}{ccccccccc}
1 & \longrightarrow & \mathbb{Z}_2 & \xrightarrow{i} & \mathbb{Z}_4 & \longrightarrow & \mathbb{Z}_2 & \longrightarrow & 1 \\
& & \downarrow & & \downarrow{\scriptstyle\partial} & & \downarrow & & \\
1 & \longrightarrow & \mathbb{Z}_2 & \longrightarrow & SU(2) & \xrightarrow{p} & SO(3) & \longrightarrow & 1
\end{array}
\tag{B.42}
$$

We show that the exact sequence

$$1 \to \mathbb{Z}_2 \xrightarrow{i} \mathbb{Z}_4 \xrightarrow{\partial} SU(2) \xrightarrow{p} SO(3) \to 1 \tag{B.43}$$

realizes the non-trivial element $\beta$ in $H^3(BSO(3), \mathbb{Z}_2) = \mathbb{Z}_2$.

Let us denote the group elements in $SU(2)$ by $g$ and the group elements in $SO(3)$ by the conjugacy class $\{g, -g\}$. The identity element of $SO(3)$ is $\{I, -I\}$. The maps in (B.43) are

$$i : \ a \to 2a , \quad \partial : \ a \to e^{\pi i a} I , \quad p : \ g \to \{g, -g\} . \tag{B.44}$$

Let us choose the section $s : SO(3) \to SU(2)$ as a canonical way to embed an $SO(3)$ element into $SU(2)$. The group multiplication in $SO(3)$ can be written as

$$\{g, -g\} \cdot \{h, -h\} = \begin{cases} \{gh, -gh\} & s(g)s(h) = s(gh) \\ \{-gh, gh\} & s(g)s(h) = -s(gh) \end{cases} . \tag{B.45}$$

Hence we can write down the function $f(g, h)$ (in the multiplicative notation) and its uplift $F(g, h)$ (in the additive notation):

$$f(g, h) = \begin{cases} I & \{g, -g\} \cdot \{h, -h\} = \{gh, -gh\} \\ -I & \{g, -g\} \cdot \{h, -h\} = \{-gh, gh\} \end{cases} . \tag{B.46}$$

$$F(g, h) = \begin{cases} 0 & \{g, -g\} \cdot \{h, -h\} = \{gh, -gh\} \\ 1 & \{g, -g\} \cdot \{h, -h\} = \{-gh, gh\} \end{cases} . \tag{B.47}$$

Plug in the formula (2.9), we can compute the following discontinuous map $c(g, h, k)$

$$
\begin{aligned}
c(g, h, k) &= \frac{1}{2}(F(h, k) + F(g, hk) - F(g, h) - F(gh, k)) \\
&= \begin{cases} 1 & s(g)s(hk) = -s(gh)s(k) \,,\; s(g)s(h)s(k) = s(ghk) \\ 0 & \text{other cases} \end{cases} .
\end{aligned}
\tag{B.48}
$$

$c(g, h, k)$ corresponds to the non-trivial element in $H^3(BSO(3); \mathbb{Z}_2)$, because it cannot be written as a coboundary ($\frac{1}{2}F(g, h)$ is not a well-defined function).

On the other hand, if one uses the exact sequence

$$1 \to \mathbb{Z}_2 \xrightarrow{i} \mathbb{Z}_2 \times \mathbb{Z}_2 \xrightarrow{\partial} SU(2) \xrightarrow{p} SO(3) \to 1 \,, \tag{B.49}$$

one can derive $c(g, h, k) \equiv 0$ and the Postnikov class is trivial.

**Physical Example**  Let us show an physical example corresponding to the case of non-trivial $c(g, h, k) \in H^3(BSO(3); \mathbb{Z}_2)$, which can be applied to the case of 5d $SU(2)_0$ SCFT in [26]. Consider a gauge theory with gauge group $G_{\text{gauge}} = SU(2)$ and flavor algebra $\mathfrak{su}(2)$ (with simply-connected group $G = SU(2)$), and the following charges of matter fields:

|          | $\mathfrak{u}(1)_{\text{gauge}}$ | $\mathfrak{u}(1)_{\text{flavor}}$ | $\mathbb{Z}_{2,\text{flavor}}$ |
|----------|-------------|-------------|---------------|
| $\phi_1$ | 2 | $-1$ | 1 |
| $\phi_2$ | 0 | 2 | 0 |

$$\tag{B.50}$$

The listed numbers are the charges under the Cartan subalgebra and center of $G_{\text{gauge}} = SU(2)$ and $\mathfrak{su}(2)$. We have the following observations:

1. Each matter field has integral charge under the linear combination $\frac{1}{4}(\mathfrak{u}(1)_{\text{gauge}} + 2\mathfrak{u}(1)_{\text{flavor}})$. This is the source of $H = \mathbb{Z}_4$. Roughly speaking, the computation of $H = \mathbb{Z}_4$ is achieved by computing the Smith normal form of the charge matrix which involves both $\mathfrak{u}(1)_{\text{gauge}}$ and $\mathfrak{u}(1)_{\text{flavor}}$.

2. The $\mathbb{Z}_2$ center part of the $G = SU(2)$ is a part of the $U(1)_{\text{gauge}}$, hence the actual flavor symmetry group is $\Pi_1 = G/\mathbb{Z}_2 = SO(3)$, which is consistent with the weak 2-group structure.

In summary, one can see that in the strictification (B.43), the group $G = SU(2)$ is bigger than the actual 0-form symmetry group $\Pi_1 = SO(3)$. The group $H = \mathbb{Z}_4$ is also bigger than the actual 1-form symmetry group $\Pi_2 = \mathbb{Z}_2$.

**Gauge theory**    For the strict 2-group, we have the following gauge transformation (following [55] for example)

$$
\begin{aligned}
A' &= \lambda A \lambda^{-1} + \lambda \mathrm{d}\lambda^{-1} + \underline{\partial}\Lambda \\
B' &= B + \delta\Lambda \,.
\end{aligned}
\tag{B.51}
$$

$A$ is the gauge field for $G = SU(2)$, which takes value in the Lie algebra $\mathfrak{su}(2) = \mathfrak{so}(3)$. $B$ is the discrete gauge field for $H = \mathbb{Z}_4$. The gauge parameters $\lambda \in SU(2)$, $\Lambda \in \mathbb{Z}_4$. Finally, $\underline{\partial}\Lambda$ corresponds to the center gauge transformation $\mathbb{Z}_2 \subset SU(2)$ when $\Lambda \in \{1, 3\}$, which is not a part of the $\mathfrak{su}(2)$ Lie algebra.

**Representations**    We investigate the representation of this strict (weak) 2-group. First let us take the 2d fundamental rep. $\pi$ of $SU(2)$. Now note that we can naturally define a unitary projective representation $\tilde{\pi} : SO(3) \to SU(2)$ of $SO(3)$, such that $\pi = \tilde{\pi} \circ p, \tilde{\pi} = s \circ \pi$. For any $\{g, -g\}, \{h, -h\} \in SO(3)$, we have

$$
\tilde{\pi}(\{g, -g\})\tilde{\pi}(\{h, -h\}) = \tilde{\pi}(\{gh, -gh\})f(g, h)\,.
\tag{B.52}
$$

Similar to the computation of Postnikov class, the phase factor $f(g, h) = \pm I$ can also be uplifted to $F(g, h) = 0, 1$ and one can compute the non-trivial $c(g, h, k) \in H^3(BSO(3); \mathbb{Z}_2)$.

If we start from other irreducible representation $\pi$, there are two different scenarios. If $\pi$ is even-dimensional, it corresponds to a projective representation of $SO(3)$, and the above discussions still hold. If $\pi$ is odd-dimensional, it would correspond to a representation of $SO(3)$, hence in (B.52) the phase factor $f(g, h) = I$, and we are unable to generate a non-trivial Postnikov class (the representation is not faithful).

Nonetheless, this is not the notion of 2-representations introduced in the main text.

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
