# Peer review of "Higher-Matter and Landau-Ginzburg Theory of Higher-Group Symmetries"

_SciPost Physics_

## Round 2 · Referee Report · Anonymous (Referee 1) · 2024-11-5

Strengths

This paper serves as a nice reference which connects between the notions of the weak and strict higher group.

Weaknesses

it is unclear (at least to me) what the motivation for physicists is to introduce the notion of the strict higher groups.

Report

This paper studies higher-group symmetries, which are formulated in terms of the strictification of weak higher-groups. The authors start with reviewing and bridging the concepts of strict and weak 2-group symmetries.
The authors then investigate higher-matter charged under 2-group symmetries. They construct a Lagrangian formulation of such higher-matter fields coupled to 2-group gauge fields, which they interpret as the Landau-Ginzburg theory for 2-group symmetries. The paper discusses SSB of 2-group symmetries under this framework. The authors also briefly discuss the strictification of weak 3-groups, weak 3-group gauge fields and 3-representations in special cases.

In quantum field theory, a theory sometimes can have global symmetries generated by the operators with a number of distinct dimensions (p-form symmetry). These symmetries with distinct dimensionality generally does not form a direct product, but rather forms a mixture called a higher group. Physicists typically express the higher-group symmetries using the formalism of ‘weak’ higher-group, while mathematicians seem to use more often the formalism of ‘strict’ higher-group. I believe that this paper serves as a nice reference which connects between the notions of the weak and strict higher group.

Meanwhile, it was still not clear to me what the motivation for physicists is to introduce the notion of the strict higher groups. For instance, the strict 2-group employs the two groups G and H, but I am not sure how these two groups are associated with the symmetry operators that acts on the theory. I assume Pi_j of weak higher groups corresponds to the faithful symmetry generators, but no sense about G and H. I am wondering if the authors can offer some discussions about how the concepts of the strict higher-group corresponds to the operators or correlation functions in the field theory.
So far, I don’t see if there is much benefit of introducing the formalism of strict higher groups. For instance, I feel the most of physical consequence described in the paper (such as the nonsplit 2-group can break into the split 2-group) would be understood using the weak higher-group without employing the strict one.

Also, there are discussions in the paper that the same weak 2-group can correspond to distinct strict higher groups. Do these two distinct choices of the strict higher group give the same global symmetry or not? If not, what is the way/algebraic data to distinguish the distinct global symmetries? I would feel it would be interesting if such two strong symmetries can lead to the distinct algebraic structure of symmetries.
Meanwhile, if two such distinct strict higher groups lead to the same global symmetries, I am again not sure what is the benefit of expressing the symmetry using the strict formalism, since this would just add some redundancy in the expression of global symmetries.

Requested changes

  1. The authors could consider adding some discussions what the objects in the strict higher groups correspond to the operators in the theory, and bring some intuitions for understanding the physical perspective of the strict higher groups.

  2. The authors could clarify if the strict 2-group gives a faithful expression for the algebraic structure of global symmetry? This question is based on the point that two distinct strict groups could lead to the same weak group.

  3. This is the minor point, but the path space is more commonly referred to as a loop space?

Recommendation

Ask for minor revision

---

## Round 2 · Referee Report · Yu-An Chen (Referee 2) · 2024-11-6

Strengths

  1. Novel Approach to Higher-Group Symmetries: The paper presents a systematic study of higher-matter charged under 2-group symmetries, offering an innovative approach that extends the understanding of higher-group symmetries.

  2. Lagrangian Formulation and Landau-Ginzburg Interpretation: The authors provide a Lagrangian formulation of higher-matter fields coupled to 2-group gauge fields, interpreted as a Landau-Ginzburg theory for 2-group symmetries. This is a significant conceptual advancement that connects higher-group symmetries with established field theory methods.

  3. Insights on Spontaneous Symmetry Breaking (SSB): The discussion on spontaneous symmetry breaking (SSB) of 2-group symmetries provides valuable insights, particularly the transition from non-split to split 2-group symmetries, offering a novel perspective on higher-group symmetries.

  4. Exploration of 3-Groups: The brief exploration of the strictification of weak 3-groups, their gauge fields, and 3-representations is noteworthy. This extension indicates the potential for further generalization of the results, and it is a promising direction for future research.

Weaknesses

  1. Lack of a Concrete Lattice Hamiltonian Model: The paper does not provide a concrete lattice Hamiltonian as a toy model featuring higher matter and higher-group symmetries. Including such a model would enhance the clarity and applicability of the theoretical framework.

Report

The paper "Higher-Matter and Landau-Ginzburg Theory of Higher-Group Symmetries" is an outstanding contribution to the field of higher-group symmetries and higher-representation matter. It offers a novel and comprehensive approach to understanding higher-matter charged under 2-group symmetries. The authors employ the strictification of weak higher-groups and utilize automorphism 2-representations, which significantly advances the theoretical understanding of higher-group symmetries. The paper's exploration of a Lagrangian formulation of higher-matter fields, interpreted as a Landau-Ginzburg theory, is a particularly noteworthy achievement. This provides a concrete physical interpretation that bridges the gap between abstract higher-group symmetries and established field theory methods.

The insights into spontaneous symmetry breaking (SSB) of 2-group symmetries are highly valuable. The discovery that non-split 2-group symmetries can undergo SSB to a split 2-group symmetry, with the Postnikov class becoming trivialized, provides a fresh perspective on the interplay between different types of higher-group symmetries. Furthermore, the extension of the analysis to 3-groups, including the discussion on strictification, gauge fields, and 3-representations, indicates the potential for further generalization and opens up exciting directions for future research.

Although the paper could benefit from the inclusion of a concrete lattice Hamiltonian model to provide a more tangible illustration of the theoretical framework, this does not diminish the significance of the work. The paper is rigorous and insightful and presents substantial advancements in the understanding of higher-group symmetries and higher-representation matter.

I strongly recommend this paper for publication. It is a well-written and comprehensive work that will undoubtedly have a significant impact on the field and inspire further research in higher-group symmetries and related areas.

Requested changes

None

Recommendation

Publish (surpasses expectations and criteria for this Journal; among top 10%)

---

## Round 2 · Referee Report · Anonymous (Referee 3) · 2024-12-3

Report

This paper studied the strictifications and the representations of higher-group symmetries and higher-group gauge fields and use these results to develop the Landau-Ginzburg theory for higher-group symmetries. It started by connecting the two complementary formulations of 2-groups: strict 2-groups and weak 2-groups. It then applied the procedure of strictifications to higher-form gauge fields. Next, it discussed the representation theories of 2-groups based on automorphism 2-representations. In the end, it developed the Landau-Ginzburg theory for 2-groups by coupling higher-matter fields to higher-group gauge field in the path space of the spacetime manifold, which reproduced the relation between area/perimeter law of line operators and spontaneous symmetry breaking of 2-group symmetry. Along the way, it also discussed the generalization to 3-groups.

Overall, the paper is solid and well-written, making it a valuable contribution to the literature on higher-group symmetry and a useful reference for strict higher-groups and path space formulations of gauge theories. The referee recommends the publication of this paper.

Minor comments and questions that authors can choose to address, although it is not necessary:

1. In physically relevant context, $\Pi_2$ in the weak 2-group is always an abelian group because it corresponds to a one-form symmetry. However, in the strict 2-group construction of weak 2-group, it seems that $\Pi_2$ can generally be non-abelian. Is it correct? If so, is there any physical context where these “non-abelian” 2-groups appear?

2. In Section 2.2, when the authors discuss the general procedure of strictification, how does the group action $\rho$ enter into the strictification?

3. It is nice that the authors provide a physical interpretation of the automorphism 2-representations in terms of the Wilson lines in the corresponding gauge theories. It would further improve the paper if the authors could include more discussions on the physical meaning of the strictification of the 2-group and 3-group gauge fields. To the referee’s knowledge, arXiv:1309.4721 includes some preliminary discussions along this line for 2-groups, which might be helpful.

Recommendation

Publish (easily meets expectations and criteria for this Journal; among top 50%)

---

## Editorial Decision

resubmitted